# $\alpha$-LoRA: Effective Fine-Tuning via Base Model Rescaling

## Abstract

Fine-tuning has proven to be highly effective in adapting pre-trained models to perform better on new desired tasks with minimal data samples. Among the most widely used approaches are reparameterization methods, which update a target module by augmenting its frozen weight matrix with an additional trainable weight matrix. The most prominent example is Low Rank Adaption (LoRA) (Hu et al., 2022), which gained significant attention in recent years. In this paper, we introduce a new class of reparametrization methods for transfer learning, designed to enhance the generalization ability of fine-tuned models. We establish the effectiveness of our approach in a high-dimensional binary classification setting using tools from Random Matrix Theory, and further validate our theoretical findings through more realistic experiments, such as fine-tuning large language models.

## 1 Introduction

Large foundational models have driven major advances in artificial intelligence across domains such as computer vision and natural language processing. Examples include transformer-based models (Vaswani et al., 2017) operating in natural language domain (Team et al., 2023; Grattafiori et al., 2024) or vision domain (Cordonnier et al., 2020; Dosovitskiy et al., 2020). Such models are specifically known for their relatively large size and massive training corpus, which makes them more powerful and adapted for many use cases. However, even with their extensive pre-training, these large models may not excel at some specific tasks without further adjustment.

Fine-tuning addresses this need by updating a pre-trained model with task-specific data. Unlike training from scratch, it leverages general pre-trained representations while reducing data and compute requirements. The most common class of fine-tuning methods is Supervised Fine-Tuning (SFT), which relies on labeled data in that adaptation process, and one of its most popular lightweight techniques is Low-Rank Adaptation (LoRA) (Hu et al., 2022), which updates the desired module by adding a low-rank perturbation to the original (frozen) weight matrix.

In this paper, we study fine-tuning through the lens of Random Matrix Theory (RMT), where we introduce a theoretical framework to understand and improve transfer learning. Leveraging the theoretical findings, our key practical idea in the context of LoRA is to scale the frozen weights row-wise with a vector $\alpha$ before adaptation, thereby adding a new degree of freedom to the fine-tuning process. We show that this modification leads to an optimal scaling factor $\alpha^*$, which is typically different from the standard choice ($\alpha = 1$). We analyze this framework in a high-dimensional binary classification setting under a Gaussian Mixture Model, proving the existence of such an optimal $\alpha^*$ while providing its closed-form expression in terms of scalar data-dependent quantities. We then validate our theoretical insights on real tasks, including transfer learning benchmarks and large language model fine-tuning.

**Summary of contributions.** Our main contributions are summarized as follows:

1. In the context of adaptation fine-tuning (e.g., LoRA), we propose the scaling of the base model weight matrices by a non-trivial row-wise vector $\alpha$.

2. We theoretically prove the existence of an optimal parameter $\alpha^* \neq 1$ in high-dimensional binary classification and derive its closed form.

3. We design an algorithm to estimate optimal $\alpha$ in complex scenarios such as LLM fine-tuning.

## 2 RELATED WORK

**Transfer learning foundations.** Transfer Learning (TL) studies how knowledge acquired in a source task or domain can be reused to improve learning in a related target task. Early surveys (Pan & Yang, 2009; Weiss et al., 2016) outlined key settings such as domain adaptation and multitask learning. Most theoretical works established generalization bounds linking transfer success to source error and distributional divergence Ben-David et al. (2010), and showed how shared representations reduce sample complexity (Maurer et al., 2016; Tripuraneni et al., 2020). More recent studies refined these results under classification and regression settings (Hanneke & Kpotufe, 2024; Zhang et al., 2021; Klivans et al., 2024; Kpotufe & Martinet, 2021; Cai & Wei, 2021; Reeve et al., 2021).

**Fine-tuning pre-trained models.** With the advent of large-scale pre-training, fine-tuning has become the dominant strategy for transfer learning. The most popular fine-tuning techniques are Supervised Fine-tuning (SFT) and fine-tuning with Reinforcement Learning (RL). RL-based approaches such as RLHF (Ouyang et al., 2022), DPO (Rafailov et al., 2023), GRPO (Ramesh et al., 2024; Guo et al., 2025) and their variants are especially popular for reasoning and mathematics tasks, where they often outperform SFT (Shenfeld et al., 2025). In this paper, however, we focus on SFT techniques. SFT extends the training of a pre-trained model using labeled data. Because these models are typically very large, it is common to fine-tune only a small fraction of their parameters while leaving most unchanged. This strategy, known as Parameter-Efficient Fine-Tuning (PEFT) (Xu et al., 2023), aims to achieve strong performance with minimal parameter updates. PEFT methods are usually grouped into three categories: additive, selective, and reparametrized (Ji et al., 2025). Our work centers on the last category.

**Reparametrized Fine-tuning.** Reparameterization-based fine-tuning adapts a model by expressing its parameters in an alternative form, commonly through a low-rank decomposition, to reduce training costs, while the full weight matrices are reconstructed for inference. The most common technique in this class is Low Rank Adaptation (LoRA) (Hu et al., 2022), which introduces small, trainable matrices operating alongside the pre-trained weights to inject task-specific updates without burdening the inference process. Many extensions were proposed to enhance the efficiency of LoRA by either acting on the initialization of the low rank modules (Hayou et al., 2024a), their learning rates (Hayou et al., 2024b), normalizing the updates (Liu et al., 2024), setting adaptive ranks (Kim et al., 2024; Lu et al., 2024), finding optimal placements for LoRA modules (Hayou et al., 2025), and more (Zhang et al., 2023b; Dettmers et al., 2023; Kopiczko et al., 2023; Zhang et al., 2023a; Tian et al., 2024; Jiang et al., 2024).

## 3 PROBLEM SETTING AND BACKGROUND

To prove the effectiveness of our new family of fine-tuning algorithms, we will theoretically analyze a binary classification setting under a Gaussian Mixture Model (GMM) using tools from Random Matrix Theory (RMT). Through this analysis, we will prove the existence of an optimal scaling parameter $\alpha^\star$ and derive its exact theoretical formulation for these settings.

### 3.1 THEORETICAL SETTING

The goal is to fine-tune a linear classifier, initially pretrained on a dataset called **source**, in order to perform a **target** task given a relatively small target data corpus.

**Pre-training phase.** We consider that we are given pairs of pre-training (source) data samples $\{(\tilde{\boldsymbol{x}}_i, \tilde{y}_i)\}_{i=1}^N$ that are distributed, for $\tilde{\boldsymbol{x}}_i \in \mathcal{C}_a$ with $a \in \{1, 2\}$, as follows:

$$\tilde{\boldsymbol{x}}_i \in \mathcal{C}_a \quad \Leftrightarrow \quad \begin{cases} \tilde{\boldsymbol{x}}_i = \boldsymbol{\mu}_a + \tilde{\boldsymbol{z}}_i, & \tilde{\boldsymbol{z}}_i \sim \mathcal{N}(\mathbf{0}, \mathbf{I}_p), \\ \tilde{y}_i = (-1)^a. \end{cases} \tag{1}$$

For convenience and without loss of generality, we further assume that $\boldsymbol{\mu}_a = (-1)^a \boldsymbol{\mu}$ for some vector $\boldsymbol{\mu} \in \mathbb{R}^p$. This setting can be recovered by subtracting $\frac{\boldsymbol{\mu}_1 + \boldsymbol{\mu}_2}{2}$ from each data point, as such $\boldsymbol{\mu} = \frac{\boldsymbol{\mu}_2 - \boldsymbol{\mu}_1}{2}$ and therefore the SNR $\|\boldsymbol{\mu}\|$ controls the difficulty of the classification problem, in the sense that large values of $\|\boldsymbol{\mu}\|$ yield a simple classification problem whereas when $\|\boldsymbol{\mu}\| \to 0$,

the classification becomes impossible. Denoting $\tilde{\mathbf{X}} = [\tilde{\boldsymbol{x}}_1, \ldots, \tilde{\boldsymbol{x}}_N] \in \mathbb{R}^{p \times N}$ the data matrix and $\tilde{\boldsymbol{y}} = [\tilde{y}_1, \ldots, \tilde{y}_N]^\top \in \mathbb{R}^N$ the corresponding labels vector, we have in matrix form

$$\tilde{\mathbf{X}} = \boldsymbol{\mu}\tilde{\boldsymbol{y}}^\top + \tilde{\mathbf{Z}}, \tag{2}$$

where $\tilde{\mathbf{Z}}$ is a random matrix with $\mathcal{N}(0,1)$ i.i.d. entries.
We then consider training a classifier, called $\tilde{\boldsymbol{w}}$, on this source dataset by solving:

$$\min_{\boldsymbol{w}} \frac{1}{N} \sum_{i=1}^{N} \ell(\boldsymbol{w}^\top \boldsymbol{x}_i, y_i) + \tilde{\gamma}\|\boldsymbol{w}\|_2^2 \tag{3}$$

for some loss function $\ell$ and a positive regularization parameter $\tilde{\gamma}$. Taking a generic or a non-intuitive loss, such as the binary cross entropy, leads to **intractable** solution $\tilde{\boldsymbol{w}}$. However, Mai & Liao (2024) show that in the case of a Gaussian mixture data model or more generally a data distribution with finite fourth-order moment, it is possible to optimize such a classifier using the squared ($L^2$) loss function, which also gives a closed-form solution to this problem. Thus, taking $\ell(x, y) = (x - y)^2$ leads to the following optimization problem:

$$\tilde{\boldsymbol{w}} = \arg\min_{\boldsymbol{v}} \frac{1}{N} \left\| \tilde{\mathbf{X}}^\top \boldsymbol{v} - \tilde{\boldsymbol{y}} \right\|_2^2 + \tilde{\gamma}\|\boldsymbol{v}\|_2^2, \tag{4}$$

Which gives us the following solution:

$$\tilde{\boldsymbol{w}} = \frac{1}{N} \mathbf{R}\tilde{\mathbf{X}}\tilde{\boldsymbol{y}}, \quad \mathbf{R} = \left( \frac{1}{N}\tilde{\mathbf{X}}\tilde{\mathbf{X}}^\top + \tilde{\gamma}\mathbf{I}_p \right)^{-1} \tag{5}$$

**Fine-tuning phase.** During the fine-tuning phase, we suppose that we are given pairs of target data $\{(\boldsymbol{x}_i, y_i)\}_{i=1}^n$ with $y_i \in \{-1, 1\}$ that are distributed such that $\mathbf{X} = [\boldsymbol{x}_1, \ldots, \boldsymbol{x}_n] \in \mathbb{R}^{p \times n}$ is given by:

$$\mathbf{X} = \boldsymbol{\mu}_\beta \boldsymbol{y}^\top + \mathbf{Z}, \quad \boldsymbol{\mu}_\beta = \beta\boldsymbol{\mu} + \boldsymbol{\mu}^\perp, \tag{6}$$

where $\mathbf{Z}$ is a random matrix with $\mathcal{N}(0,1)$ i.i.d. entries, $\boldsymbol{\mu}^\perp$ is an orthogonal vector to $\boldsymbol{\mu}$ and the factor $\beta \in \mathbb{R}$ quantifies the **alignment** between the source and target data, as we have that: $\langle \boldsymbol{\mu}_\beta, \boldsymbol{\mu} \rangle = \beta\|\boldsymbol{\mu}\|^2$. Leveraging the pre-trained weights $\tilde{\boldsymbol{w}} \in \mathbb{R}^p$, we consider the training of adapter weights $\boldsymbol{a}$ as:

$$\boldsymbol{a} = \arg\min_{\boldsymbol{v}} \frac{1}{n} \left\| \mathbf{X}^\top (\alpha\tilde{\boldsymbol{w}} + \boldsymbol{v}) - \boldsymbol{y} \right\|_2^2 + \gamma\|\boldsymbol{v}\|_2^2, \tag{7}$$

for a scalar $\alpha \in \mathbb{R}$. In fact, classical reparametrization approaches can be modeled by the same setting using $\alpha = 1$. Solving the previous minimization problem, $\boldsymbol{a}$ expresses as:

$$\boldsymbol{a} = \frac{1}{n} \left( \frac{1}{n}\mathbf{X}\mathbf{X}^\top + \gamma\mathbf{I}_p \right)^{-1} \left( \mathbf{X}\boldsymbol{y} - \alpha\mathbf{X}\mathbf{X}^\top\tilde{\boldsymbol{w}} \right). \tag{8}$$

We define the resolvent matrices $\mathbf{Q}$ and $\mathbf{R}$ by:

$$\mathbf{Q} = \left( \frac{1}{n}\mathbf{X}\mathbf{X}^\top + \gamma\mathbf{I}_p \right)^{-1}, \quad \mathbf{R} = \left( \frac{1}{N}\tilde{\mathbf{X}}\tilde{\mathbf{X}}^\top + \tilde{\gamma}\mathbf{I}_p \right)^{-1}, \tag{9}$$

Then our obtained fine-tuned classifier $\boldsymbol{w}_\alpha$ writes:

$$\boldsymbol{w}_\alpha = \alpha\tilde{\boldsymbol{w}} + \boldsymbol{a} = \frac{1}{n}\mathbf{Q}(\gamma)\mathbf{X}\boldsymbol{y} + \alpha\gamma\mathbf{Q}\tilde{\boldsymbol{w}}$$

We denote by $\boldsymbol{w} \equiv \boldsymbol{w}_0$ the classifier obtained through learning directly on target data (without fine-tuning), which is given by:

$$\boldsymbol{w} = \frac{1}{n}\mathbf{Q}(\gamma)\mathbf{X}\boldsymbol{y} \tag{No-FT}$$

Then we finally get the expression of our $\alpha$-Fine-tuned classifier as follows:

$$\boldsymbol{w}_\alpha = \boldsymbol{w} + \alpha\gamma\mathbf{Q}\tilde{\boldsymbol{w}} \tag{$\alpha$-FTC}$$

**Remark 3.1** (About the interpretability of our fine-tuned classifier). *Remark that the parameter $\alpha$ introduced in the expression of the fine-tuned classifier $\boldsymbol{w}_\alpha$ characterizes the contribution of each training dataset (source and target) to the test performance on the target task. In fact, since the prediction of the class label does not change by multiplying $\boldsymbol{w}_\alpha$ by a positive constant, then by taking a positive $\alpha$ and for $\rho = \frac{\alpha}{1+\alpha} \in (0,1)$, the fine-tuned classifier is equivalent to this convex weighted classifier:*

$$\boldsymbol{w}_\rho = \rho \tilde{\boldsymbol{w}} + (1-\rho)\boldsymbol{a}$$

*and therefore, this new parameter $\rho$ can be interpreted as the percentage of the contribution of the source task to the test performance on the target task.*

**Remark 3.2** (About the regularization parameter $\gamma$). *We remark from the expression of $\boldsymbol{w}_\alpha$ in equation $\alpha$-FTC that the weight decay $\gamma$ is essential to have the dependence of $\boldsymbol{w}_\alpha$ on $\alpha$. In fact, taking $\gamma \to 0$ leads to a fine-tuned classifier of the form:*

$$\boldsymbol{w}_\alpha = (\mathbf{X}\mathbf{X}^\top)^+ \mathbf{X}\boldsymbol{y}$$

*where $(\mathbf{X}\mathbf{X}^\top)^+$ is the Moore-Penrose inverse of the symmetric semi-definite matrix $\mathbf{X}\mathbf{X}^\top$. Therefore, the obtained classifier does **not** depend on $\alpha$ here, nor on the pre-trained model $\tilde{\boldsymbol{w}}$. Additionally, having such a regularization technique is essential in transfer learning since the target dataset is generally much smaller than the pre-training one, and therefore the fine-tuning process can easily lead to overfitting in the absence of a regularization technique.*

## 3.2 RMT Background

To analyze the performance of the fine-tuned classifier $\boldsymbol{w}_\alpha$, we can leverage tools from Random Matrix Theory. In mathematical terms, the understanding of the asymptotic performance of the classifier $\boldsymbol{w}_\alpha$ boils down to the characterization of the statistical behavior of the *resolvent matrices* $\mathbf{Q}(z)$ and $\mathbf{R}(z)$ introduced in equation 9. In the following, we will recall some important notions and results from random matrix theory, which will be at the heart of our analysis. We start by defining the main object, which is the resolvent matrix.

**Definition 3.3** (Resolvent). *For a symmetric matrix $\mathbf{M} \in \mathbb{R}^{p \times p}$, the resolvent $\mathbf{Q}_M(z)$ of $\mathbf{M}$ is defined for $z \in \mathbb{C} \backslash \mathcal{S}(\mathbf{M})$ as:*

$$\mathbf{Q}_M(z) = (\mathbf{M} - z\mathbf{I}_p)^{-1},$$

*where $\mathcal{S}(\mathbf{M})$ is the set of eigenvalues or spectrum of $\mathbf{M}$.*

In fact, the study of the asymptotic performance of $\boldsymbol{w}_\alpha$ involves the estimation of linear forms of the resolvents $\mathbf{Q}$ and $\mathbf{R}$ in equation 9, such as $\frac{1}{n}\operatorname{Tr}\mathbf{Q}$ and $\boldsymbol{a}^\top \mathbf{Q}\boldsymbol{b}$ with $\boldsymbol{a}, \boldsymbol{b} \in \mathbb{R}^p$ of bounded Euclidean norms. Therefore, the notion of a *deterministic equivalent* (Hachem et al., 2007) is crucial as it allows the design of a **deterministic** matrix, having (in probability or almost surely) asymptotically the same *scalar observations* as the random ones in the sense of *linear forms*. A rigorous definition is provided below.

**Definition 3.4** (Deterministic equivalent (Hachem et al., 2007)). *We say that $\bar{\mathbf{Q}} \in \mathbb{R}^{p \times p}$ is a deterministic equivalent for the random resolvent matrix $\mathbf{Q} \in \mathbb{R}^{p \times p}$ if, for any bounded linear form $u : \mathbb{R}^{p \times p} \to \mathbb{R}$, we have that, as $p \to \infty$:*

$$u(\mathbf{Q}) \xrightarrow{a.s.} u(\bar{\mathbf{Q}}),$$

*where the convergence is in the almost sure sense.*

In particular, a deterministic equivalent for the resolvents $\mathbf{Q}(z)$ and $\mathbf{R}(z)$ defined in equation 9 is given by the following Lemma (the proof is presented in Appendix A.2).

**Lemma 3.5** (Deterministic equivalent of $\mathbf{Q}$ and $\mathbf{R}$). *Under the high-dimensional regime, when $p, n, N \to \infty$ with $\frac{p}{n} \to \eta \in (0,\infty)$ and $\frac{p}{N} \to \tilde{\eta} \in (0,\infty)$ and assuming $\|\boldsymbol{\mu}\| = \mathcal{O}(1)$, a deterministic equivalent for $\mathbf{Q} \equiv \mathbf{Q}(\gamma)$ and for $\mathbf{R} \equiv \mathbf{R}(\gamma)$, previously defined in equation 9, denoted $\bar{\mathbf{Q}}$ and $\bar{\mathbf{R}}$ respectively, are given by:*

$$\bar{\mathbf{Q}}(\gamma) = \left( \frac{\boldsymbol{\mu}_\beta \boldsymbol{\mu}_\beta^\top + \mathbf{I}_p}{1 + \delta_Q} + \gamma \mathbf{I}_p \right)^{-1}, \quad \bar{\mathbf{R}}(\gamma) = \left( \frac{\boldsymbol{\mu}\boldsymbol{\mu}^\top + \mathbf{I}_p}{1 + \delta_R} + \gamma \mathbf{I}_p \right)^{-1}.$$

*Where:*

$$\delta_Q = \frac{1}{n}\operatorname{Tr}\bar{\mathbf{Q}} = \frac{\eta - \gamma - 1 + \sqrt{(\eta - \gamma - 1)^2 + 4\eta\gamma}}{2\gamma}, \quad \delta_R = \frac{\tilde{\eta} - \tilde{\gamma} - 1 + \sqrt{(\tilde{\eta} - \tilde{\gamma} - 1)^2 + 4\tilde{\eta}\tilde{\gamma}}}{2\tilde{\gamma}}.$$

## 4 MAIN RESULTS

After having defined the setting and needed background, we will now present our main technical results, which describe the asymptotic behavior of the fine-tuned classifier defined in equation $\alpha$-FTC. Specifically, we provide our results under the following growth rate assumptions (classical assumptions in Random Matrix Theory).

**Assumption 4.1** (Growth Rates). *Suppose that as $p, n, N \to \infty$:*

$$1) \ \frac{p}{n} \to \eta \in [0, \infty), \qquad 2) \ \frac{p}{N} \to \tilde{\eta} \in [0, \infty), \qquad 3) \ \|\boldsymbol{\mu}\| = \mathcal{O}(1), \qquad 4) \ \|\boldsymbol{\mu}_\beta\| = \mathcal{O}(1).$$

The first and second assumptions simply state that our analysis considers both the low ($\eta, \tilde{\eta} \ll 1$) and high ($\eta, \tilde{\eta} \gg 1$) dimensional regimes. The third and last assumptions are also fundamental and state that the norm of the source $\boldsymbol{\mu}$ and target $\boldsymbol{\mu}_\beta$ data means do not scale with the dimension $p$, which makes the classification problem neither easy ($\|\boldsymbol{\mu}\| \to \infty$) nor impossible ($\|\boldsymbol{\mu}\| \to 0$) in high dimensions. Having stated the main assumptions, we are now in a position to present our main technical findings about the theoretical test performance of the fine-tuned classifier $\alpha$-FTC. But beforehand, let us define some scalar quantities that will be useful in our derivations:

$$\lambda_Q = \|\boldsymbol{\mu}_\beta\|^2 + 1 + \gamma(1 + \delta_Q), \quad \lambda_R = \|\boldsymbol{\mu}\|^2 + 1 + \tilde{\gamma}(1 + \delta_R), \quad h = 1 - \frac{\eta}{(1 + \gamma(1 + \delta_Q))^2},$$

$$\tilde{h} = 1 - \frac{\tilde{\eta}}{(1 + \tilde{\gamma}(1 + \delta_R))^2}$$

Our main theorem below describes the behavior of the decision function of our fine-tuned classifier.

**Theorem 4.2** (Gaussianity of the fine-tuned Ridge model). *Let $\boldsymbol{w}_\alpha$ be the fine-tuned classifier as defined in equation $\alpha$-FTC and suppose that Assumption 4.1 holds. The decision function $\boldsymbol{w}_\alpha^\top \boldsymbol{x}$, on some test sample $\boldsymbol{x} \in \mathcal{C}_a$ independent of $\mathbf{X}$, satisfies:*

$$\boldsymbol{w}_\alpha^\top \boldsymbol{x} \xrightarrow{\mathcal{D}} \mathcal{N}\left((-1)^a m_\alpha, \nu_\alpha - m_\alpha^2\right),$$

*where:*

$$m_\alpha = \frac{1}{\lambda_Q}\left(\|\boldsymbol{\mu}_\beta\|^2 + \frac{\alpha\beta\gamma(1 + \delta_Q)}{\lambda_R}\|\boldsymbol{\mu}\|^2\right),$$

$$\nu_\alpha = T_1 + \alpha T_2 + \alpha^2 T_3.$$

*With:*

$$T_1 = \frac{\|\boldsymbol{\mu}_\beta\|^2}{h\lambda_Q}\left(\frac{\|\boldsymbol{\mu}_\beta\|^2 + 1}{\lambda_Q} - 2(1 - h)\right) + \frac{1 - h}{h},$$

$$T_2 = \frac{2\gamma\beta(1 + \delta_Q)\|\boldsymbol{\mu}\|^2}{\lambda_R\lambda_Q}\left(1 - \frac{\gamma(1 + \delta_Q)}{h\lambda_Q}\right),$$

$$T_3 = \frac{\gamma^2(1 + \delta_Q)^2}{h} \times$$

$$\left[\frac{\|\boldsymbol{\mu}\|^2}{\lambda_R^2}\left(\frac{\beta^2\|\boldsymbol{\mu}\|^2}{\lambda_Q^2} + \frac{1 - h}{\eta}\left(1 + \frac{\beta^2\|\boldsymbol{\mu}\|^2\|\boldsymbol{\mu}_\beta\|^2}{\lambda_Q^2} - \frac{2\beta^2\|\boldsymbol{\mu}\|^2}{\lambda_Q} + (1 - \tilde{h})\left(1 - \frac{2\|\boldsymbol{\mu}\|^2}{\lambda_R}\right)\right)\right)\right]$$

In simple terms, Theorem 4.2 states that the decision function of the classifier in equation $\alpha$-FTC is asymptotically equivalent to the thresholding of two monovariate Gaussian random variables with respective means $m_\alpha$ and $-m_\alpha$ and standard deviation $\nu_\alpha - m_\alpha^2$, where the statistics $m_\alpha$ and $\nu_\alpha$ are expressed in terms of the scalar quantities defined above (see Figure 6 in the Appendix). Having characterized the distribution of the decision function of $\boldsymbol{w}_\alpha$, we can now estimate its generalization performance, such as its test accuracy.

**Proposition 4.3** (Asymptotic test accuracy of $\boldsymbol{w}_\alpha$). *The asymptotic test accuracy of $\boldsymbol{w}_\alpha$ defined in equation $\alpha$-FTC, under Assumption 4.1, and as the number of test samples $n_{test} \to \infty$, is given by:*

$$\mathcal{A}_{test} \xrightarrow{a.s.} 1 - \varphi\left((\nu_\alpha - m_\alpha^2)^{-\frac{1}{2}}m_\alpha\right), \quad \text{where: } \varphi(x) = \frac{1}{\sqrt{2\pi}}\int_x^{+\infty} e^{-\frac{t^2}{2}}\, \mathrm{dt}.$$

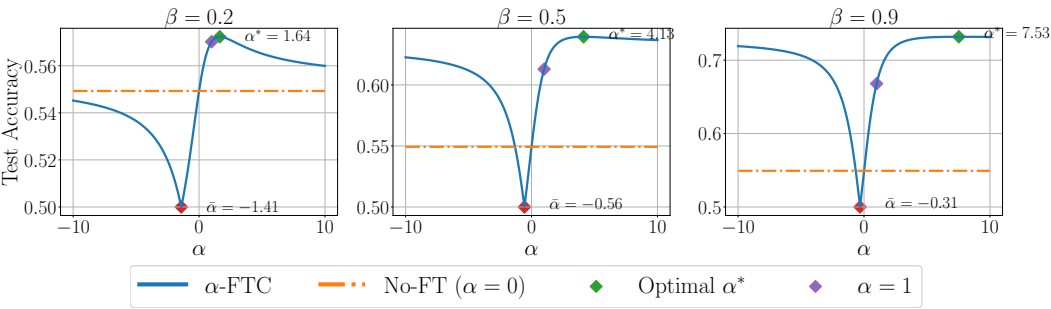

Figure 1: Theoretical Test Accuracy variation with $\alpha$ for $N = 5000$, $n = 40$, $p = 1000$, and the theoretical model is modified to take $\beta$ in $(0,1)$: $\boldsymbol{\mu}_\beta = \beta\boldsymbol{\mu} + \sqrt{1-\beta^2}\boldsymbol{\mu}^\perp$, where $\|\boldsymbol{\mu}\| = \|\boldsymbol{\mu}^\perp\| = 0.8$. Finally the regularization parameters are: $\tilde{\gamma} = 2$ and $\gamma = 10^{-1}$.

Therefore, thanks to Proposition 4.3, we now have the exact formulas of the theoretical test accuracy of our classifier $\boldsymbol{w}_\alpha$, which can be used to characterize the expression of the optimal/worst parameters of the model (for instance, the $\alpha$) to use for the fine-tuning process. In particular, we will derive the theoretical expressions of the extremum of $\alpha$ that lead to either the best or the worst test accuracy on the target task (proof in Appendix B).

**Theorem 4.4** (Optimal $\alpha$). *Maximizing the term* $\left((\nu_\alpha - m_\alpha^2)^{-\frac{1}{2}} m_\alpha\right)$ *in terms of $\alpha$ leads to maximizing the test accuracy $\mathcal{A}_{test}$, and gives a unique maximizer $\alpha^\star$ given by:*

$$\alpha^\star = \frac{\lambda_R T_2 \|\boldsymbol{\mu}_\beta\|^2 - 2\beta\gamma T_1(1+\delta_Q)\|\boldsymbol{\mu}\|^2}{\beta\gamma T_2(1+\delta_Q)\|\boldsymbol{\mu}\|^2 - 2\lambda_R T_3\|\boldsymbol{\mu}_\beta\|^2}$$

*Plus, solving $(\nu_\alpha - m_\alpha^2)^{-\frac{1}{2}} m_\alpha = 0$ leads to the unique minimizer $\bar{\alpha}$ of $\mathcal{A}_{test}$, which is given by:*

$$\bar{\alpha} = -\frac{\lambda_R \|\boldsymbol{\mu}_\beta\|^2}{\beta\gamma(1+\delta_Q)\|\boldsymbol{\mu}\|^2}$$

Figure 1 shows the evolution of the theoretical test accuracy with the parameter $\alpha$ for different source datasets (i.e, different alignments $\beta$). In particular, we observe the existence of an optimal parameter $\alpha^\star$ that is generally different from 1 (standard approach), and as can be previously anticipated, its impact on the test accuracy is more visible in the case of a higher alignment factor $\beta$, which means in this case that we put higher emphasis on the base model to generalize better in the new task (see Remark 3.1).

Focusing on the optimal $\alpha^\star$, Figure 2 clearly depicts the non-trivial contribution of the dimension $p$ to the choice of $\alpha$. It is clear that $\alpha^\star$ is non-decreasing with the alignment $\beta$ between the source and target task, but its effect

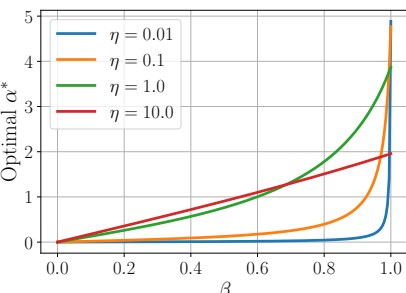

Figure 2: Variations of the optimal parameter $\alpha^\star$ with respect to the alignment between the source $\boldsymbol{\mu}$ and target $\boldsymbol{\mu}_\beta$ dataset means. These latter were chosen of norm 1, $N = 2000$, $n = 200$ and $\gamma = \tilde{\gamma} = 1$.

gets amplified with the dimension $p$ of the problem. Notably, the influence of $\alpha$ is more pronounced in low-resource settings ($p \gg n$) compared to cases where sufficient fine-tuning data is available. This further underscores the crucial role of $\alpha$ in effectively leveraging the pre-trained model and source data. Additionally, as $\beta \to 0$, we also remark that $\alpha^\star \to 0$, which means that fine-tuning has no added value when the source and target tasks are **unrelated** and orthogonal.

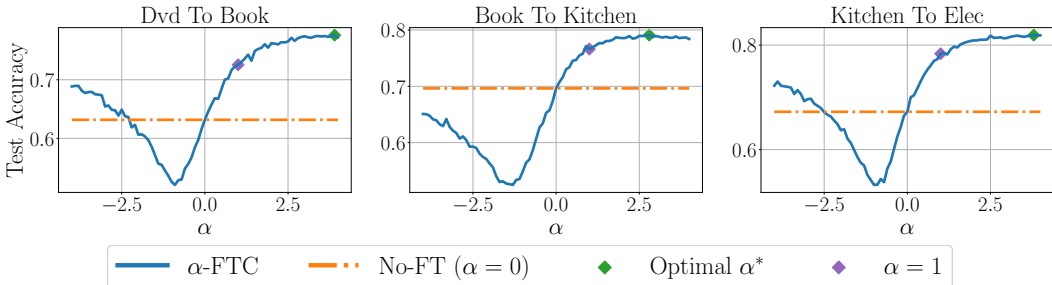

Figure 3: Test accuracy variation with $\alpha$ for different transfer learning schemes from the Amazon Review dataset (Blitzer et al., 2007). The considered parameters here are: $N = 2000$, $n = 40$, $p = 400$, $\gamma = 10^{-1}$ and $\tilde{\gamma} = 2$.

## 5 EXPERIMENTS

In this section, we present some experiments on real datasets to validate our approach. We start by fine-tuning linear models on the Amazon Review dataset (Blitzer et al., 2007) to verify our theoretical findings. After that, we formalize our new class of reparametrization methods and verify its efficiency by experiments on fine-tuning LLMs on the GLUE tasks (Wang et al., 2018).

### 5.1 WITHIN OUR THEORETICAL MODEL: LINEAR BINARY CLASSIFICATION

Here we present our experiments on the Amazon Review dataset (Blitzer et al., 2007) to validate our theory. This dataset includes several binary classification tasks corresponding to positive versus negative reviews of `books`, `dvd`, `electronics`, and `kitchen`. We apply the standard scaler from `scikit-learn` (Pedregosa et al., 2011) and estimate $\|\boldsymbol{\mu}\|$, $\|\boldsymbol{\mu}^{\perp}\|$ and $\beta$ with the normalized data. Figure 3 depicts the variation in test accuracy of three transfer tasks with respect to the parameter $\alpha$ and gives a comparison between the three main schemes: $\alpha = 0$ (i.e., learning directly on the target data without using previous source knowledge), $\alpha = 1$ (classical approach) and with the optimal $\alpha^{\star}$ obtained using the theoretical formula in Theorem 4.4. Depending on the tasks, we see a clear improvement in the test accuracy for $\alpha^{\star}$ compared to the other schemes, which further highlights the impact of this scaling parameter. Table 1 summarizes the results obtained for all the possible transfer tasks between the sub-datasets.

Table 1: Test accuracy (in %) comparison over Amazon review datasets (Blitzer et al., 2007) for $N = 2000$, $n = 40$, $p = 400$, and optimal regularization parameters $\gamma = \tilde{\gamma} = 1$. As theoretically anticipated, our new fine-tuning approach yields better classification accuracy than training directly on the target dataset ($\alpha = 0$) or using $\alpha = 1$. The results were computed for 3 random seeds.

| Source Dataset | Target Dataset | $\alpha = 0$ | $\alpha = 1$ | Optimal $\alpha^{\star}$ |
|---|---|---|---|---|
| Books | Dvd ($\beta = 0.8$) | $64.12 \pm 0.03$ | $75.67 \pm 0.24$ | $77.35 \pm 0.14$ ($\alpha^{\star} = 2.47$) |
| | Electronics ($\beta = 0.71$) | $68.61 \pm 0.74$ | $76.65 \pm 0.02$ | $77.12 \pm 0.17$ ($\alpha^{\star} = 1.68$) |
| | Kitchen ($\beta = 0.79$) | $69.24 \pm 0.95$ | $78.19 \pm 0.05$ | $78.96 \pm 0.26$ ($\alpha^{\star} = 1.9$) |
| Dvd | Books ($\beta = 0.78$) | $63.43 \pm 0.67$ | $75.22 \pm 0.24$ | $77.59 \pm 0.07$ ($\alpha^{\star} = 2.47$) |
| | Electronics ($\beta = 0.71$) | $68.61 \pm 0.74$ | $76.72 \pm 0.17$ | $76.88 \pm 0.42$ ($\alpha^{\star} = 1.69$) |
| | Kitchen ($\beta = 0.78$) | $69.24 \pm 0.95$ | $78.11 \pm 0.23$ | $78.72 \pm 0.54$ ($\alpha^{\star} = 1.88$) |
| Electronics | Books ($\beta = 0.51$) | $63.43 \pm 0.67$ | $72.2 \pm 0.1$ | $73.29 \pm 0.13$ ($\alpha^{\star} = 1.67$) |
| | Dvd ($\beta = 0.52$) | $64.12 \pm 0.03$ | $72.41 \pm 0.16$ | $73.48 \pm 0.17$ ($\alpha^{\star} = 1.69$) |
| | Kitchen ($\beta = 0.9$) | $69.24 \pm 0.95$ | $81.58 \pm 0.15$ | $83.02 \pm 0.1$ ($\alpha^{\star} = 2.29$) |
| Kitchen | Books ($\beta = 0.52$) | $63.43 \pm 0.67$ | $72.86 \pm 0.1$ | $74.27 \pm 0.14$ ($\alpha^{\star} = 1.84$) |
| | Dvd ($\beta = 0.53$) | $64.12 \pm 0.03$ | $73.15 \pm 0.08$ | $74.15 \pm 0.09$ ($\alpha^{\star} = 1.82$) |
| | Electronics ($\beta = 0.83$) | $68.61 \pm 0.74$ | $80.14 \pm 0.02$ | $81.89 \pm 0.18$ ($\alpha^{\star} = 2.31$) |

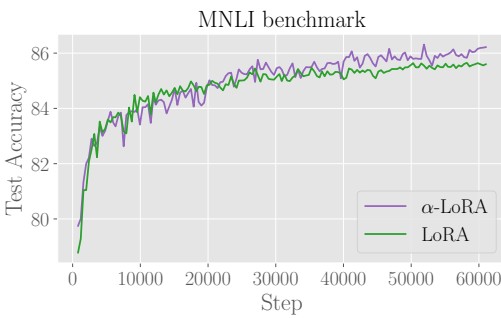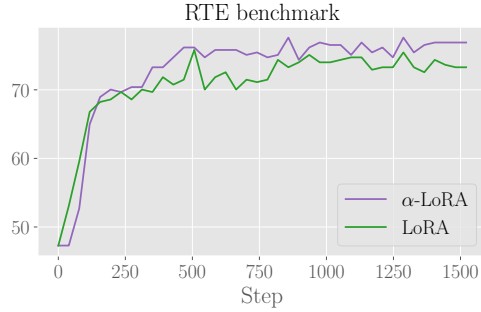

Figure 4: Test accuracy evolution of `roberta-base` finetuned on MNLI and RTE for a single fixed seed (seed 5 for MNLI and seed 123 for RTE).

We note that our approach yields optimal results for all transfer tasks, which clearly validates our theoretical results and underscores the efficiency of our method in terms of its generalization capabilities. This can also be observed in Figure 3, which shows that the optimal test accuracy is obtained for a parameter $\alpha$ that is not necessarily equal, nor even close, to 1.

## 5.2 BEYOND OUR THEORETICAL MODEL: SUPERVISED FINE-TUNING FOR LLMs

To go beyond linear models, we now fine-tune `roberta-base` language model (Liu et al., 2019)) on downstream classification taken from GLUE benchmarks (Wang et al., 2018). To adapt our theoretical insights from the linear model to complex, multi-layered architectures like LLMs, we generalize the scalar scaling parameter $\alpha$ to a vector $\boldsymbol{\alpha}$. This extension provides finer-grained control, allowing the model to rescale the contribution of the frozen base weights on a per-output-neuron basis. This added flexibility is crucial for capturing the intricate functional specialization within different dimensions of a neural network's hidden states. Consequently, the update rule for a weight matrix $\mathbf{W}^\star$ is modified from a simple scalar product to a row-wise scaling operation, as detailed below:

$$\boxed{\mathbf{W}_{\text{new}} = \boldsymbol{\alpha} \odot \mathbf{W}^\star + \mathbf{W}} \tag{10}$$

where $\odot$ is the element-wise product between vectors, $\mathbf{W}^\star \in \mathbb{R}^{d_{out} \times d_{in}}$ is the original layer weights (frozen during training), $\boldsymbol{\alpha} \in \mathbb{R}^{d_{out}}$ (each element in the output dimension is then multiplied by a scalar), and $\mathbf{W} \in \mathbb{R}^{d_{out} \times d_{in}}$ is the trainable weight matrix. Additionally, $\mathbf{W}$ can be approximated with a low-rank matrix: $\mathbf{W} = \mathbf{AB}$, where: $\mathbf{A} \in \mathbb{R}^{d_{out} \times r}$ and $\mathbf{B} \in \mathbb{R}^{r \times d_{in}}$, a method that we call $\alpha$-LoRA. We then report in Table 2 the test performance obtained using standard LoRA and our $\alpha$-LoRA method evaluated on six GLUE tasks: MNLI, QNLI, MRPC, RTE, SST-2, and QQP.

Table 2: Test accuracy comparison over GLUE classification tasks (Wang et al., 2018) using `roberta-base` model. As theoretically anticipated, our new fine-tuning approach yields better test classification accuracy than the standard LoRA method ($\boldsymbol{\alpha} = 1$). The details about these experiments are presented in Appendix E.

| Method | MNLI | QNLI | MRPC | RTE | SST-2 | QQP |
|---|---|---|---|---|---|---|
| **LoRA** | $85.77 \pm 0.16$ | $91.95 \pm 0.03$ | $88.40 \pm 0.31$ | $74.01 \pm 1.64$ | $94.00 \pm 0.11$ | $88.80 \pm 0.02$ |
| $\alpha$-**LoRA** | $\mathbf{86.12 \pm 0.06}$ | $\mathbf{92.20 \pm 0.13}$ | $\mathbf{89.46 \pm 0.53}$ | $\mathbf{77.62 \pm 0.59}$ | $\mathbf{94.38 \pm 0.01}$ | $\mathbf{88.86 \pm 0.03}$ |

We note that from Table 2 and Figure 4, our method leads to higher generalization performance compared to standard LoRA across all GLUE benchmarks, which further validates our theoretical findings of the previous section.

**Finding the parameters $\boldsymbol{\alpha}$.** We designed a practical heuristic algorithm to automatically update $\boldsymbol{\alpha}$ during training. In fact, we consider each vector $\boldsymbol{\alpha}$ as a trainable parameter and update these vectors once each $T$ step (which can be tuned) with either `Adam` or `AdamW` by sampling a new batch, different from the one used to train the reparametrization weights $\mathbf{W}$, and then taking a gradient step over this new batch. The design choices of our algorithm can be justified by the following:

- Because the vectors $\boldsymbol{\alpha}$ applied to each module lie in the whole Euclidean space $\mathbb{R}^d$, it is not possible to find such a parameter through a simple grid search, as this will give a very costly and impractical algorithm.

- Additionally, finding theoretical formulas for each vector $\boldsymbol{\alpha}$ is very hard, if not impossible. Therefore, it is crucial to have an algorithm that updates the vectors $\boldsymbol{\alpha}$ automatically.

- Finally, because we want to optimize the **generalization** performance of our fine-tuning method, training $\boldsymbol{\alpha}$ in the same way as the reparametrization weights $\mathbf{W}$ can easily lead to overfitting of the model, which justifies sampling of new batches to update $\boldsymbol{\alpha}$ and the update rate $T$. Our specific choices are detailed for reproducibility in Appendix E.

Figure 5 shows that our algorithm leads to optimal scaling vectors $\boldsymbol{\alpha}^\star$ in their neighborhood, which proves the effectiveness of our algorithm and the fine-tuning method in general. The pseudo-code 1 of our algorithm is detailed in the Appendix E.

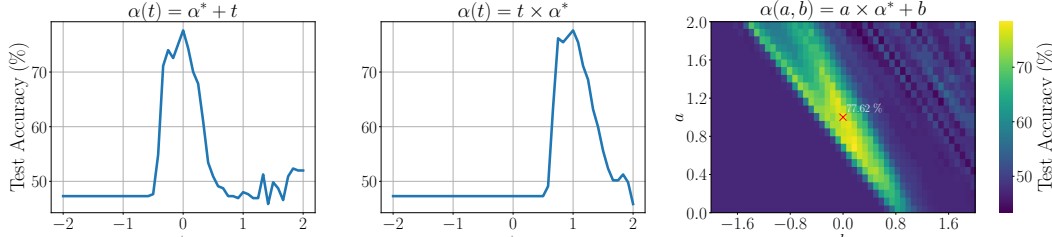

Figure 5: Test accuracy of `roberta-base` finetuned on RTE for different values of $\boldsymbol{\alpha}$ in the neighborhood of the obtained $\boldsymbol{\alpha}^\star$. The values of the parameters $\boldsymbol{\alpha}^\star$ in this experiment range between $0.85$ and $1.14$.

**Overhead induced by the additional parameters $\boldsymbol{\alpha}$.** We note that the number of additional trainable parameters $\boldsymbol{\alpha}$ induced by our algorithm 1 is negligible compared to the standard approach (fixed $\boldsymbol{\alpha} = 1$), for example in the case of our experiments with `roberta-base` model, the increase in the number of trainable parameters is only of $0.02\%$. Additionally, investigating the resulting values of these learned $\boldsymbol{\alpha}$ vectors as reported in Figures 7, 8 and 9 in the Appendix, we notice that we get similar values for query and value matrices, thus we can use a shared parameter for both weight matrices (or for the whole attention module more generally), reducing the overhead even further.

## 6 CONCLUSION AND LIMITATIONS

In this work, we introduced a new class of reparametrization-based fine-tuning methods that leverage an additional scaling parameter to improve the generalization of transfer learning. Using tools from Random Matrix Theory, we proved the existence and impact of an optimal scaling factor in high-dimensional binary classification. We show that this factor is typically different from the standard choice. Our theoretical analysis was further supported by experiments on real-world tasks, where our proposed approach consistently outperformed standard LoRA on multiple benchmarks.

Although promising, our framework also has limitations. Theoretical guarantees are derived under specific assumptions on data distributions and model structure, which may not fully capture the complexity of modern deep architectures. We believe future work could extend these insights to broader model families, design more efficient algorithms for parameter selection, and further explore the trade-off between generalization and efficiency in transfer learning. Furthermore, an exciting avenue for investigation is the integration of our $\alpha$-scaling technique with other advanced adapter methods. Since our approach is complementary to improvements in the adapter's architecture, such as DoRA or other LoRA variants, combining them could lead to synergistic gains in fine-tuning performance.

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

## APPENDIX

This appendix is organized as follows: Section A lists some useful lemmas that will be at the core of our analysis. In Section B, we provide a proof of Theorem 4.2 using Random Matrix Theory. Section C extends the theory to the case of an arbitrary source classifier (not necessarily Ridge) which will be useful also for another extension of the theory to multi-source fine-tuning D. Finally, E lists the details about our experiments on LLM fine-tuning.

# Contents

Throughout the whole Appendix, we will try to analyze the performance of the fine-tuned classifier defined in equation $\alpha$-FTC:

$$\boldsymbol{w}_\alpha = \boldsymbol{w} + \alpha\tilde{\boldsymbol{w}} - \frac{\alpha}{n}\mathbf{Q}(\gamma)\mathbf{X}\mathbf{X}^\top\tilde{\boldsymbol{w}}$$

**Notations.**   Here are two notations that we will use along the whole analysis:

$$\lambda_Q = \|\boldsymbol{\mu}_\beta\|^2 + 1 + \gamma(1 + \delta_Q), \quad \lambda_R = \|\boldsymbol{\mu}\|^2 + 1 + \tilde{\gamma}(1 + \delta_R) \tag{11}$$

## A  USEFUL RESULTS

### A.1  GENERAL LEMMAS

Here we will list useful lemmas used in our analysis.

**Lemma A.1** (Resolvent identity). *For invertible matrices* $\mathbf{A}$ *and* $\mathbf{B}$, *we have:*

$$\mathbf{A}^{-1} - \mathbf{B}^{-1} = \mathbf{A}^{-1}(\mathbf{B} - \mathbf{A})\mathbf{B}^{-1}.$$

**Lemma A.2** (Sherman-Morisson). *For* $\mathbf{A} \in \mathbb{R}^{p\times p}$ *invertible and* $\boldsymbol{u}, \boldsymbol{v} \in \mathbb{R}^p$, $\mathbf{A} + \boldsymbol{u}\boldsymbol{v}^\top$ *is invertible if and only if:* $1 + \boldsymbol{v}^\top\mathbf{A}^{-1}\boldsymbol{u} \neq 0$, *and:*

$$(\mathbf{A} + \boldsymbol{u}\boldsymbol{v}^\top)^{-1} = \mathbf{A}^{-1} - \frac{\mathbf{A}^{-1}\boldsymbol{u}\boldsymbol{v}^\top\mathbf{A}^{-1}}{1 + \boldsymbol{v}^\top\mathbf{A}^{-1}\boldsymbol{u}}.$$

*Besides,*

$$(\mathbf{A} + \boldsymbol{u}\boldsymbol{v}^\top)^{-1}\boldsymbol{u} = \frac{\mathbf{A}^{-1}\boldsymbol{u}}{1 + \boldsymbol{v}^\top\mathbf{A}^{-1}\boldsymbol{u}}.$$

## A.2 DETERMINISTIC EQUIVALENTS

Recall the expression of the resolvents defined in equation equation 9:

$$\mathbf{Q} = \left(\frac{1}{n}\mathbf{X}\mathbf{X}^\top + \gamma \mathbf{I}_p\right)^{-1}, \mathbf{R} = \left(\frac{1}{N}\tilde{\mathbf{X}}\tilde{\mathbf{X}}^\top + \tilde{\gamma}\mathbf{I}_p\right)^{-1}$$

We define the matrices $\mathbf{Q}_{-i}$ and $\mathbf{R}_{-i}$ as the resolvents obtained by removing the contribution of the $i^{th}$ sample, i.e:

$$\mathbf{Q}_{-i} = \left(\frac{1}{n}\sum_{k \neq i}\boldsymbol{x}_k\boldsymbol{x}_k^\top + \gamma\mathbf{I}_p\right)^{-1}, \quad \mathbf{R}_{-i} = \left(\frac{1}{N}\sum_{k \neq i}\tilde{\boldsymbol{x}}_k\tilde{\boldsymbol{x}}_k^\top + \tilde{\gamma}\mathbf{I}_p\right)^{-1}$$

then we have that:

$$\mathbf{Q} = \left(\mathbf{Q}_{-i}^{-1} + \frac{1}{n}\boldsymbol{x}_i\boldsymbol{x}_i^\top\right)^{-1}, \quad \mathbf{R} = \left(\mathbf{R}_{-i}^{-1} + \frac{1}{N}\tilde{\boldsymbol{x}}_i\tilde{\boldsymbol{x}}_i^\top\right)^{-1}$$

Thus by Sherman-Morisson's lemma:

$$\mathbf{Q} = \mathbf{Q}_{-i} - \frac{1}{n}\frac{\mathbf{Q}_{-i}\boldsymbol{x}_i\boldsymbol{x}_i^\top\mathbf{Q}_{-i}}{1 + \delta_Q}, \quad \mathbf{R} = \mathbf{R}_{-i} - \frac{\frac{1}{N}\mathbf{R}_{-i}\tilde{\boldsymbol{x}}_i\tilde{\boldsymbol{x}}_i^\top\mathbf{R}_{-i}}{1 + \delta_R}$$

where:

$$\delta_Q = \frac{1}{n}\operatorname{Tr}\bar{\mathbf{Q}} = \frac{\eta - \gamma - 1 + \sqrt{(\eta - \gamma - 1)^2 + 4\eta\gamma}}{2\gamma}, \quad \delta_R = \frac{1}{N}\operatorname{Tr}\bar{\mathbf{R}} = \frac{\tilde{\eta} - \tilde{\gamma} - 1 + \sqrt{(\tilde{\eta} - \tilde{\gamma} - 1)^2 + 4\tilde{\eta}\tilde{\gamma}}}{2\tilde{\gamma}}$$

Thus, we get that:

$$\mathbf{Q}\boldsymbol{x}_i = \frac{\mathbf{Q}_{-i}\boldsymbol{x}_i}{1 + \delta_Q}, \quad \mathbf{R}\tilde{\boldsymbol{x}}_i = \frac{\mathbf{R}_{-i}\tilde{\boldsymbol{x}}_i}{1 + \delta_R} \tag{12}$$

Using the above identities, we can easily prove the deterministic equivalents of $\mathbf{Q}$ and $\mathbf{R}$ stated in Lemma 3.5, which we will do in the following.

**Lemma A.3** (Deterministic equivalent of $\mathbf{Q}$ and $\mathbf{R}$). *Under the high-dimensional regime and the assumptions 4.1, a deterministic equivalent for $\mathbf{Q} \equiv \mathbf{Q}(\gamma)$ and for $\mathbf{R} \equiv \mathbf{R}(\gamma)$, denoted $\bar{\mathbf{Q}}$ and $\bar{\mathbf{R}}$ respectively, as defined in equation 9 are given by:*

$$\bar{\mathbf{Q}}(\gamma) = \left(\frac{\boldsymbol{\mu}_\beta\boldsymbol{\mu}_\beta^\top + \mathbf{I}_p}{1 + \delta_Q} + \gamma\mathbf{I}_p\right)^{-1}, \quad \bar{\mathbf{R}}(\gamma) = \left(\frac{\boldsymbol{\mu}\boldsymbol{\mu}^\top + \mathbf{I}_p}{1 + \delta_R} + \gamma\mathbf{I}_p\right)^{-1}.$$

*Where:*

$$\delta_Q = \frac{1}{n}\operatorname{Tr}\bar{\mathbf{Q}} = \frac{\eta - \gamma - 1 + \sqrt{(\eta - \gamma - 1)^2 + 4\eta\gamma}}{2\gamma}, \quad \delta_R = \frac{1}{N}\operatorname{Tr}\bar{\mathbf{R}} = \frac{\tilde{\eta} - \tilde{\gamma} - 1 + \sqrt{(\tilde{\eta} - \tilde{\gamma} - 1)^2 + 4\tilde{\eta}\tilde{\gamma}}}{2\tilde{\gamma}}.$$

*Proof.* We will prove the deterministic equivalent of $\mathbf{Q}$, and the proof of $\bar{\mathbf{R}}$ can be derived similarly. In general, we want to find a deterministic equivalent $\bar{\mathbf{Q}}$ of the same form of $\mathbf{Q}$, i.e we consider $\bar{\mathbf{Q}}(\gamma) = (\mathbf{S} + \gamma\mathbf{I}_p)^{-1}$ and we want to find a deterministic matrix $\mathbf{S} \in \mathbb{R}^{p\times p}$ such that for any linear form $u$:

$$u(\mathbf{Q}) \xrightarrow{\text{a.s.}} u(\bar{\mathbf{Q}}),$$

Or more simply:

$$u(\mathbb{E}[\mathbf{Q}] - \bar{\mathbf{Q}}) \to 0.$$

We have that:

$$\mathbb{E}[\mathbf{Q}] - \bar{\mathbf{Q}} = \mathbb{E}[\mathbf{Q} - \bar{\mathbf{Q}}]$$

$$= \mathbb{E}[\mathbf{Q}\left(\mathbf{S} - \frac{1}{n}\mathbf{X}\mathbf{X}^\top\right)\bar{\mathbf{Q}}]$$

$$= \mathbb{E}\left[\left(\mathbf{Q}\mathbf{S} - \frac{1}{n}\sum_{i=1}^n\mathbf{Q}\boldsymbol{x}_i\boldsymbol{x}_i^\top\right)\bar{\mathbf{Q}}\right]$$

And since: $\mathbf{Q}\boldsymbol{x}_i = \frac{\mathbf{Q}_{-i}\boldsymbol{x}_i}{1+\delta_Q}$ and that we want $\mathbb{E}[\mathbf{Q}] = \bar{\mathbf{Q}}$ in linear forms, we get that:

$$\mathbb{E}\left[\left(\mathbf{QS} - \frac{1}{n}\sum_{i=1}^{n} \mathbf{Q}\boldsymbol{x}_i\boldsymbol{x}_i^\top\right)\bar{\mathbf{Q}}\right] = \bar{\mathbf{Q}}\mathbf{S}\bar{\mathbf{Q}} - \frac{1}{n}\sum_{i=1}^{n}\frac{1}{1+\delta_Q}\mathbb{E}[\mathbf{Q}_{-i}\boldsymbol{x}_i\boldsymbol{x}_i^\top]\bar{\mathbf{Q}}$$

$$= \bar{\mathbf{Q}}\mathbf{S}\bar{\mathbf{Q}} - \frac{1}{n}\sum_{i=1}^{n}\frac{1}{1+\delta_Q}\bar{\mathbf{Q}}(\boldsymbol{\mu}_\beta\boldsymbol{\mu}_\beta^\top + \mathbf{I}_p)\bar{\mathbf{Q}} \qquad \text{(By independence of } \boldsymbol{x}_i \text{ and } \mathbf{Q}_{-i})$$

$$= \bar{\mathbf{Q}}\left(\mathbf{S} - \frac{\boldsymbol{\mu}_\beta\boldsymbol{\mu}_\beta^\top + \mathbf{I}_p}{1+\delta_Q}\right)\bar{\mathbf{Q}}$$

Finally, it suffices to take: $\mathbf{S} = \frac{\boldsymbol{\mu}_\beta\boldsymbol{\mu}_\beta^\top + \mathbf{I}_p}{1+\delta_Q}$ to get the desired result. $\qquad\square$

**Lemma A.4** (Trace identities). *Let $\bar{\mathbf{Q}}, \bar{\mathbf{R}} \in \mathbb{R}^{p\times p}$ be the deterministic matrices defined in lemma 3.5. Then:*

$$\frac{1}{n}\frac{\mathrm{Tr}((\Sigma_\beta\bar{\mathbf{Q}})^2)}{(1+\delta_Q)^2} = \frac{\eta}{(1+\gamma(1+\delta_Q))^2}, \quad \frac{1}{N}\frac{\mathrm{Tr}((\Sigma\bar{\mathbf{R}})^2)}{(1+\delta_R)^2} = \frac{\tilde{\eta}}{(1+\tilde{\gamma}(1+\delta_R))^2}.$$

*And:*

$$\frac{1}{N}\mathrm{Tr}(\bar{\mathbf{R}}^2\bar{\mathbf{Q}}^2) = \tilde{\eta}\left(\frac{(1+\delta_R)(1+\delta_Q)}{(1+\tilde{\gamma}(1+\delta_R))(1+\gamma(1+\delta_Q))}\right)^2$$

**Lemma A.5** (Relevant Identities). *Let $\bar{\mathbf{Q}}, \bar{\mathbf{R}} \in \mathbb{R}^{p\times p}$ be the deterministic matrices defined in lemma 3.5. Then we have the following identities:*

$$\boldsymbol{\mu}_\beta^\top\bar{\mathbf{Q}}\boldsymbol{\mu}_\beta = \frac{(1+\delta_Q)\|\boldsymbol{\mu}_\beta\|^2}{\|\boldsymbol{\mu}_\beta\|^2 + 1 + \gamma(1+\delta_Q)}, \quad \boldsymbol{\mu}_\beta^\top\bar{\mathbf{Q}}^2\boldsymbol{\mu}_\beta = \left(\frac{(1+\delta_Q)\|\boldsymbol{\mu}_\beta\|}{\|\boldsymbol{\mu}_\beta\|^2 + 1 + \gamma(1+\delta_Q)}\right)^2,$$

$$\boldsymbol{\mu}^\top\bar{\mathbf{R}}\boldsymbol{\mu} = \frac{(1+\delta_R)\|\boldsymbol{\mu}\|^2}{\|\boldsymbol{\mu}\|^2 + 1 + \tilde{\gamma}(1+\delta_R)}, \quad \boldsymbol{\mu}^\top\bar{\mathbf{R}}^2\boldsymbol{\mu} = \left(\frac{(1+\delta_R)\|\boldsymbol{\mu}\|}{\|\boldsymbol{\mu}\|^2 + 1 + \tilde{\gamma}(1+\delta_R)}\right)^2,$$

$$\boldsymbol{\mu}^\top\bar{\mathbf{R}}\bar{\mathbf{Q}}\boldsymbol{\mu}_\beta = \frac{(1+\delta_R)(1+\delta_Q)\beta\|\boldsymbol{\mu}\|^2}{(\|\boldsymbol{\mu}\|^2 + 1 + \tilde{\gamma}(1+\delta_R))(\|\boldsymbol{\mu}_\beta\|^2 + 1 + \gamma(1+\delta_Q))},$$

$$\boldsymbol{\mu}^\top\bar{\mathbf{R}}\bar{\mathbf{Q}}^2\boldsymbol{\mu}_\beta = \frac{(1+\delta_R)}{(\|\boldsymbol{\mu}\|^2 + 1 + \tilde{\gamma}(1+\delta_R))}\left(\frac{(1+\delta_Q)}{(\|\boldsymbol{\mu}_\beta\|^2 + 1 + \gamma(1+\delta_Q))}\right)^2\beta\|\boldsymbol{\mu}\|^2,$$

*And finally:*

$$\boldsymbol{\mu}^\top\bar{\mathbf{R}}\bar{\mathbf{Q}}^2\bar{\mathbf{R}}\boldsymbol{\mu}$$

$$= \left(\frac{(1+\delta_R)(1+\delta_Q)\|\boldsymbol{\mu}\|}{(1+\gamma(1+\delta_Q))(\|\boldsymbol{\mu}\|^2 + 1 + \tilde{\gamma}(1+\delta_R))}\right)^2\left(1 + \frac{\beta^3\|\boldsymbol{\mu}\|^4}{(\|\boldsymbol{\mu}_\beta\|^2 + 1 + \gamma(1+\delta_Q))^2} - \frac{2\beta^2\|\boldsymbol{\mu}\|^2}{\|\boldsymbol{\mu}_\beta\|^2 + 1 + \gamma(1+\delta_Q)}\right).$$

*Proof.* The proof of all these identities relies on the following results:

$$\bar{\mathbf{R}} = \left(\frac{\boldsymbol{\mu}\boldsymbol{\mu}^\top}{1+\delta_R} + \left(\tilde{\gamma} + \frac{1}{1+\delta_R}\right)\mathbf{I}_p\right)^{-1}$$

$$= (1+\delta_R)\left(\boldsymbol{\mu}\boldsymbol{\mu}^\top + (1 + \tilde{\gamma}(1+\delta_R)\mathbf{I}_p)\right)^{-1}$$

$$= \frac{1+\delta_R}{1+\tilde{\gamma}(1+\delta_R)}\left(\frac{\boldsymbol{\mu}\boldsymbol{\mu}^\top}{1+\tilde{\gamma}(1+\delta_R)} + \mathbf{I}_p\right)^{-1}$$

$$= \frac{1+\delta_R}{1+\tilde{\gamma}(1+\delta_R)}\left(\mathbf{I}_p - \frac{\boldsymbol{\mu}\boldsymbol{\mu}^\top}{\|\boldsymbol{\mu}\|^2 + 1 + \tilde{\gamma}(1+\delta_R)}\right) \qquad \text{(lemma A.2)}$$

where the last equality is obtained using Sherman-Morisson's identity (lemma A.2). Hence,

$$(\bar{\mathbf{R}})^2 = \frac{(1+\delta_R)^2}{(1+\tilde{\gamma}(1+\delta_R))^2}\left(\mathbf{I}_p + \frac{(\boldsymbol{\mu}\boldsymbol{\mu}^\top)^2}{(\|\boldsymbol{\mu}\|^2+1+\tilde{\gamma}(1+\delta_R))^2} - \frac{2\boldsymbol{\mu}\boldsymbol{\mu}^\top}{\|\boldsymbol{\mu}\|^2+1+\tilde{\gamma}(1+\delta_R)}\right).$$

And the same for $\bar{\mathbf{Q}}$:

$$\bar{\mathbf{Q}} = \frac{1+\delta_Q}{1+\gamma(1+\delta_Q)}\left(\mathbf{I}_p - \frac{\boldsymbol{\mu}_\beta\boldsymbol{\mu}_\beta^\top}{\|\boldsymbol{\mu}_\beta\|^2+1+\gamma(1+\delta_Q)}\right),$$

$$(\bar{\mathbf{Q}})^2 = \frac{(1+\delta_Q)^2}{(1+\gamma(1+\delta_Q))^2}\left(\mathbf{I}_p + \frac{(\boldsymbol{\mu}_\beta\boldsymbol{\mu}_\beta^\top)^2}{(\|\boldsymbol{\mu}_\beta\|^2+1+\gamma(1+\delta_Q))^2} - \frac{2\boldsymbol{\mu}_\beta\boldsymbol{\mu}_\beta^\top}{\|\boldsymbol{\mu}_\beta\|^2+1+\gamma(1+\delta_Q)}\right).$$

And using the second identity in Sherman-Morisson's lemma A.2:

$$\bar{\mathbf{R}}\boldsymbol{\mu} = \frac{(1+\delta_R)}{\|\boldsymbol{\mu}\|^2+1+\tilde{\gamma}(1+\delta_R)}\boldsymbol{\mu}, \quad \bar{\mathbf{Q}}\boldsymbol{\mu}_\beta = \frac{(1+\delta_Q)}{\|\boldsymbol{\mu}_\beta\|^2+1+\gamma(1+\delta_Q)}\boldsymbol{\mu}_\beta$$

$\square$

**Lemma A.6** (Expectation some classifiers). *Let $\tilde{\boldsymbol{w}}$ and $\boldsymbol{w}$ be the classifiers defined earlier equation $\alpha$-FTC. We have that:*

$$\mathbb{E}[\tilde{\boldsymbol{w}}] = \frac{1}{1+\delta_R}\bar{\mathbf{R}}\boldsymbol{\mu}, \quad \mathbb{E}[\boldsymbol{w}] = \frac{1}{1+\delta_Q}\bar{\mathbf{Q}}\boldsymbol{\mu}_\beta.$$

*Proof.*

$$\mathbb{E}[\tilde{\boldsymbol{w}}] = \frac{1}{N}\sum_{i=1}^N \mathbb{E}[\tilde{y}_i\mathbf{R}\tilde{\boldsymbol{x}}_i]$$

$$= \frac{1}{N}\sum_{i=1}^N \frac{1}{1+\delta_R}\mathbb{E}[\tilde{y}_i\mathbf{R}_{-i}\tilde{\boldsymbol{x}}_i]$$

$$= \frac{1}{1+\delta_R}\bar{\mathbf{R}}\boldsymbol{\mu}$$

The proof of $\mathbb{E}[\boldsymbol{w}]$ is similar to this latter. $\square$

**Lemma A.7** (Deterministic equivalent). *For any positive semi-definite matrix $\mathbf{A}$, we have:*

$$\mathbf{Q}\mathbf{A}\mathbf{Q} \leftrightarrow \bar{\mathbf{Q}}\mathbf{A}\bar{\mathbf{Q}} + \frac{1}{n}\frac{\mathrm{Tr}(\Sigma_\beta\bar{\mathbf{Q}}\mathbf{A}\bar{\mathbf{Q}})}{(1+\delta_Q)^2}\mathbb{E}[\mathbf{Q}\Sigma_\beta\mathbf{Q}],$$

*and:*

$$\mathbf{R}\mathbf{A}\mathbf{R} \leftrightarrow \bar{\mathbf{R}}\mathbf{A}\bar{\mathbf{R}} + \frac{1}{N}\frac{\mathrm{Tr}(\Sigma\bar{\mathbf{R}}\mathbf{A}\bar{\mathbf{R}})}{(1+\delta_R)^2}\mathbb{E}[\mathbf{R}\Sigma\mathbf{R}].$$

*In particular for every $\boldsymbol{a}, \boldsymbol{b} \in \mathbb{R}^p$:*

$$\boldsymbol{a}^\top\mathbb{E}[\mathbf{Q}\Sigma_\beta\mathbf{Q}]\boldsymbol{b} = \frac{1}{h}\boldsymbol{a}^\top\bar{\mathbf{Q}}\Sigma_\beta\bar{\mathbf{Q}}\boldsymbol{b}, \quad \boldsymbol{a}^\top\mathbb{E}[\mathbf{R}\Sigma\mathbf{R}]\boldsymbol{b} = \frac{1}{\tilde{h}}\boldsymbol{a}^\top\bar{\mathbf{R}}\Sigma\bar{\mathbf{R}}\boldsymbol{b}.$$

*Proof.* The proof is derived similarly as in the appendix of Firdoussi & Seddik (2024). Again, the proof is similar for both $\mathbf{Q}$ and $\mathbf{R}$.
Let $\bar{\mathbf{Q}}$ be a deterministic equivalent of $\mathbf{Q}$. The following equations and identities are valid in terms of linear forms. We have that:

$$\mathbb{E}[\mathbf{Q}\mathbf{A}\mathbf{Q}] = \mathbb{E}[\bar{\mathbf{Q}}\mathbf{A}\mathbf{Q}] + \mathbb{E}[(\mathbf{Q}-\bar{\mathbf{Q}})\mathbf{A}\mathbf{Q}]$$

$$= \bar{\mathbf{Q}}(\mathbb{E}[\mathbf{A}\mathbf{Q}] + \mathbf{A}\mathbb{E}[\mathbf{Q}-\bar{\mathbf{Q}}]) + \mathbb{E}[(\mathbf{Q}-\bar{\mathbf{Q}})\mathbf{A}\mathbf{Q}]$$

$$= \bar{\mathbf{Q}}\mathbf{A}\bar{\mathbf{Q}} + \mathbb{E}[(\mathbf{Q}-\bar{\mathbf{Q}})\mathbf{A}\mathbf{Q}]$$

Using lemma A.1, we have that:

$$\mathbf{Q} - \bar{\mathbf{Q}} = \mathbf{Q}(\bar{\mathbf{Q}}^{-1} - \mathbf{Q}^{-1})\bar{\mathbf{Q}}$$

$$= \mathbf{Q}\left(\frac{\Sigma_\beta}{1+\delta_Q} - \frac{1}{n}\mathbf{X}\mathbf{X}^\top\right)\bar{\mathbf{Q}}$$

$$= \mathbf{Q}\left(\mathbf{S} - \frac{1}{n}\mathbf{X}\mathbf{X}^\top\right)\bar{\mathbf{Q}}$$

Thus:

$$\mathbb{E}[\mathbf{Q}\mathbf{A}\mathbf{Q}] = \bar{\mathbf{Q}}\mathbf{A}\bar{\mathbf{Q}} + \mathbb{E}[\mathbf{Q}(\mathbf{S} - \frac{1}{n}\mathbf{X}\mathbf{X}^\top)\bar{\mathbf{Q}}\mathbf{A}\mathbf{Q}]$$

$$= \bar{\mathbf{Q}}\mathbf{A}\bar{\mathbf{Q}} + \mathbb{E}[\mathbf{Q}\mathbf{S}\bar{\mathbf{Q}}\mathbf{A}\mathbf{Q}] - \frac{1}{n}\sum_{i=1}^n \mathbb{E}[\mathbf{Q}\boldsymbol{x}_i\boldsymbol{x}_i^\top\bar{\mathbf{Q}}\mathbf{A}\mathbf{Q}]$$

We have that:

$$\mathbb{E}[\mathbf{Q}\boldsymbol{x}_i\boldsymbol{x}_i^\top\bar{\mathbf{Q}}\mathbf{A}\mathbf{Q}] = \frac{1}{1+\delta_Q}\,\mathbb{E}[\mathbf{Q}_{-i}\boldsymbol{x}_i\boldsymbol{x}_i^\top\bar{\mathbf{Q}}\mathbf{A}\mathbf{Q}]$$

$$= \frac{1}{1+\delta_Q}\left(\mathbb{E}[\mathbf{Q}_{-i}\boldsymbol{x}_i\boldsymbol{x}_i^\top\bar{\mathbf{Q}}\mathbf{Q}_{-i}] - \mathbb{E}[\mathbf{Q}_{-i}\boldsymbol{x}_i\boldsymbol{x}_i^\top\bar{\mathbf{Q}}\mathbf{A}\frac{\mathbf{Q}_{-i}\boldsymbol{x}_i\boldsymbol{x}_i^\top\mathbf{Q}_{-i}}{n(1+\delta_Q)}]\right)$$

$$= \frac{1}{1+\delta_Q}\left(\mathbb{E}[\mathbf{Q}_{-i}\Sigma_\beta\bar{\mathbf{Q}}\mathbf{A}\mathbf{Q}_{-i}] - \mathbb{E}[\mathbf{Q}_{-i}\boldsymbol{x}_i\boldsymbol{x}_i^\top\bar{\mathbf{Q}}\mathbf{A}\frac{\mathbf{Q}_{-i}\boldsymbol{x}_i\boldsymbol{x}_i^\top\mathbf{Q}_{-i}}{n(1+\delta_Q)}]\right)$$

$$= \frac{1}{1+\delta_Q}\left(\mathbb{E}[\mathbf{Q}\Sigma_\beta\bar{\mathbf{Q}}\mathbf{A}\mathbf{Q}] - \mathbb{E}[\mathbf{Q}_{-i}\boldsymbol{x}_i\boldsymbol{x}_i^\top\bar{\mathbf{Q}}\mathbf{A}\frac{\mathbf{Q}_{-i}\boldsymbol{x}_i\boldsymbol{x}_i^\top\mathbf{Q}_{-i}}{n(1+\delta_Q)}]\right)$$

Therefore, by replacing the obtained expression of $\mathbb{E}[\mathbf{Q}\boldsymbol{x}_i\boldsymbol{x}_i^\top\bar{\mathbf{Q}}\mathbf{A}\mathbf{Q}]$ in the equation of $\mathbb{E}[\mathbf{Q}\mathbf{A}\mathbf{Q}]$, we get that:

$$\mathbb{E}[\mathbf{Q}\mathbf{A}\mathbf{Q}] = \bar{\mathbf{Q}}\mathbf{A}\bar{\mathbf{Q}} + \frac{1}{n^2(1+\delta_Q)^2}\sum_{i=1}^n \mathbb{E}[\mathbf{Q}_{-i}\boldsymbol{x}_i\boldsymbol{x}_i^\top\bar{\mathbf{Q}}\mathbf{A}\mathbf{Q}_{-i}\boldsymbol{x}_i\boldsymbol{x}_i^\top\mathbf{Q}_{-i}]$$

$$= \bar{\mathbf{Q}}\mathbf{A}\bar{\mathbf{Q}} + \frac{1}{n^2(1+\delta_Q)^2}\sum_{i=1}^n \text{Tr}(\Sigma_\beta\bar{\mathbf{Q}}\mathbf{A}\bar{\mathbf{Q}})\,\mathbb{E}[\mathbf{Q}_{-i}\boldsymbol{x}_i\boldsymbol{x}_i^\top\mathbf{Q}_{-i}]$$

$$= \bar{\mathbf{Q}}\mathbf{A}\bar{\mathbf{Q}} + \frac{1}{n^2(1+\delta_Q)^2}\sum_{i=1}^n \text{Tr}(\Sigma_\beta\bar{\mathbf{Q}}\mathbf{A}\bar{\mathbf{Q}})\,\mathbb{E}[\mathbf{Q}_{-i}\Sigma_\beta\mathbf{Q}_{-i}]$$

$$= \bar{\mathbf{Q}}\mathbf{A}\bar{\mathbf{Q}} + \frac{1}{n}\frac{\text{Tr}(\Sigma_\beta\bar{\mathbf{Q}}\mathbf{A}\bar{\mathbf{Q}})}{(1+\delta_Q)^2}\,\mathbb{E}[\mathbf{Q}\Sigma_\beta\mathbf{Q}]$$

Which finally concludes the proof. $\square$

Now we will provide the result of a useful quantity that we will be using for computing the variance.

**Lemma A.8** (Expectation of $\tilde{\boldsymbol{w}}^\top\mathbf{A}\tilde{\boldsymbol{w}}$). *Let $\mathbf{A} \in \mathbb{R}^{p\times p}$ be a random matrix independent of $\tilde{\boldsymbol{w}}$. We have that:*

$$\mathbb{E}[\tilde{\boldsymbol{w}}^\top\mathbf{A}\tilde{\boldsymbol{w}}] = \frac{1}{(1+\delta_R)^2}\left(\boldsymbol{\mu}^\top\,\mathbb{E}[\mathbf{R}\mathbf{A}\mathbf{R}]\boldsymbol{\mu} - \frac{2}{N(1+\delta_R)}\,\text{Tr}(\Sigma\,\mathbb{E}[\mathbf{R}\mathbf{A}\mathbf{R}])\boldsymbol{\mu}^\top\bar{\mathbf{R}}\boldsymbol{\mu} + \frac{1}{N}\,\text{Tr}(\Sigma\,\mathbb{E}[\mathbf{R}\mathbf{A}\mathbf{R}])\right)$$

*Proof.* We have that:

$$\mathbb{E}[\tilde{\boldsymbol{w}}^\top\mathbf{A}\tilde{\boldsymbol{w}}] = \frac{1}{N^2}\sum_{i,j=1}^N \mathbb{E}[\tilde{y}_i\tilde{y}_j\tilde{\boldsymbol{x}}_i^\top\mathbf{R}\mathbf{A}\mathbf{R}\tilde{\boldsymbol{x}}_j]$$

$$= \frac{1}{N^2}\sum_{i\neq j}\mathbb{E}[\tilde{y}_i\tilde{y}_j\tilde{\boldsymbol{x}}_i^\top\mathbf{R}\mathbf{A}\mathbf{R}\tilde{\boldsymbol{x}}_j] + \frac{1}{N^2}\sum_{i=1}^N \mathbb{E}[\tilde{\boldsymbol{x}}_i^\top\mathbf{R}\mathbf{A}\mathbf{R}\tilde{\boldsymbol{x}}_i]$$

We have for $i \neq j$:

$$\mathbb{E}[\tilde{y}_i \tilde{y}_j \tilde{\boldsymbol{x}}_i^\top \mathbf{R} \mathbf{A} \mathbf{R} \tilde{\boldsymbol{x}}_j] = \frac{1}{(1 + \delta_R)^2} \mathbb{E}[\tilde{y}_i \tilde{y}_j \tilde{\boldsymbol{x}}_i \mathbf{R}_{-i} \mathbf{A} \mathbf{R}_{-i} \tilde{\boldsymbol{x}}_j]$$

$$= \frac{1}{(1 + \delta_R)^2} \mathbb{E}\left[\tilde{y}_i \tilde{y}_j \tilde{\boldsymbol{x}}_i^\top \left(\mathbf{R}_{-ij} - \frac{\frac{1}{N}\mathbf{R}_{-ij} \tilde{\boldsymbol{x}}_j \tilde{\boldsymbol{x}}_j^\top \mathbf{R}_{-ij}}{1 + \delta_R}\right) \mathbf{A} \left(\mathbf{R}_{-ij} - \frac{\frac{1}{N}\mathbf{R}_{-ij} \tilde{\boldsymbol{x}}_i \tilde{\boldsymbol{x}}_i^\top \mathbf{R}_{-ij}}{1 + \delta_R}\right) \tilde{\boldsymbol{x}}_j\right]$$

$$= A_{11} - A_{12} - A_{13} + A_{14}$$

So let us compute each term independently:

$$A_{11} = \frac{1}{(1 + \delta_R)^2} \mathbb{E}[\tilde{y}_i \tilde{y}_j \tilde{\boldsymbol{x}}_i^\top \mathbf{R}_{-ij} \mathbf{A} \mathbf{R}_{-ij} \tilde{\boldsymbol{x}}_j]$$

$$= \frac{1}{(1 + \delta_R)^2} \boldsymbol{\mu}^\top \mathbb{E}[\mathbf{R} \mathbf{A} \mathbf{R}] \boldsymbol{\mu}$$

And :

$$A_{12} = \frac{1}{N(1 + \delta_R)^3} \mathbb{E}[\tilde{y}_i \tilde{y}_j \tilde{\boldsymbol{x}}_i^\top \mathbf{R}_{-ij} \mathbf{A} \mathbf{R}_{-ij} \tilde{\boldsymbol{x}}_i \tilde{\boldsymbol{x}}_i^\top \mathbf{R}_{-ij} \tilde{\boldsymbol{x}}_j]$$

$$= \frac{1}{N(1 + \delta_R)^3} \text{Tr}(\Sigma \mathbb{E}[\mathbf{R} \mathbf{A} \mathbf{R}]) \mathbb{E}[\tilde{y}_i \tilde{y}_j \tilde{\boldsymbol{x}}_i^\top \mathbf{R}_{-ij} \tilde{\boldsymbol{x}}_j]$$

$$= \frac{1}{N(1 + \delta_R)^3} \text{Tr}(\Sigma \mathbb{E}[\mathbf{R} \mathbf{A} \mathbf{R}]) \boldsymbol{\mu}^\top \bar{\mathbf{R}} \boldsymbol{\mu}$$

And also we can easily observe that:

$$A_{13} = A_{12}, \quad A_{14} = \mathcal{O}(N^{-1}).$$

Thus:

$$\mathbb{E}[\tilde{y}_i \tilde{y}_j \tilde{\boldsymbol{x}}_i^\top \mathbf{R} \mathbf{A} \mathbf{R} \tilde{\boldsymbol{x}}_j] = \frac{1}{(1 + \delta_R)^2} \left(\boldsymbol{\mu}^\top \mathbb{E}[\mathbf{R} \mathbf{A} \mathbf{R}] \boldsymbol{\mu} - \frac{2}{N(1 + \delta_R)} \text{Tr}(\Sigma \mathbb{E}[\mathbf{R} \mathbf{A} \mathbf{R}]) \boldsymbol{\mu}^\top \bar{\mathbf{R}} \boldsymbol{\mu}\right)$$

And for the second term in the equation of $\mathbb{E}[\tilde{\boldsymbol{w}} \mathbf{A} \tilde{\boldsymbol{w}}]$, we have:

$$\mathbb{E}[\tilde{\boldsymbol{x}}_i^\top \mathbf{R} \mathbf{A} \mathbf{R} \tilde{\boldsymbol{x}}_i] = \frac{1}{(1 + \delta_R)^2} \mathbb{E}[\tilde{\boldsymbol{x}}_i^\top \mathbf{R}_{-i} \mathbf{A} \mathbf{R}_{-i} \tilde{\boldsymbol{x}}_i]$$

$$= \frac{1}{(1 + \delta_R)^2} \mathbb{E}[\text{Tr}(\tilde{\boldsymbol{x}}_i \tilde{\boldsymbol{x}}_i^\top \mathbf{R}_{-i} \mathbf{A} \mathbf{R}_{-i})]$$

$$= \frac{1}{(1 + \delta_R)^2} \text{Tr}(\mathbb{E}[\tilde{\boldsymbol{x}}_i \tilde{\boldsymbol{x}}_i^\top] \mathbb{E}[\mathbf{R}_{-i} \mathbf{A} \mathbf{R}_{-i}])$$

$$= \frac{1}{(1 + \delta_R)^2} \text{Tr}(\Sigma \mathbb{E}[\mathbf{R} \mathbf{A} \mathbf{R}])$$

Hence, finally:

$$\mathbb{E}[\tilde{\boldsymbol{w}}^\top \mathbf{A} \tilde{\boldsymbol{w}}] = \frac{1}{(1 + \delta_R)^2} \left(\boldsymbol{\mu}^\top \mathbb{E}[\mathbf{R} \mathbf{A} \mathbf{R}] \boldsymbol{\mu} - \frac{2}{N(1 + \delta_R)} \text{Tr}(\Sigma \mathbb{E}[\mathbf{R} \mathbf{A} \mathbf{R}]) \boldsymbol{\mu}^\top \bar{\mathbf{R}} \boldsymbol{\mu} + \frac{1}{N} \text{Tr}(\Sigma \mathbb{E}[\mathbf{R} \mathbf{A} \mathbf{R}])\right)$$

$\square$

**Lemma A.9** (Commutativity). *Let $\bar{\mathbf{R}}$ and $\bar{\mathbf{Q}}$ be the resolvent matrices defined in lemma 3.5. We have that:*

$$\bar{\mathbf{Q}} \Sigma_\beta = \Sigma_\beta \bar{\mathbf{Q}}, \quad \bar{\mathbf{R}} \Sigma = \Sigma \bar{\mathbf{R}}.$$

*Proof.* We will just prove it for $\bar{\mathbf{Q}}$ and $\Sigma_\beta$ because the other proof of the second identity is similar. We know that:

$$\Sigma_\beta = (1 + \delta_Q)(\bar{\mathbf{Q}}^{-1} - \gamma \mathbf{I}_p)$$

Thus:

$$\bar{\mathbf{Q}} \Sigma_\beta = (1 + \delta_Q) \bar{\mathbf{Q}}(\bar{\mathbf{Q}}^{-1} - \gamma \mathbf{I}_p) = (1 + \delta_Q)(\mathbf{I}_p - \gamma \bar{\mathbf{Q}})$$

$$\Sigma_\beta \bar{\mathbf{Q}} = (1 + \delta_Q)(\bar{\mathbf{Q}}^{-1} - \gamma \mathbf{I}_p) \bar{\mathbf{Q}} = (1 + \delta_Q)(\mathbf{I}_p - \gamma \bar{\mathbf{Q}})$$

which concludes the proof. $\square$

# B    RMT ANALYSIS OF THE FINE-TUNED CLASSIFIER

Let $x \sim \mathcal{N}((-1)^a \boldsymbol{\mu}_\beta, \mathbf{I}_p)$ independent of the fine-tuning dataset $\mathbf{X}$. We recall that:

$$w_\alpha = w + \alpha \tilde{w} - \frac{\alpha}{n} \mathbf{Q}(\gamma) \mathbf{X} \mathbf{X}^\top \tilde{w},$$

where:

$$w = \frac{1}{n} \mathbf{Q}(\gamma) \mathbf{X} y, \quad \tilde{w} = \frac{1}{N} \mathbf{R}(\tilde{\gamma}) \tilde{\mathbf{X}} \tilde{y}$$

## B.1    TEST EXPECTATION

We have that:

$$\mathbb{E}[w_\alpha^\top x] = \mathbb{E}[w^\top x] + \alpha \mathbb{E}[\tilde{w}^\top x] - \frac{\alpha}{n} \mathbb{E}[\tilde{w}^\top \mathbf{X} \mathbf{X}^\top \mathbf{Q} x] \tag{13}$$

Let us compute each term of this previous sum.
First, using lemma A.6, we have that, since $x$ is independent of $\mathbf{X}$ and of $\tilde{\mathbf{X}}$:

$$\mathbb{E}[w^\top x] = \mathbb{E}[w]^\top \mathbb{E}[x] = \frac{(-1)^a}{1+\delta_Q} \boldsymbol{\mu}_\beta^\top \bar{\mathbf{Q}} \boldsymbol{\mu}_\beta$$

$$\mathbb{E}[\tilde{w}^\top x] = \mathbb{E}[\tilde{w}]^\top \mathbb{E}[x] = \frac{(-1)^a}{1+\delta_R} \boldsymbol{\mu}^\top \bar{\mathbf{R}} \boldsymbol{\mu}_\beta$$

And we have that:

$$\mathbb{E}[\tilde{w}^\top \mathbf{X} \mathbf{X}^\top \mathbf{Q} x] = \mathbb{E}[\tilde{w}]^\top \mathbb{E}[\mathbf{X} \mathbf{X}^\top \mathbf{Q}] \mathbb{E}[x]$$

And:

$$\mathbb{E}[\mathbf{X} \mathbf{X}^\top \mathbf{Q}] = \sum_{i=1}^n \mathbb{E}[x_i x_i^\top \mathbf{Q}]$$

$$= \sum_{i=1}^n \frac{1}{1+\delta_Q} \mathbb{E}[x_i x_i^\top \mathbf{Q}_i]$$

$$= \sum_{i=1}^n \frac{1}{1+\delta_Q} \mathbb{E}[x_i x_i^\top] \bar{\mathbf{Q}}$$

$$= \frac{n}{1+\delta_Q} \Sigma_\beta \bar{\mathbf{Q}}$$

Thus:

$$\frac{1}{n} \mathbb{E}[\tilde{w}^\top \mathbf{X} \mathbf{X}^\top \mathbf{Q} x] = \frac{(-1)^a}{(1+\delta_R)} \frac{1}{(1+\delta_Q)} \boldsymbol{\mu}^\top \bar{\mathbf{R}} \Sigma_\beta \bar{\mathbf{Q}} \boldsymbol{\mu}_\beta$$

$$= \frac{(-1)^a}{1+\delta_R} \boldsymbol{\mu}^\top \bar{\mathbf{R}} (\mathbf{I}_p - \gamma \bar{\mathbf{Q}}) \boldsymbol{\mu}_\beta$$

Finally:

$$\mathbb{E}[w_\alpha^\top x] = (-1)^a \left( \frac{1}{1+\delta_Q} \boldsymbol{\mu}_\beta^\top \bar{\mathbf{Q}} \boldsymbol{\mu}_\beta + \frac{\alpha}{1+\delta_R} \boldsymbol{\mu}^\top \bar{\mathbf{R}} \boldsymbol{\mu}_\beta - \frac{\alpha}{1+\delta_R} \boldsymbol{\mu}^\top \bar{\mathbf{R}} (\mathbf{I}_p - \gamma \bar{\mathbf{Q}}) \boldsymbol{\mu}_\beta \right)$$

$$= (-1)^a \left( \frac{1}{1+\delta_Q} \boldsymbol{\mu}_\beta^\top \bar{\mathbf{Q}} \boldsymbol{\mu}_\beta + \frac{\alpha\gamma}{1+\delta_R} \boldsymbol{\mu}^\top \bar{\mathbf{R}} \bar{\mathbf{Q}} \boldsymbol{\mu}_\beta \right)$$

And using the identities in lemma A.5:

$$\mathbb{E}[w_\alpha^\top x] = \frac{(-1)^a}{(\|\boldsymbol{\mu}_\beta\|^2 + 1 + \gamma(1+\delta_Q))} \left( \|\boldsymbol{\mu}_\beta\|^2 + \frac{\alpha\gamma(1+\delta_Q)}{(\|\boldsymbol{\mu}\|^2 + 1 + \tilde{\gamma}(1+\delta_R))} \beta \|\boldsymbol{\mu}\|^2 \right) \tag{14}$$

$$= \frac{(-1)^a}{\lambda_Q} \left( \|\boldsymbol{\mu}_\beta\|^2 + \frac{\alpha\beta\gamma(1+\delta_Q)}{\lambda_R} \|\boldsymbol{\mu}\|^2 \right) \tag{15}$$

### B.2 TEST VARIANCE

To compute the variance of $\boldsymbol{w}_\alpha^\top \boldsymbol{x}$, it suffices to compute the second moment: $\mathbb{E}[(\boldsymbol{w}_\alpha^\top \boldsymbol{x})^2]$.

$$\mathbb{E}[(\boldsymbol{w}_\alpha^\top \boldsymbol{x})^2] = \mathbb{E}[(\boldsymbol{w}^\top \boldsymbol{x} + \alpha\tilde{\boldsymbol{w}}^\top \boldsymbol{x})^2 + \frac{\alpha^2}{n^2}(\tilde{\boldsymbol{w}}^\top \mathbf{X}\mathbf{X}^\top \mathbf{Q}\boldsymbol{x})^2 - \frac{2\alpha}{n}\tilde{\boldsymbol{w}}^\top \mathbf{X}\mathbf{X}^\top \mathbf{Q}\boldsymbol{x}(\boldsymbol{w}^\top \boldsymbol{x} + \alpha\tilde{\boldsymbol{w}}^\top \boldsymbol{x})]$$
(16)

**First term:** We have that, as proved in Firdoussi & Seddik (2024):

$$\mathbb{E}[(\boldsymbol{w}^\top \boldsymbol{x})^2] = \frac{1}{h(1+\delta_Q)}\left(\frac{1}{1+\delta_Q}\boldsymbol{\mu}_\beta^\top \bar{\mathbf{Q}}\Sigma_\beta \bar{\mathbf{Q}}\boldsymbol{\mu}_\beta - 2(1-h)\boldsymbol{\mu}_\beta^\top \bar{\mathbf{Q}}\boldsymbol{\mu}_\beta\right) + \frac{1-h}{h}$$

$$= \frac{1}{h(1+\delta_Q)}\left(\frac{1}{1+\delta_Q}\left((\boldsymbol{\mu}_\beta^\top \bar{\mathbf{Q}}\boldsymbol{\mu}_\beta)^2 + \boldsymbol{\mu}_\beta^\top \bar{\mathbf{Q}}^2\boldsymbol{\mu}_\beta\right) - 2(1-h)\boldsymbol{\mu}_\beta^\top \bar{\mathbf{Q}}\boldsymbol{\mu}_\beta\right) + \frac{1-h}{h}$$

$$= \frac{\|\boldsymbol{\mu}_\beta\|^2}{h(\|\boldsymbol{\mu}_\beta\|^2 + 1 + \gamma(1+\delta_Q))}\left(\frac{\|\boldsymbol{\mu}_\beta\|^2 + 1}{\|\boldsymbol{\mu}_\beta\|^2 + 1 + \gamma(1+\delta_Q)} - 2(1-h)\right) + \frac{1-h}{h}$$

And:

$$\mathbb{E}[(\tilde{\boldsymbol{w}}^\top x)^2] = \mathbb{E}[\tilde{\boldsymbol{w}}^\top \boldsymbol{x}\tilde{\boldsymbol{w}}^\top \boldsymbol{x}]$$
$$= \mathbb{E}[\tilde{\boldsymbol{w}}^\top \boldsymbol{x}\boldsymbol{x}^\top \tilde{\boldsymbol{w}}]$$
$$= \mathbb{E}[\tilde{\boldsymbol{w}}^\top \Sigma_\beta \tilde{\boldsymbol{w}}]$$

Therefore by lemma A.8:

$$\mathbb{E}[(\tilde{\boldsymbol{w}}^\top x)^2] = \frac{1}{(1+\delta_R)^2}\left(\boldsymbol{\mu}^\top \mathbb{E}[\mathbf{R}\Sigma_\beta \mathbf{R}]\boldsymbol{\mu} - \frac{2}{(1+\delta_R)}\frac{1}{N}\text{Tr}(\Sigma\mathbb{E}[\mathbf{R}\Sigma_\beta \mathbf{R}])\boldsymbol{\mu}^\top \bar{\mathbf{R}}\boldsymbol{\mu} + \frac{1}{N}\text{Tr}(\Sigma\mathbb{E}[\mathbf{R}\Sigma_\beta \mathbf{R}])\right)$$
(17)

And, we have that:

$$\mathbb{E}[\boldsymbol{w}^\top \boldsymbol{x}\tilde{\boldsymbol{w}}^\top \boldsymbol{x}] = \mathbb{E}[\boldsymbol{w}^\top \boldsymbol{x}\boldsymbol{x}^\top \tilde{\boldsymbol{w}}]$$
$$= \mathbb{E}[\boldsymbol{w}]^\top \Sigma_\beta \mathbb{E}[\tilde{\boldsymbol{w}}]$$
$$= \frac{1}{(1+\delta_Q)(1+\delta_R)}\boldsymbol{\mu}_\beta^\top \bar{\mathbf{Q}}\Sigma_\beta \bar{\mathbf{R}}\boldsymbol{\mu}$$
$$= \frac{1}{(1+\delta_R)}\boldsymbol{\mu}_\beta^\top (\mathbf{I}_p - \gamma\bar{\mathbf{Q}})\bar{\mathbf{R}}\boldsymbol{\mu}$$
$$= \frac{1}{(1+\delta_R)}\boldsymbol{\mu}_\beta^\top \bar{\mathbf{R}}\boldsymbol{\mu} - \frac{\gamma}{(1+\delta_R)}\boldsymbol{\mu}_\beta^\top \bar{\mathbf{Q}}\bar{\mathbf{R}}\boldsymbol{\mu}$$

And since $\mathbb{E}[\boldsymbol{w}^\top \boldsymbol{x}\tilde{\boldsymbol{w}}^\top \boldsymbol{x}] = \mathbb{E}[\tilde{\boldsymbol{w}}^\top \boldsymbol{x}\boldsymbol{w}^\top \boldsymbol{x}]$, then:

$$\mathbb{E}[\boldsymbol{w}^\top \boldsymbol{x}\tilde{\boldsymbol{w}}^\top \boldsymbol{x}] = \frac{1}{(1+\delta_R)}\boldsymbol{\mu}_\beta^\top \bar{\mathbf{R}}\boldsymbol{\mu} - \frac{\gamma}{(1+\delta_R)}\boldsymbol{\mu}_\beta^\top \bar{\mathbf{R}}\bar{\mathbf{Q}}\boldsymbol{\mu}$$

and thus:

$$\boldsymbol{\mu}_\beta^\top \bar{\mathbf{R}}\bar{\mathbf{Q}}\boldsymbol{\mu} = \boldsymbol{\mu}_\beta^\top \bar{\mathbf{Q}}\bar{\mathbf{R}}\boldsymbol{\mu}$$
(18)

**Second term:** Now let us compute the expectation of the second term in equation 28:

$$\frac{1}{n^2}\mathbb{E}[(\tilde{\boldsymbol{w}}^\top \mathbf{X}\mathbf{X}^\top \mathbf{Q}\boldsymbol{x})^2] = \frac{1}{n^2}\mathbb{E}[\tilde{\boldsymbol{w}}^\top \mathbf{X}\mathbf{X}^\top \mathbf{Q}\boldsymbol{x}\tilde{\boldsymbol{w}}^\top \mathbf{X}\mathbf{X}^\top \mathbf{Q}\boldsymbol{x}]$$
$$= \frac{1}{n^2}\mathbb{E}[\tilde{\boldsymbol{w}}^\top \mathbf{X}\mathbf{X}^\top \mathbf{Q}\boldsymbol{x}\boldsymbol{x}^\top \mathbf{X}\mathbf{X}^\top \mathbf{Q}\tilde{\boldsymbol{w}}]$$
$$= \frac{1}{n^2}\mathbb{E}[\tilde{\boldsymbol{w}}^\top \mathbf{X}\mathbf{X}^\top \mathbf{Q}\Sigma_\beta \mathbf{X}\mathbf{X}^\top \mathbf{Q}\tilde{\boldsymbol{w}}]$$
$$= \mathbb{E}[\tilde{\boldsymbol{w}}^\top (\mathbf{I}_p - \gamma\mathbf{Q})\Sigma_\beta(\mathbf{I}_p - \gamma\mathbf{Q})\tilde{\boldsymbol{w}}]$$

Therefore, by lemma A.8:

$$\frac{1}{n^2}\mathbb{E}[(\tilde{\boldsymbol{w}}^\top \mathbf{X}\mathbf{X}^\top \mathbf{Q}\boldsymbol{x})^2] = \frac{1}{(1+\delta_R)^2}\boldsymbol{\mu}^\top \mathbb{E}[\mathbf{R}(\mathbf{I}_p - \gamma\mathbf{Q})\Sigma_\beta(\mathbf{I}_p - \gamma\mathbf{Q})\mathbf{R}]\boldsymbol{\mu}$$

$$+ \frac{\text{Tr}(\Sigma\mathbb{E}[\mathbf{R}(\mathbf{I}_p - \gamma\mathbf{Q})\Sigma_\beta(\mathbf{I}_p - \gamma\mathbf{Q})\mathbf{R}])}{N(1+\delta_R)^2}\left(1 - \frac{2}{(1+\delta_R)}\boldsymbol{\mu}^\top \bar{\mathbf{R}}\boldsymbol{\mu}\right)$$

**Third term:** Now we want to compute $\frac{2\alpha}{n} \mathbb{E}[\tilde{w}^\top \mathbf{X}\mathbf{X}^\top \mathbf{Q}x(w^\top x + \alpha\tilde{w}^\top x)]$. So we have that:

$$\mathbb{E}[\tilde{w}^\top \mathbf{X}\mathbf{X}^\top \mathbf{Q}xw^\top x] = \mathbb{E}[\tilde{w}]^\top \mathbb{E}[\mathbf{X}\mathbf{X}^\top \mathbf{Q}xx^\top w]$$

$$= \mathbb{E}[\tilde{w}]^\top \mathbb{E}[\mathbf{X}\mathbf{X}^\top \mathbf{Q}\Sigma_\beta w]$$

$$= \mathbb{E}[\tilde{w}]^\top \mathbb{E}[\frac{1}{n}\mathbf{X}\mathbf{X}^\top \mathbf{Q}\Sigma_\beta \mathbf{Q}\mathbf{X}y]$$

$$= \mathbb{E}[\tilde{w}]^\top \mathbb{E}[(\mathbf{Q}^{-1} - \gamma\mathbf{I}_p)\mathbf{Q}\Sigma_\beta \mathbf{Q}\mathbf{X}y]$$

$$= \mathbb{E}[\tilde{w}]^\top \mathbb{E}[(\mathbf{I}_p - \gamma\mathbf{Q})\Sigma_\beta \mathbf{Q}\mathbf{X}y]$$

$$= \mathbb{E}[\tilde{w}]^\top (\mathbb{E}[\Sigma_\beta \mathbf{Q}\mathbf{X}y] - \gamma \mathbb{E}[\mathbf{Q}\Sigma_\beta \mathbf{Q}\mathbf{X}y])$$

And we have that:

$$\mathbb{E}[\Sigma_\beta \mathbf{Q}\mathbf{X}y] = \sum_{i=1}^n \mathbb{E}[y_i \Sigma_\beta \mathbf{Q}x_i]$$

$$= \frac{n}{(1+\delta_Q)} \mathbb{E}[y_i \Sigma_\beta \mathbf{Q}_{-i}x_i]$$

$$= \frac{n}{(1+\delta_Q)} \Sigma_\beta \bar{\mathbf{Q}}\mu_\beta$$

$$= n(\mathbf{I}_p - \gamma\bar{\mathbf{Q}})\mu_\beta$$

And:

$$\mathbb{E}[\mathbf{Q}\Sigma_\beta \mathbf{Q}\mathbf{X}y] = \sum_{i=1}^n \mathbb{E}[y_i \mathbf{Q}\Sigma_\beta \mathbf{Q}x_i]$$

$$= \frac{n}{(1+\delta_Q)} \mathbb{E}[y_i \mathbf{Q}\Sigma_\beta \mathbf{Q}_{-i}x_i]$$

$$= \frac{n}{(1+\delta_Q)} \mathbb{E}\left[y_i \left(\mathbf{Q}_{-i} - \frac{\frac{1}{n}\mathbf{Q}_{-i}x_ix_i^\top \mathbf{Q}_{-i}}{1+\delta_Q}\right) \Sigma_\beta \mathbf{Q}_{-i}x_i\right]$$

$$= \frac{n}{(1+\delta_Q)} \left(\mathbb{E}[y_i \mathbf{Q}_{-i}\Sigma_\beta \mathbf{Q}_{-i}x_i] - \frac{1}{n(1+\delta_Q)} \mathbb{E}[y_i \mathbf{Q}_{-i}x_ix_i^\top \mathbf{Q}_{-i}\Sigma_\beta \mathbf{Q}_{-i}x_i]\right)$$

$$= \frac{n}{(1+\delta_Q)} \left(\mathbb{E}[\mathbf{Q}\Sigma_\beta \mathbf{Q}]\mu_\beta - \frac{1}{n(1+\delta_Q)} \mathrm{Tr}(\Sigma_\beta \mathbb{E}[\mathbf{Q}\Sigma_\beta \mathbf{Q}])\bar{\mathbf{Q}}\mu_\beta\right)$$

$$= \frac{n}{h(1+\delta_Q)} \bar{\mathbf{Q}}\Sigma_\beta \bar{\mathbf{Q}}\mu_\beta - \frac{n(1-h)}{h} \bar{\mathbf{Q}}\mu_\beta$$

$$= n\left(\frac{1}{h}(\mathbf{I}_p - \gamma\bar{\mathbf{Q}})\bar{\mathbf{Q}}\mu_\beta - \frac{1-h}{h}\bar{\mathbf{Q}}\mu_\beta\right)$$

$$= n(\bar{\mathbf{Q}}\mu_\beta - \frac{\gamma}{h}\bar{\mathbf{Q}}^2\mu_\beta)$$

Thus:

$$\frac{1}{n} \mathbb{E}[\tilde{w}^\top \mathbf{X}\mathbf{X}^\top \mathbf{Q}xw^\top x] = \frac{1}{(1+\delta_R)} \mu^\top \bar{\mathbf{R}}\left(\mathbf{I}_p - 2\gamma\bar{\mathbf{Q}} + \frac{\gamma^2}{h}\bar{\mathbf{Q}}^2\right)\mu_\beta \tag{19}$$

Let us now compute the remaining term:

$$\frac{1}{n} \mathbb{E}[\tilde{w}^\top \mathbf{X}\mathbf{X}^\top \mathbf{Q}x\tilde{w}^\top x] = \frac{1}{n} \mathbb{E}[\tilde{w}^\top \mathbf{X}\mathbf{X}^\top \mathbf{Q}xx^\top \tilde{w}]$$

$$= \frac{1}{n} \mathbb{E}[\tilde{w}^\top \mathbf{X}\mathbf{X}^\top \mathbf{Q}\Sigma_\beta \tilde{w}]$$

$$= \mathbb{E}[\tilde{w}^\top (\mathbf{I}_p - \gamma\mathbf{Q})\Sigma_\beta \tilde{w}]$$

And again by lemma A.8:

$$\frac{1}{n} \mathbb{E}[\tilde{w}^\top \mathbf{X}\mathbf{X}^\top \mathbf{Q}x\tilde{w}^\top x] = \frac{1}{(1+\delta_R)^2} \mu^\top \mathbb{E}[\mathbf{R}(\mathbf{I}_p - \gamma\mathbf{Q})\Sigma_\beta \mathbf{R}]\mu + \frac{\mathrm{Tr}(\Sigma \mathbb{E}[\mathbf{R}(\mathbf{I}_p - \gamma\mathbf{Q})\Sigma_\beta \mathbf{R}])}{N(1+\delta_R)^2} \left(1 - \frac{2}{(1+\delta_R)}\mu^\top \bar{\mathbf{R}}\mu\right)$$

Now let us group all the results as follows.

**Terms without $\alpha$:**   There is only one term which is:

$$T_1 = \mathbb{E}[(\boldsymbol{w}^\top \boldsymbol{x})^2] = \frac{1}{h(1+\delta_Q)} \left((2h-1)\boldsymbol{\mu}_\beta^\top \bar{\mathbf{Q}}\boldsymbol{\mu}_\beta - \gamma\boldsymbol{\mu}_\beta^\top \bar{\mathbf{Q}}^2\boldsymbol{\mu}_\beta\right) + \frac{1-h}{h}$$

$$= \frac{\|\boldsymbol{\mu}_\beta\|^2}{h(\|\boldsymbol{\mu}_\beta\|^2 + 1 + \gamma(1+\delta_Q))} \left(\frac{\|\boldsymbol{\mu}_\beta\|^2 + 1}{\|\boldsymbol{\mu}_\beta\|^2 + 1 + \gamma(1+\delta_Q)} - 2(1-h)\right) + \frac{1-h}{h}$$

**Terms in $\alpha$:**   There are two: $2\,\mathbb{E}[\boldsymbol{w}^\top \boldsymbol{x}\tilde{\boldsymbol{w}}^\top \boldsymbol{x}]$ and $\frac{2}{n}\,\mathbb{E}[\tilde{\boldsymbol{w}}^\top \mathbf{X}\mathbf{X}^\top \mathbf{Q}\boldsymbol{x}\boldsymbol{w}^\top \boldsymbol{x}]$:

$$T_2 = 2\,\mathbb{E}[\boldsymbol{w}^\top \boldsymbol{x}\tilde{\boldsymbol{w}}^\top \boldsymbol{x}] - \frac{2}{n}\,\mathbb{E}[\tilde{\boldsymbol{w}}^\top \mathbf{X}\mathbf{X}^\top \mathbf{Q}\boldsymbol{x}\boldsymbol{w}^\top \boldsymbol{x}]$$

$$= \frac{2}{(1+\delta_R)} \left(\boldsymbol{\mu}_\beta \bar{\mathbf{R}}\boldsymbol{\mu} - \gamma\boldsymbol{\mu}_\beta^\top \bar{\mathbf{R}}\bar{\mathbf{Q}}\boldsymbol{\mu} - \boldsymbol{\mu}^\top \bar{\mathbf{R}}(\mathbf{I}_p - 2\gamma\bar{\mathbf{Q}} + \frac{\gamma^2}{h}\bar{\mathbf{Q}}^2)\boldsymbol{\mu}_\beta\right)$$

$$= \frac{2\gamma}{(1+\delta_R)}\boldsymbol{\mu}^\top \bar{\mathbf{R}}\bar{\mathbf{Q}}\left(\mathbf{I}_p - \frac{\gamma}{h}\bar{\mathbf{Q}}\right)\boldsymbol{\mu}_\beta$$

And using lemma A.5:

$$T_2 = \frac{2\gamma(1+\delta_Q)\beta\|\boldsymbol{\mu}\|^2}{(\|\boldsymbol{\mu}\|^2 + 1 + \tilde{\gamma}(1+\delta_R))\,(\|\boldsymbol{\mu}_\beta\|^2 + 1 + \gamma(1+\delta_Q))} \left(1 - \frac{\gamma(1+\delta_Q)}{h(\|\boldsymbol{\mu}_\beta\|^2 + 1 + \gamma(1+\delta_Q))}\right)$$

**Terms in $\alpha^2$:**   we have three terms:   $\mathbb{E}[(\tilde{\boldsymbol{w}}^\top \boldsymbol{x})^2]$,   $\frac{1}{n^2}\,\mathbb{E}[(\tilde{\boldsymbol{w}}^\top \mathbf{X}\mathbf{X}^\top \mathbf{Q}\boldsymbol{x})^2]$   and $\frac{-2}{n}\,\mathbb{E}[\tilde{\boldsymbol{w}}^\top \mathbf{X}\mathbf{X}^\top \mathbf{Q}\boldsymbol{x}\tilde{\boldsymbol{w}}^\top \boldsymbol{x}]$:

$$T_3 = \mathbb{E}[(\tilde{\boldsymbol{w}}^\top \boldsymbol{x})^2] + \frac{1}{n^2}\,\mathbb{E}[(\tilde{\boldsymbol{w}}^\top \mathbf{X}\mathbf{X}^\top \mathbf{Q}\boldsymbol{x})^2] - \frac{2}{n}\,\mathbb{E}[\tilde{\boldsymbol{w}}^\top \mathbf{X}\mathbf{X}^\top \mathbf{Q}\boldsymbol{x}\tilde{\boldsymbol{w}}^\top \boldsymbol{x}]$$

$$= \frac{\gamma}{(1+\delta_R)^2}\boldsymbol{\mu}^\top \left(\mathbb{E}[\mathbf{R}\bar{\mathbf{Q}}\Sigma_\beta \mathbf{R}] - \mathbb{E}[\mathbf{R}\Sigma_\beta \bar{\mathbf{Q}}\mathbf{R}] + \gamma\,\mathbb{E}[\mathbf{R}\mathbf{Q}\Sigma_\beta \mathbf{Q}\mathbf{R}]\right)\boldsymbol{\mu}$$

$$+ \frac{\gamma}{N(1+\delta_R)^2}\left(1 - \frac{2}{(1+\delta_R)}\boldsymbol{\mu}^\top \bar{\mathbf{R}}\boldsymbol{\mu}\right)\mathrm{Tr}\left(\Sigma(\mathbb{E}[\mathbf{R}\bar{\mathbf{Q}}\Sigma_\beta \mathbf{R}] - \mathbb{E}[\mathbf{R}\Sigma_\beta \bar{\mathbf{Q}}\mathbf{R}] + \gamma\,\mathbb{E}[\mathbf{R}\mathbf{Q}\Sigma_\beta \mathbf{Q}\mathbf{R}])\right)$$

$$= \frac{\gamma^2}{(1+\delta_R)^2}\left[\boldsymbol{\mu}^\top \mathbb{E}[\mathbf{R}\mathbf{Q}\Sigma_\beta \mathbf{Q}\mathbf{R}]\boldsymbol{\mu} + \left(1 - \frac{2}{(1+\delta_R)}\boldsymbol{\mu}^\top \bar{\mathbf{R}}\boldsymbol{\mu}\right)\frac{1}{N}\,\mathrm{Tr}(\Sigma\,\mathbb{E}[\mathbf{R}\mathbf{Q}\Sigma_\beta \mathbf{Q}\mathbf{R}])\right]$$

where the last equality is gotten using lemma A.9.
We also have that:

$$\frac{1}{N}\,\mathrm{Tr}(\Sigma\,\mathbb{E}[\mathbf{R}\mathbf{Q}\Sigma_\beta \mathbf{Q}\mathbf{R}]) = \frac{1}{N}\,\mathrm{Tr}(\mathbb{E}[\Sigma\mathbf{R}\mathbf{Q}\Sigma_\beta \mathbf{Q}\mathbf{R}])$$

$$= \frac{1}{N}\,\mathbb{E}[\mathrm{Tr}(\Sigma\mathbf{R}\mathbf{Q}\Sigma_\beta \mathbf{Q}\mathbf{R})]$$

$$= \frac{1}{N}\,\mathbb{E}[\mathrm{Tr}(\mathbf{R}\Sigma\mathbf{R}\mathbf{Q}\Sigma_\beta \mathbf{Q})]$$

$$= \frac{1}{N}\,\mathrm{Tr}(\mathbb{E}[\mathbf{R}\Sigma\mathbf{R}\mathbf{Q}\Sigma_\beta \mathbf{Q}])$$

$$= \frac{1}{N}\,\mathrm{Tr}(\mathbb{E}[\mathbf{R}\Sigma\mathbf{R}]\,\mathbb{E}[\mathbf{Q}\Sigma_\beta \mathbf{Q}])$$

$$= \frac{1}{h\tilde{h}}\frac{1}{N}\,\mathrm{Tr}(\bar{\mathbf{R}}\Sigma\bar{\mathbf{R}}\bar{\mathbf{Q}}\Sigma_\beta \bar{\mathbf{Q}})$$

And:

$$\boldsymbol{\mu}^\top \mathbb{E}[\mathbf{R}\mathbf{Q}\Sigma_\beta \mathbf{Q}\mathbf{R}]\boldsymbol{\mu} = \mathrm{Tr}(\mathbb{E}[\boldsymbol{\mu}^\top \mathbf{R}\mathbf{Q}\Sigma_\beta \mathbf{Q}\mathbf{R}\boldsymbol{\mu}])$$

$$= \mathbb{E}[\mathrm{Tr}(\mathbf{R}\boldsymbol{\mu}\boldsymbol{\mu}^\top \mathbf{R}\mathbf{Q}\Sigma_\beta \mathbf{Q})]$$

$$= \mathrm{Tr}(\mathbb{E}[\mathbf{R}\boldsymbol{\mu}\boldsymbol{\mu}^\top \mathbf{R}]\,\mathbb{E}[\mathbf{Q}\Sigma_\beta \mathbf{Q}])$$

$$= \frac{1}{h}\,\mathrm{Tr}(\mathbb{E}[\mathbf{R}\boldsymbol{\mu}\boldsymbol{\mu}^\top \mathbf{R}]\bar{\mathbf{Q}}\Sigma_\beta \bar{\mathbf{Q}})$$

Thus:

$$T_3 = \frac{\gamma^2}{h(1+\delta_R)^2}\left[\mathrm{Tr}(\mathbb{E}[\mathbf{R}\boldsymbol{\mu}\boldsymbol{\mu}^\top\mathbf{R}]\bar{\mathbf{Q}}\Sigma_\beta\bar{\mathbf{Q}}) + \left(1 - \frac{2}{(1+\delta_R)}\boldsymbol{\mu}^\top\bar{\mathbf{R}}\boldsymbol{\mu}\right)\frac{1}{\tilde{h}}\frac{1}{N}\mathrm{Tr}(\bar{\mathbf{R}}\Sigma\bar{\mathbf{R}}\bar{\mathbf{Q}}\Sigma_\beta\bar{\mathbf{Q}})\right]$$

Now remains to compute $\mathbb{E}[\mathbf{R}\boldsymbol{\mu}\boldsymbol{\mu}^\top\mathbf{R}]$. For that, we use lemma A.7:

$$\mathbb{E}[\mathbf{R}\boldsymbol{\mu}\boldsymbol{\mu}^\top\mathbf{R}] = \bar{\mathbf{R}}\boldsymbol{\mu}\boldsymbol{\mu}^\top\bar{\mathbf{R}} + \frac{1}{N}\frac{\mathrm{Tr}(\Sigma\bar{\mathbf{R}}\boldsymbol{\mu}\boldsymbol{\mu}^\top\bar{\mathbf{R}})}{(1+\delta_R)^2}\mathbb{E}[\mathbf{R}\Sigma\mathbf{R}]$$

$$= \bar{\mathbf{R}}\boldsymbol{\mu}\boldsymbol{\mu}^\top\bar{\mathbf{R}} + \frac{1}{N}\frac{\boldsymbol{\mu}^\top\bar{\mathbf{R}}\Sigma\bar{\mathbf{R}}\boldsymbol{\mu}}{(1+\delta_R)^2}\frac{1}{\tilde{h}}\bar{\mathbf{R}}\Sigma\bar{\mathbf{R}}$$

And since we are in the regime of $N \to \infty$, then:

$$\frac{1}{N}\boldsymbol{\mu}^\top\bar{\mathbf{R}}\Sigma\bar{\mathbf{R}}\boldsymbol{\mu} = \mathcal{O}(N^{-1})$$

Thus:

$$\mathbb{E}[\mathbf{R}\boldsymbol{\mu}\boldsymbol{\mu}^\top\mathbf{R}] = \bar{\mathbf{R}}\boldsymbol{\mu}\boldsymbol{\mu}^\top\bar{\mathbf{R}} \tag{20}$$

Hence, $T_3$ becomes:

$$T_3 = \frac{\gamma^2}{h(1+\delta_R)^2}\left[\boldsymbol{\mu}^\top\bar{\mathbf{R}}\bar{\mathbf{Q}}\Sigma_\beta\bar{\mathbf{Q}}\bar{\mathbf{R}}\boldsymbol{\mu} + \left(1 - \frac{2}{(1+\delta_R)}\boldsymbol{\mu}^\top\bar{\mathbf{R}}\boldsymbol{\mu}\right)\frac{1}{\tilde{h}}\frac{1}{N}\mathrm{Tr}(\bar{\mathbf{R}}\Sigma\bar{\mathbf{R}}\bar{\mathbf{Q}}\Sigma_\beta\bar{\mathbf{Q}})\right]$$

And we also have that:

$$\boldsymbol{\mu}^\top\bar{\mathbf{R}}\bar{\mathbf{Q}}\Sigma_\beta\bar{\mathbf{Q}}\bar{\mathbf{R}}\boldsymbol{\mu} = \boldsymbol{\mu}^\top\bar{\mathbf{R}}\bar{\mathbf{Q}}\boldsymbol{\mu}_\beta\boldsymbol{\mu}_\beta^\top\bar{\mathbf{Q}}\bar{\mathbf{R}}\boldsymbol{\mu} + \boldsymbol{\mu}^\top\bar{\mathbf{R}}\bar{\mathbf{Q}}^2\bar{\mathbf{R}}\boldsymbol{\mu}$$

$$= \left(\boldsymbol{\mu}^\top\bar{\mathbf{R}}\bar{\mathbf{Q}}\boldsymbol{\mu}_\beta\right)^2 + \boldsymbol{\mu}^\top\bar{\mathbf{R}}\bar{\mathbf{Q}}^2\bar{\mathbf{R}}\boldsymbol{\mu}$$

And:

$$\frac{1}{N}\mathrm{Tr}(\bar{\mathbf{R}}\Sigma\bar{\mathbf{R}}\bar{\mathbf{Q}}\Sigma_\beta\bar{\mathbf{Q}}) = \frac{1}{N}\mathrm{Tr}(\bar{\mathbf{R}}^2\bar{\mathbf{Q}}^2)$$

Therefore:

$$T_3 = \frac{\gamma^2}{h(1+\delta_R)^2}\left[\left(\boldsymbol{\mu}^\top\bar{\mathbf{R}}\bar{\mathbf{Q}}\boldsymbol{\mu}_\beta\right)^2 + \boldsymbol{\mu}^\top\bar{\mathbf{R}}\bar{\mathbf{Q}}^2\bar{\mathbf{R}}\boldsymbol{\mu} + \left(1 - \frac{2}{(1+\delta_R)}\boldsymbol{\mu}^\top\bar{\mathbf{R}}\boldsymbol{\mu}\right)\frac{1}{\tilde{h}}\frac{1}{N}\mathrm{Tr}(\bar{\mathbf{R}}^2\bar{\mathbf{Q}}^2)\right] \tag{21}$$

Then using lemmas A.4 and A.5:

$$T_3 = \frac{\gamma^2}{h(1+\delta_R)^2}\left[\left(\boldsymbol{\mu}^\top\bar{\mathbf{R}}\bar{\mathbf{Q}}\boldsymbol{\mu}_\beta\right)^2 + \boldsymbol{\mu}^\top\bar{\mathbf{R}}\bar{\mathbf{Q}}^2\bar{\mathbf{R}}\boldsymbol{\mu}\right] + \frac{\gamma^2}{h(1+\delta_R)^2}\left(1 - \frac{2}{(1+\delta_R)}\boldsymbol{\mu}^\top\bar{\mathbf{R}}\boldsymbol{\mu}\right)\frac{1}{\tilde{h}}\frac{1}{N}\mathrm{Tr}(\bar{\mathbf{R}}^2\bar{\mathbf{Q}}^2)$$

$$= \frac{\gamma^2(1+\delta_Q)^2}{h}[\frac{\|\boldsymbol{\mu}\|^2}{\lambda_R^2}\left(\frac{\beta^2\|\boldsymbol{\mu}\|^2}{\lambda_Q^2} + \frac{1}{(1+\gamma(1+\delta_Q))^2}\left(1 + \frac{\beta^2\|\boldsymbol{\mu}\|^2\|\boldsymbol{\mu}_\beta\|^2}{\lambda_Q^2} - \frac{2\beta^2\|\boldsymbol{\mu}\|^2}{\lambda_Q}\right)\right) +$$

$$\frac{\tilde{\eta}}{(1+\gamma(1+\delta_Q))^2(1+\tilde{\gamma}(1+\delta_R))^2}\left(1 - \frac{2\|\boldsymbol{\mu}\|^2}{\lambda_R}\right)]$$

$$= \frac{\gamma^2(1+\delta_Q)^2}{h}\left[\frac{\|\boldsymbol{\mu}\|^2}{\lambda_R^2}\left(\frac{\beta^2\|\boldsymbol{\mu}\|^2}{\lambda_Q^2} + \frac{1-h}{\eta}\left(1 + \frac{\beta^2\|\boldsymbol{\mu}\|^2\|\boldsymbol{\mu}_\beta\|^2}{\lambda_Q^2} - \frac{2\beta^2\|\boldsymbol{\mu}\|^2}{\lambda_Q} + (1-\tilde{h})\left(1 - \frac{2\|\boldsymbol{\mu}\|^2}{\lambda_R}\right)\right)\right)\right]$$

Finally:

$$T_1 = \frac{\|\boldsymbol{\mu}_\beta\|^2}{h\lambda_Q}\left(\frac{\|\boldsymbol{\mu}_\beta\|^2 + 1}{\lambda_Q} - 2(1-h)\right) + \frac{1-h}{h} \tag{22}$$

$$T_2 = \frac{2\gamma\beta(1+\delta_Q)\|\boldsymbol{\mu}\|^2}{\lambda_R\lambda_Q}\left(1 - \frac{\gamma(1+\delta_Q)}{h\lambda_Q}\right) \tag{23}$$

$$T_3 = \frac{\gamma^2(1+\delta_Q)^2}{h}\left[\frac{\|\boldsymbol{\mu}\|^2}{\lambda_R^2}\left(\frac{\beta^2\|\boldsymbol{\mu}\|^2}{\lambda_Q^2} + \frac{1-h}{\eta}\left(1 + \frac{\beta^2\|\boldsymbol{\mu}\|^2\|\boldsymbol{\mu}_\beta\|^2}{\lambda_Q^2} - \frac{2\beta^2\|\boldsymbol{\mu}\|^2}{\lambda_Q} + (1-\tilde{h})\left(1 - \frac{2\|\boldsymbol{\mu}\|^2}{\lambda_R}\right)\right)\right)\right] \tag{24}$$

And the expression of the second order expectation reads:

$$\mathbb{E}[(\boldsymbol{w}_\alpha^\top\boldsymbol{x})^2] = T_1 + \alpha T_2 + \alpha^2 T_3 \tag{25}$$

And finally, Theorem 4.2 follows:

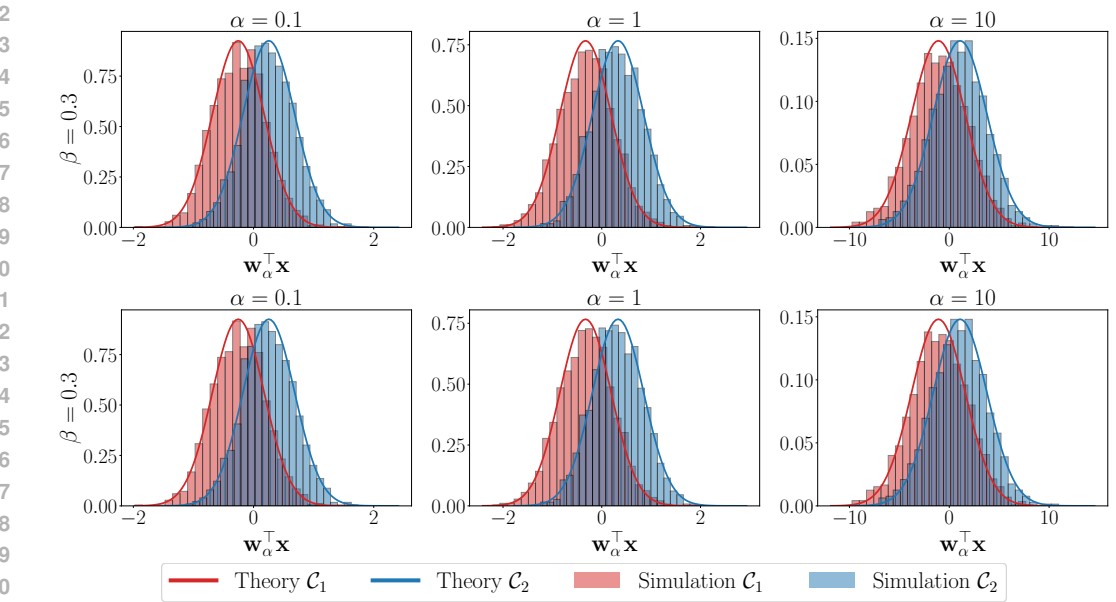

Figure 6: Distribution of the decision function $\boldsymbol{w}_\alpha^\top \boldsymbol{x}$ for different values of $\alpha$ (per column) and $\beta$ (per row). Here we have $N = 5000$, $n = 200$, $p = 400$, $\|\boldsymbol{\mu}\| = 1.5$, $\|\boldsymbol{\mu}^\perp\| = 1$, $\gamma = \tilde{\gamma} = 1$. The theoretical Gaussian distributions are predicted as per Theorem 4.2.

**Theorem B.1** (Gaussianity of the fine-tuned Ridge model). *Let $\boldsymbol{w}_\alpha$ be the fine-tuned classifier as defined in equation $\alpha$-FTC and suppose that Assumption 4.1 holds. The decision function $\boldsymbol{w}_\alpha^\top \boldsymbol{x}$, on some test sample $\boldsymbol{x} \in \mathcal{C}_a$ independent of $\mathbf{X}$, satisfies:*

$$\boldsymbol{w}_\alpha^\top \boldsymbol{x} \xrightarrow{\mathcal{D}} \mathcal{N}\left((-1)^a m_\alpha, \nu_\alpha - m_\alpha^2\right),$$

*where:*

$$m_\alpha = \frac{1}{\lambda_Q}\left(\|\boldsymbol{\mu}_\beta\|^2 + \frac{\alpha\beta\gamma(1+\delta_Q)}{\lambda_R}\|\boldsymbol{\mu}\|^2\right),$$

$$\nu_\alpha = T_1 + \alpha T_2 + \alpha^2 T_3.$$

*With:*

$$T_1 = \frac{\|\boldsymbol{\mu}_\beta\|^2}{h\lambda_Q}\left(\frac{\|\boldsymbol{\mu}_\beta\|^2 + 1}{\lambda_Q} - 2(1-h)\right) + \frac{1-h}{h},$$

$$T_2 = \frac{2\gamma\beta(1+\delta_Q)\|\boldsymbol{\mu}\|^2}{\lambda_R\lambda_Q}\left(1 - \frac{\gamma(1+\delta_Q)}{h\lambda_Q}\right),$$

$$T_3 = \frac{\gamma^2(1+\delta_Q)^2}{h}\times$$

$$\left[\frac{\|\boldsymbol{\mu}\|^2}{\lambda_R^2}\left(\frac{\beta^2\|\boldsymbol{\mu}\|^2}{\lambda_Q^2} + \frac{1-h}{\eta}\left(1 + \frac{\beta^2\|\boldsymbol{\mu}\|^2\|\boldsymbol{\mu}_\beta\|^2}{\lambda_Q^2} - \frac{2\beta^2\|\boldsymbol{\mu}\|^2}{\lambda_Q}\right)\right) + \frac{(1-h)(1-\tilde{h})}{\eta}\left(1 - \frac{2\|\boldsymbol{\mu}\|^2}{\lambda_R}\right)\right].$$

### B.3 FINDING OPTIMAL $\alpha^*$

Since the test accuracy is given by $\mathcal{A}_{\text{test}} = 1 - \varphi\left((\nu_\alpha - m_\alpha^2)^{-\frac{1}{2}} m_\alpha\right)$ as in Proposition 4.3, and that $\phi(x)$ is a non-increasing function, then finding the optimal $\alpha^*$ that maximizes the test accuracy boils down to maximizing the term inside $\phi$. Thus, by computing the derivative with respect to $\alpha$ of $(\nu_\alpha - m_\alpha^2)^{-\frac{1}{2}} m_\alpha$ and finding the zero of the gradient gives us the final form of the best scaling parameter $\alpha^*$:

$$\alpha^* = \frac{\lambda_R T_2\|\boldsymbol{\mu}_\beta\|^2 - 2\beta\gamma T_1(1+\delta_Q)\|\boldsymbol{\mu}\|^2}{\beta\gamma T_2(1+\delta_Q)\|\boldsymbol{\mu}\|^2 - 2\lambda_R T_3\|\boldsymbol{\mu}_\beta\|^2}$$

And since the worst test accuracy is 50% (random classification), which is obtained for $m_\alpha = 0$, then solving the previous equation gives the worst scaling $\bar{\alpha}$ to use:

$$\bar{\alpha} = -\frac{\lambda_R \|\boldsymbol{\mu}_\beta\|^2}{\beta\gamma(1 + \delta_Q)\|\boldsymbol{\mu}\|^2}$$

## C   RMT ANALYSIS OF THE FINE-TUNED CLASSIFIER: THE CASE OF RANDOM SOURCE VECTOR

Let $\boldsymbol{x} \sim \mathcal{N}((-1)^a\boldsymbol{\mu}_\beta, \mathbf{I}_p)$ be an independent test sample. Let $\tilde{\boldsymbol{w}}$ be the source classifier (obtained through some optimization algorithm). We recall that:

$$\boldsymbol{w}_\alpha = \boldsymbol{w} + \alpha\tilde{\boldsymbol{w}} - \frac{\alpha}{n}\mathbf{Q}(\gamma)\mathbf{X}\mathbf{X}^\top\tilde{\boldsymbol{w}}, \quad \boldsymbol{w} = \frac{1}{n}\mathbf{Q}(\gamma)\mathbf{X}\boldsymbol{y}$$

### C.1   TEST EXPECTATION

We have that:

$$\mathbb{E}[\boldsymbol{w}_\alpha^\top \boldsymbol{x}] = \mathbb{E}[\boldsymbol{w}^\top \boldsymbol{x}] + \alpha\,\mathbb{E}[\tilde{\boldsymbol{w}}^\top \boldsymbol{x}] - \frac{\alpha}{n}\,\mathbb{E}[\tilde{\boldsymbol{w}}^\top\mathbf{X}\mathbf{X}^\top\mathbf{Q}\boldsymbol{x}] \tag{26}$$

Let us compute each term of this previous sum.
First, using lemma A.6, we have that, since $\boldsymbol{x}$ is independent of $\mathbf{X}$:

$$\mathbb{E}[\boldsymbol{w}^\top \boldsymbol{x}] = \mathbb{E}[\boldsymbol{w}]^\top \mathbb{E}[\boldsymbol{x}] = \frac{(-1)^a}{1 + \delta_Q}\boldsymbol{\mu}_\beta^\top \bar{\mathbf{Q}}\boldsymbol{\mu}_\beta$$

And we have that:

$$\mathbb{E}[\tilde{\boldsymbol{w}}^\top \boldsymbol{x}] = (-1)^a\tilde{\boldsymbol{w}}^\top \boldsymbol{\mu}_\beta$$

And:

$$\frac{\alpha}{n}\,\mathbb{E}[\tilde{\boldsymbol{w}}^\top\mathbf{X}\mathbf{X}^\top\mathbf{Q}\boldsymbol{x}] = \frac{\alpha}{n}\sum_{i=1}^{n}\mathbb{E}[\tilde{\boldsymbol{w}}^\top\boldsymbol{x}_i\boldsymbol{x}_i^\top\mathbf{Q}\boldsymbol{x}]$$

$$= \frac{\alpha}{n(1 + \delta_Q)}\sum_{i=1}^{n}\mathbb{E}[\tilde{\boldsymbol{w}}^\top\boldsymbol{x}_i\boldsymbol{x}_i^\top\mathbf{Q}_{-i}\boldsymbol{x}]$$

$$= \frac{\alpha}{n(1 + \delta_Q)}\sum_{i=1}^{n}\mathbb{E}[\tilde{\boldsymbol{w}}^\top\Sigma_\beta\mathbf{Q}_{-i}\boldsymbol{x}]$$

$$= \frac{(-1)^a\alpha}{1 + \delta_Q}\tilde{\boldsymbol{w}}^\top\Sigma_\beta\bar{\mathbf{Q}}\boldsymbol{\mu}_\beta$$

Thus:

$$\mathbb{E}[\boldsymbol{w}_\alpha^\top \boldsymbol{x}] = (-1)^a\left(\frac{1}{1 + \delta_Q}\boldsymbol{\mu}_\beta^\top\bar{\mathbf{Q}}\boldsymbol{\mu}_\beta + \alpha\tilde{\boldsymbol{w}}^\top\boldsymbol{\mu}_\beta - \frac{\alpha}{1 + \delta_Q}\tilde{\boldsymbol{w}}^\top\Sigma_\beta\bar{\mathbf{Q}}\boldsymbol{\mu}_\beta\right)$$

$$= (-1)^a\left(\frac{1}{1 + \delta_Q}\boldsymbol{\mu}_\beta^\top\bar{\mathbf{Q}}\boldsymbol{\mu}_\beta + \alpha\tilde{\boldsymbol{w}}^\top\boldsymbol{\mu}_\beta - \alpha\tilde{\boldsymbol{w}}^\top(\bar{\mathbf{Q}}^{-1} - \gamma\mathbf{I}_p)\bar{\mathbf{Q}}\boldsymbol{\mu}_\beta\right)$$

$$= (-1)^a\left(\frac{1}{1 + \delta_Q}\boldsymbol{\mu}_\beta^\top\bar{\mathbf{Q}}\boldsymbol{\mu}_\beta + \alpha\gamma\tilde{\boldsymbol{w}}^\top\bar{\mathbf{Q}}\boldsymbol{\mu}_\beta\right)$$

Using the forumlas in lemma A.5:

$$\mathbb{E}[\boldsymbol{w}_\alpha^\top \boldsymbol{x}] = \frac{(-1)^a}{\|\boldsymbol{\mu}_\beta\|^2 + 1 + \gamma(1 + \delta_Q)}\left(\|\boldsymbol{\mu}_\beta\|^2 + \alpha\gamma(1 + \delta_Q)\tilde{\boldsymbol{w}}^\top\boldsymbol{\mu}_\beta\right) \tag{27}$$

### C.2   TEST VARIANCE

To compute the variance of $\boldsymbol{w}_\alpha^\top\boldsymbol{x}$, it suffices to compute the second moment: $\mathbb{E}[(\boldsymbol{w}_\alpha^\top\boldsymbol{x})^2]$.

$$\mathbb{E}[(\boldsymbol{w}_\alpha^\top \boldsymbol{x})^2] = \mathbb{E}[(\boldsymbol{w}^\top\boldsymbol{x} + \alpha\tilde{\boldsymbol{w}}^\top\boldsymbol{x})^2] + \frac{\alpha^2}{n^2}(\tilde{\boldsymbol{w}}^\top\mathbf{X}\mathbf{X}^\top\mathbf{Q}\boldsymbol{x})^2 - \frac{2\alpha}{n}\tilde{\boldsymbol{w}}^\top\mathbf{X}\mathbf{X}^\top\mathbf{Q}\boldsymbol{x}(\boldsymbol{w}^\top\boldsymbol{x} + \alpha\tilde{\boldsymbol{w}}^\top\boldsymbol{x})]$$

$$\tag{28}$$

**First term:** We start by computing
$$\mathbb{E}[(\boldsymbol{w}^\top \boldsymbol{x} + \alpha \tilde{\boldsymbol{w}}^\top \boldsymbol{x})^2] = \mathbb{E}[(\boldsymbol{w}^\top \boldsymbol{x})^2] + \alpha^2 \, \mathbb{E}[(\tilde{\boldsymbol{w}}^\top \boldsymbol{x})^2] + 2\alpha \, \mathbb{E}[\boldsymbol{w}^\top \boldsymbol{x} \tilde{\boldsymbol{w}}^\top \boldsymbol{x}]$$

We have that, as proved in Firdoussi & Seddik (2024):

$$\mathbb{E}[(\boldsymbol{w}^\top \boldsymbol{x})^2] = \frac{1}{h(1+\delta_Q)} \left( \frac{1}{1+\delta_Q} \boldsymbol{\mu}_\beta^\top \bar{\mathbf{Q}} \Sigma_\beta \bar{\mathbf{Q}} \boldsymbol{\mu}_\beta - 2(1-h) \boldsymbol{\mu}_\beta^\top \bar{\mathbf{Q}} \boldsymbol{\mu}_\beta \right) + \frac{1-h}{h}$$

$$= \frac{1}{h(1+\delta_Q)} \left( \frac{1}{1+\delta_Q} \left( (\boldsymbol{\mu}_\beta^\top \bar{\mathbf{Q}} \boldsymbol{\mu}_\beta)^2 + \boldsymbol{\mu}_\beta^\top \bar{\mathbf{Q}}^2 \boldsymbol{\mu}_\beta \right) - 2(1-h) \boldsymbol{\mu}_\beta^\top \bar{\mathbf{Q}} \boldsymbol{\mu}_\beta \right) + \frac{1-h}{h}$$

$$= \frac{\|\boldsymbol{\mu}_\beta\|^2}{h(\|\boldsymbol{\mu}_\beta\|^2 + 1 + \gamma(1+\delta_Q))} \left( \frac{\|\boldsymbol{\mu}_\beta\|^2 + 1}{\|\boldsymbol{\mu}_\beta\|^2 + 1 + \gamma(1+\delta_Q)} - 2(1-h) \right) + \frac{1-h}{h}$$

And we have that:
$$\mathbb{E}[(\tilde{\boldsymbol{w}}^\top \boldsymbol{x})^2] = \tilde{\boldsymbol{w}}^\top \Sigma_\beta \tilde{\boldsymbol{w}}$$

And:
$$\mathbb{E}[\boldsymbol{w}^\top \boldsymbol{x} \tilde{\boldsymbol{w}}^\top \boldsymbol{x}] = \mathbb{E}[\boldsymbol{w}]^\top \Sigma_\beta \tilde{\boldsymbol{w}} = \frac{1}{1+\delta_Q} \boldsymbol{\mu}_\beta^\top \bar{\mathbf{Q}} \Sigma_\beta \tilde{\boldsymbol{w}}$$

Thus we have the first sum.

**Second term:** Now let us compute the expectation of the second term:
$$\frac{1}{n^2} \, \mathbb{E}[(\tilde{\boldsymbol{w}}^\top \mathbf{X} \mathbf{X}^\top \mathbf{Q} \boldsymbol{x})^2] = \frac{1}{n^2} \, \mathbb{E}[\tilde{\boldsymbol{w}}^\top \mathbf{X} \mathbf{X}^\top \mathbf{Q} \boldsymbol{x} \boldsymbol{x}^\top \mathbf{X} \mathbf{X}^\top \mathbf{Q} \tilde{\boldsymbol{w}}]$$

$$= \tilde{\boldsymbol{w}}^\top \, \mathbb{E}[\frac{1}{n} \mathbf{X} \mathbf{X}^\top \mathbf{Q} \Sigma_\beta \frac{1}{n} \mathbf{X} \mathbf{X}^\top \mathbf{Q}] \tilde{\boldsymbol{w}}$$

$$= \tilde{\boldsymbol{w}}^\top \, \mathbb{E}[(\mathbf{Q}^{-1} - \gamma \mathbf{I}_p) \mathbf{Q} \Sigma_\beta (\mathbf{Q}^{-1} - \gamma \mathbf{I}_p) \mathbf{Q}] \tilde{\boldsymbol{w}}$$

$$= \tilde{\boldsymbol{w}}^\top \, \mathbb{E}[(\mathbf{I}_p - \gamma \mathbf{Q}) \Sigma_\beta (\mathbf{I}_p - \gamma \mathbf{Q})] \tilde{\boldsymbol{w}}$$

$$= \tilde{\boldsymbol{w}}^\top \, \mathbb{E}\left[ \Sigma_\beta - \gamma \Sigma_\beta \mathbf{Q} - \gamma \mathbf{Q} \Sigma_\beta + \gamma^2 \mathbf{Q} \Sigma_\beta \mathbf{Q} \right] \tilde{\boldsymbol{w}}$$

$$= \tilde{\boldsymbol{w}}^\top \left( \Sigma_\beta - \gamma \Sigma_\beta \bar{\mathbf{Q}} - \gamma \bar{\mathbf{Q}} \Sigma_\beta + \gamma^2 \right) \tilde{\boldsymbol{w}}$$

$$= \tilde{\boldsymbol{w}}^\top \Sigma_\beta \boldsymbol{w} - 2\gamma \tilde{\boldsymbol{w}}^\top \Sigma_\beta \bar{\mathbf{Q}} \tilde{\boldsymbol{w}} + \gamma^2 \tilde{\boldsymbol{w}}^\top \, \mathbb{E}[\mathbf{Q} \Sigma_\beta \mathbf{Q}] \tilde{\boldsymbol{w}}$$

**Third term:** Now we will compute the last term: $\frac{2\alpha}{n} \, \mathbb{E}[\tilde{\boldsymbol{w}}^\top \mathbf{X} \mathbf{X}^\top \mathbf{Q} \boldsymbol{x} (\boldsymbol{w}^\top \boldsymbol{x} + \alpha \tilde{\boldsymbol{w}}^\top \boldsymbol{x})]$.
We have that:
$$\frac{1}{n} \, \mathbb{E}[\tilde{\boldsymbol{w}}^\top \mathbf{X} \mathbf{X}^\top \mathbf{Q} \boldsymbol{x} \boldsymbol{x}^\top \boldsymbol{w}] = \tilde{\boldsymbol{w}}^\top \, \mathbb{E}[(\mathbf{Q}^{-1} - \gamma \mathbf{I}_p) \mathbf{Q} \Sigma_\beta \boldsymbol{w}]$$

$$= \tilde{\boldsymbol{w}}^\top \, \mathbb{E}[(\mathbf{I}_p - \gamma \mathbf{Q}) \Sigma_\beta \boldsymbol{w}]$$

$$= \tilde{\boldsymbol{w}}^\top \Sigma_\beta \, \mathbb{E}[\boldsymbol{w}] - \gamma \tilde{\boldsymbol{w}}^\top \, \mathbb{E}[\mathbf{Q} \Sigma_\beta \boldsymbol{w}]$$

$$= \tilde{\boldsymbol{w}}^\top \frac{\Sigma_\beta}{1+\delta_Q} \bar{\mathbf{Q}} \boldsymbol{\mu}_\beta - \gamma \tilde{\boldsymbol{w}}^\top \, \mathbb{E}[\mathbf{Q} \Sigma_\beta \boldsymbol{w}]$$

$$= \tilde{\boldsymbol{w}}^\top \frac{\Sigma_\beta}{1+\delta_Q} \bar{\mathbf{Q}} \boldsymbol{\mu}_\beta - \gamma \frac{1}{n} \sum_{i=1}^n \tilde{\boldsymbol{w}}^\top \, \mathbb{E}[\mathbf{Q} \Sigma_\beta \mathbf{Q} y_i \boldsymbol{x}_i]$$

$$= \tilde{\boldsymbol{w}}^\top \frac{\Sigma_\beta}{1+\delta_Q} \bar{\mathbf{Q}} \boldsymbol{\mu}_\beta - \gamma \tilde{\boldsymbol{w}}^\top \, \mathbb{E}[\mathbf{Q} \Sigma_\beta \mathbf{Q} y_i \boldsymbol{x}_i]$$

$$= \tilde{\boldsymbol{w}}^\top \frac{\Sigma_\beta}{1+\delta_Q} \bar{\mathbf{Q}} \boldsymbol{\mu}_\beta - \frac{\gamma}{1+\delta_Q} \tilde{\boldsymbol{w}}^\top \, \mathbb{E}[\mathbf{Q} \Sigma_\beta \mathbf{Q}_{-i} y_i \boldsymbol{x}_i]$$

$$= \tilde{\boldsymbol{w}}^\top \frac{\Sigma_\beta}{1+\delta_Q} \bar{\mathbf{Q}} \boldsymbol{\mu}_\beta - \frac{\gamma}{1+\delta_Q} \tilde{\boldsymbol{w}}^\top \, \mathbb{E}\left[ \left( \mathbf{Q}_{-i} - \frac{\frac{1}{n} \mathbf{Q}_{-i} \boldsymbol{x}_i \boldsymbol{x}_i^\top \mathbf{Q}_{-i}}{1+\delta_Q} \right) \Sigma_\beta \mathbf{Q}_{-i} y_i \boldsymbol{x}_i \right]$$

$$= \tilde{\boldsymbol{w}}^\top \frac{\Sigma_\beta}{1+\delta_Q} \bar{\mathbf{Q}} \boldsymbol{\mu}_\beta - \frac{\gamma}{1+\delta_Q} \tilde{\boldsymbol{w}}^\top \, \mathbb{E}[\mathbf{Q}_{-i} \Sigma_\beta \mathbf{Q}_{-i} y_i \boldsymbol{x}_i] + \frac{\gamma}{n(1+\delta_Q)^2} \tilde{\boldsymbol{w}}^\top \, \mathbb{E}[\mathbf{Q}_{-i} \boldsymbol{x}_i \boldsymbol{x}_i^\top \mathbf{Q}_{-i} \Sigma_\beta \mathbf{Q}_{-i} y_i \boldsymbol{x}_i]$$

$$= \tilde{\boldsymbol{w}}^\top \frac{\Sigma_\beta}{1+\delta_Q} \bar{\mathbf{Q}} \boldsymbol{\mu}_\beta - \frac{\gamma}{1+\delta_Q} \tilde{\boldsymbol{w}}^\top \, \mathbb{E}[\mathbf{Q} \Sigma_\beta \mathbf{Q}] \boldsymbol{\mu}_\beta + \frac{\gamma}{n(1+\delta_Q)^2} \mathrm{Tr}(\Sigma_\beta \, \mathbb{E}[\mathbf{Q} \Sigma_\beta \mathbf{Q}]) \tilde{\boldsymbol{w}}^\top \bar{\mathbf{Q}} \boldsymbol{\mu}_\beta$$

And:

$$\frac{1}{n}\mathbb{E}[\tilde{\boldsymbol{w}}^\top \mathbf{X}\mathbf{X}^\top \mathbf{Q}\boldsymbol{x}\boldsymbol{x}^\top \tilde{\boldsymbol{w}}] = \tilde{\boldsymbol{w}}^\top \mathbb{E}[(\mathbf{Q}^{-1} - \gamma\mathbf{I}_p)\mathbf{Q}\Sigma_\beta]\tilde{\boldsymbol{w}}$$
$$= \tilde{\boldsymbol{w}}^\top \mathbb{E}[(\mathbf{I}_p - \gamma\mathbf{Q})\Sigma_\beta]\tilde{\boldsymbol{w}}$$
$$= \tilde{\boldsymbol{w}}^\top \Sigma_\beta \tilde{\boldsymbol{w}} - \gamma\tilde{\boldsymbol{w}}^\top \bar{\mathbf{Q}}\Sigma_\beta\tilde{\boldsymbol{w}}$$

**Grouping all the terms:** Thus, we now that we have the expression of all the term, we will group them in the following way:

$$\mathbb{E}[(\boldsymbol{w}_\alpha^\top \boldsymbol{x})^2] = T_1 + \alpha T_2 + \alpha^2 T_3$$

**Terms without $\alpha$:**

$$T_1 = \frac{\|\boldsymbol{\mu}_\beta\|^2}{h(\|\boldsymbol{\mu}_\beta\|^2 + 1 + \gamma(1 + \delta_Q))}\left(\frac{\|\boldsymbol{\mu}_\beta\|^2 + 1}{\|\boldsymbol{\mu}_\beta\|^2 + 1 + \gamma(1 + \delta_Q)} - 2(1 - h)\right) + \frac{1 - h}{h} \qquad (29)$$

**Terms in $\alpha$:** There are two : $2\,\mathbb{E}[\boldsymbol{w}^\top \boldsymbol{x}\tilde{\boldsymbol{w}}^\top \boldsymbol{x}]$ and $\frac{2}{n}\mathbb{E}[\tilde{\boldsymbol{w}}^\top \mathbf{X}\mathbf{X}^\top \mathbf{Q}\boldsymbol{x}\boldsymbol{w}^\top \boldsymbol{x}]$:

$$T_2 = 2\,\mathbb{E}[\boldsymbol{w}^\top \boldsymbol{x}\tilde{\boldsymbol{w}}^\top \boldsymbol{x}] - \frac{2}{n}\mathbb{E}[\tilde{\boldsymbol{w}}^\top \mathbf{X}\mathbf{X}^\top \mathbf{Q}\boldsymbol{x}\boldsymbol{w}^\top \boldsymbol{x}]$$
$$= \frac{2\gamma}{h(1 + \delta_Q)}\left(\tilde{\boldsymbol{w}}^\top \bar{\mathbf{Q}}\Sigma_\beta\bar{\mathbf{Q}}\boldsymbol{\mu}_\beta - (1 - h)(1 + \delta_Q)\tilde{\boldsymbol{w}}^\top \bar{\mathbf{Q}}\boldsymbol{\mu}_\beta\right)$$
$$= \frac{2\gamma}{h(1 + \delta_Q)}\left(\tilde{\boldsymbol{w}}^\top \bar{\mathbf{Q}}\boldsymbol{\mu}_\beta\boldsymbol{\mu}_\beta^\top \bar{\mathbf{Q}}\boldsymbol{\mu}_\beta + \tilde{\boldsymbol{w}}^\top \bar{\mathbf{Q}}^2\boldsymbol{\mu}_\beta - (1 - h)(1 + \delta_Q)\tilde{\boldsymbol{w}}^\top \bar{\mathbf{Q}}\boldsymbol{\mu}_\beta\right)$$

And we have that:

$$\tilde{\boldsymbol{w}}^\top \bar{\mathbf{Q}}\boldsymbol{\mu}_\beta\boldsymbol{\mu}_\beta^\top \bar{\mathbf{Q}}\boldsymbol{\mu}_\beta = \frac{(1 + \delta_Q)^2\|\boldsymbol{\mu}_\beta\|^2\tilde{\boldsymbol{w}}^\top \boldsymbol{\mu}_\beta}{(\|\boldsymbol{\mu}_\beta\|^2 + 1 + \gamma(1 + \delta_Q))^2}, \quad \tilde{\boldsymbol{w}}^\top \bar{\mathbf{Q}}^2\boldsymbol{\mu}_\beta = \frac{(1 + \delta_Q)^2\tilde{\boldsymbol{w}}^\top \boldsymbol{\mu}_\beta}{(\|\boldsymbol{\mu}_\beta\|^2 + 1 + \gamma(1 + \delta_Q))^2}.$$

Thus:

$$T_2 = \frac{2\gamma(1 + \delta_Q)\tilde{\boldsymbol{w}}^\top \boldsymbol{\mu}_\beta}{h(\|\boldsymbol{\mu}_\beta\|^2 + 1 + \gamma(1 + \delta_Q))}\left(\frac{\|\boldsymbol{\mu}_\beta\|^2 + 1}{\|\boldsymbol{\mu}_\beta\|^2 + 1 + \gamma(1 + \delta_Q)} - (1 - h)\right)$$

**Terms in $\alpha^2$:** we have three terms: $\mathbb{E}[(\tilde{\boldsymbol{w}}^\top \boldsymbol{x})^2]$, $\frac{1}{n^2}\mathbb{E}[(\tilde{\boldsymbol{w}}^\top \mathbf{X}\mathbf{X}^\top \mathbf{Q}\boldsymbol{x})^2]$ and $\frac{-2}{n}\mathbb{E}[\tilde{\boldsymbol{w}}^\top \mathbf{X}\mathbf{X}^\top \mathbf{Q}\boldsymbol{x}\tilde{\boldsymbol{w}}^\top \boldsymbol{x}]$:

$$T_3 = \mathbb{E}[(\tilde{\boldsymbol{w}}^\top \boldsymbol{x})^2] + \frac{1}{n^2}\mathbb{E}[(\tilde{\boldsymbol{w}}^\top \mathbf{X}\mathbf{X}^\top \mathbf{Q}\boldsymbol{x})^2] - \frac{2}{n}\mathbb{E}[\tilde{\boldsymbol{w}}^\top \mathbf{X}\mathbf{X}^\top \mathbf{Q}\boldsymbol{x}\tilde{\boldsymbol{w}}^\top \boldsymbol{x}]$$
$$= \tilde{\boldsymbol{w}}^\top \Sigma_\beta\tilde{\boldsymbol{w}} + \tilde{\boldsymbol{w}}^\top \Sigma_\beta\tilde{\boldsymbol{w}} - 2\gamma\tilde{\boldsymbol{w}}^\top \Sigma_\beta\bar{\mathbf{Q}}\tilde{\boldsymbol{w}} + \gamma^2\tilde{\boldsymbol{w}}^\top \mathbb{E}[\mathbf{Q}\Sigma_\beta\mathbf{Q}]\tilde{\boldsymbol{w}} - 2\tilde{\boldsymbol{w}}^\top \Sigma_\beta\tilde{\boldsymbol{w}} + 2\gamma\tilde{\boldsymbol{w}}^\top \bar{\mathbf{Q}}\Sigma_\beta\tilde{\boldsymbol{w}}$$
$$= \gamma^2\tilde{\boldsymbol{w}}^\top \mathbb{E}[\mathbf{Q}\Sigma_\beta\mathbf{Q}]\tilde{\boldsymbol{w}}$$
$$= \frac{\gamma^2}{h}\tilde{\boldsymbol{w}}^\top \bar{\mathbf{Q}}\Sigma_\beta\bar{\mathbf{Q}}\tilde{\boldsymbol{w}}$$
$$= \frac{\gamma^2}{h}\left((\tilde{\boldsymbol{w}}^\top \bar{\mathbf{Q}}\boldsymbol{\mu}_\beta)^2 + \tilde{\boldsymbol{w}}^\top \bar{\mathbf{Q}}^2\tilde{\boldsymbol{w}}\right)$$
$$= \frac{\gamma^2(1 + \delta_Q)^2}{h}\left(\frac{(\tilde{\boldsymbol{w}}^\top \boldsymbol{\mu}_\beta)^2}{(\|\boldsymbol{\mu}_\beta\|^2 + 1 + \gamma(1 + \delta_Q))^2} + \frac{1 - h}{\eta}\left(\|\tilde{\boldsymbol{w}}\|^2 + \frac{\|\boldsymbol{\mu}_\beta\|^2(\tilde{\boldsymbol{w}}^\top \boldsymbol{\mu}_\beta)^2}{(\|\boldsymbol{\mu}_\beta\|^2 + 1 + \gamma(1 + \delta_Q))^2} - \frac{2(\tilde{\boldsymbol{w}}^\top \boldsymbol{\mu}_\beta)^2}{\|\boldsymbol{\mu}_\beta\|^2 + 1 + \gamma(1 + \delta_Q)}\right)\right)$$
$$= \frac{\gamma^2(1 + \delta_Q)^2}{h}\left(\frac{(\tilde{\boldsymbol{w}}^\top \boldsymbol{\mu}_\beta)^2}{\lambda_Q^2} + \frac{1 - h}{\eta}\|\tilde{\boldsymbol{w}}\|^2 + \frac{(1 - h)(\tilde{\boldsymbol{w}}^\top \boldsymbol{\mu}_\beta)^2}{\eta\lambda_Q}\left(\frac{\|\boldsymbol{\mu}_\beta\|^2}{\lambda_Q} - 2\right)\right)$$

Which finally gives the following theorem:

**Theorem C.1** (Gaussianity of the fine-tuned model for an arbitrary $\tilde{w}$). *Let $w_\alpha$ be the fine-tuned classifier as defined in equation $\alpha$-FTC and suppose that Assumption 4.1 holds. The decision function $w_\alpha^\top x$, on some test sample $x \in \mathcal{C}_a$ independent of $\mathbf{X}$, satisfies:*

$$w_\alpha^\top x \xrightarrow{\mathcal{D}} \mathcal{N}\left((-1)^a m_\alpha, \nu_\alpha - m_\alpha^2\right),$$

*where:*

$$m_\alpha = \frac{\|\boldsymbol{\mu}_\beta\|^2 + \alpha\gamma(1+\delta_Q)\langle\tilde{w}, \boldsymbol{\mu}_\beta\rangle}{\|\boldsymbol{\mu}_\beta\|^2 + 1 + \gamma(1+\delta_Q)},$$

$$\nu_\alpha = T_1 + \alpha T_2 + \alpha^2 T_3.$$

*with:*

$$T_1 = \frac{\|\boldsymbol{\mu}_\beta\|^2}{h\lambda_Q}\left(\frac{\|\boldsymbol{\mu}_\beta\|^2 + 1}{\lambda_Q} - 2(1-h)\right) + \frac{1-h}{h},$$

$$T_2 = \frac{2\gamma(1+\delta_Q)\langle\tilde{w}, \boldsymbol{\mu}_\beta\rangle}{h\lambda_Q}\left(\frac{\|\boldsymbol{\mu}_\beta\|^2 + 1}{\lambda} - (1-h)\right),$$

$$T_3 = \frac{\gamma^2(1+\delta_Q)^2}{h}\left(\frac{\langle\tilde{w}, \boldsymbol{\mu}_\beta\rangle^2}{\lambda_Q^2} + \frac{1-h}{\eta}\|\tilde{w}\|^2 + \frac{(1-h)\langle\tilde{w}, \boldsymbol{\mu}_\beta\rangle^2}{\eta\lambda_Q}\left(\frac{\|\boldsymbol{\mu}_\beta\|^2}{\lambda_Q} - 2\right)\right).$$

### C.3 FINDING OPTIMAL $\alpha^*$

Since the test accuracy is given by $\mathcal{A}_{\text{test}} = 1 - \varphi\left((\nu_\alpha - m_\alpha^2)^{-\frac{1}{2}} m_\alpha\right)$ as in Proposition 4.3, and that $\phi(x)$ is a non-increasing function, then finding the optimal $\alpha^*$ that maximizes the test accuracy boils down to maximizing the term inside $\phi$. Thus, by computing the derivative with respect to $\alpha$ of $(\nu_\alpha - m_\alpha^2)^{-\frac{1}{2}} m_\alpha$ and finding the zero of the gradient gives us the final form of the best scaling parameter $\alpha^*$:

$$\alpha^* = \frac{\eta(1 + \gamma(1+\delta_Q))\langle\tilde{w}, \boldsymbol{\mu}_\beta\rangle}{\gamma(1+\delta_Q)\left(\lambda\|\boldsymbol{\mu}_\beta\|^2\|\tilde{w}\|^2 - (\lambda - \eta)\langle\tilde{w}, \boldsymbol{\mu}_\beta\rangle^2\right)}$$

And since the worst test accuracy is 50% (random classification), which is obtained for $m_\alpha = 0$, then solving the previous equation gives the worst scaling $\bar{\alpha}$ to use:

$$\bar{\alpha} = \frac{-\|\boldsymbol{\mu}_\beta\|^2}{\gamma(1+\delta_Q)\langle\tilde{w}, \boldsymbol{\mu}_\beta\rangle}$$

## D   EXTENSION TO MULTI-SOURCE CLASSIFIERS

Given $T$ source classifiers $\{w_t\}_{t=1}^T$ and a single target task, the goal is to fine-tune a mixture of these classifiers on the target task. Specifically, we want to find the optimal fine-tuned classifier $w_\Omega$ that is written as:

$$w_\Omega = \sum_{t=1}^T \alpha_t w_t + a$$

where $\alpha_t \in \mathbb{R}$ and $a$ is an adapter trained on the target dataset as follows:

$$a = \arg\min_v \frac{1}{n}\|\mathbf{X}^\top(\sum_{t=1}^T \alpha_t w_t + v) - y\|^2 + \gamma\|v\|^2$$

Then, $a$ expresses as:

$$a = \frac{1}{n}\left(\frac{1}{n}\mathbf{X}\mathbf{X}^\top + \gamma\mathbf{I}_p\right)^{-1}\left(\mathbf{X}y - \mathbf{X}\mathbf{X}^\top\sum_{t=1}^T \alpha_t w_t\right)$$

Thus, our new fine-tuned classifier writes as:

$$w_\Omega = \sum_{t=1}^T \alpha_t w_t + a = \frac{1}{n}\mathbf{Q}\mathbf{X}y + \gamma\sum_{t=1}^T \alpha_t \mathbf{Q}w_t$$

To compute the theoretical test accuracy of this classifier, we will take a test sample $x \sim \mathcal{N}((-1)^a\boldsymbol{\mu}_\beta, \mathbf{I}_p)$, independent from the training data $(x_i)_{i=1}^n$, and we compute the statistics of the decision function $w_\Omega^\top x$.

### D.1 TEST EXPECTATION

We have that:

$$\mathbb{E}[\boldsymbol{w}_\Omega^\top \boldsymbol{x}] = \mathbb{E}[\boldsymbol{w}^\top \boldsymbol{x}] + \gamma \sum_{t=1}^{T} \alpha_t \, \mathbb{E}[\boldsymbol{w}_t^\top \mathbf{Q} \boldsymbol{x}]$$

$$= \mathbb{E}[\boldsymbol{w}^\top \boldsymbol{x}] + (-1)^a \gamma \sum_{t=1}^{T} \alpha_t \boldsymbol{w}_t^\top \bar{\mathbf{Q}} \boldsymbol{\mu}_\beta$$

From the previous section, we have that:

$$\mathbb{E}[\boldsymbol{w}^\top \boldsymbol{x}] = \frac{(-1)^a}{1 + \delta_Q} \boldsymbol{\mu}_\beta^\top \bar{\mathbf{Q}} \boldsymbol{\mu}_\beta = \frac{(-1)^a \|\boldsymbol{\mu}_\beta\|^2}{\|\boldsymbol{\mu}_\beta\|^2 + 1 + \gamma(1 + \delta_Q)}$$

And from lemma A.5, we have that:

$$\boldsymbol{w}_t^\top \bar{\mathbf{Q}} \boldsymbol{\mu}_\beta = \frac{(1 + \delta_Q)\langle \boldsymbol{w}_t, \boldsymbol{\mu}_\beta \rangle}{\|\boldsymbol{\mu}_\beta\|^2 + 1 + \gamma(1 + \delta_Q)}$$

Finally, we get that:

$$\mathbb{E}[\boldsymbol{w}_\Omega^\top \boldsymbol{x}] = \frac{(-1)^a}{\|\boldsymbol{\mu}_\beta\|^2 + 1 + \gamma(1 + \delta_Q)} \left( \|\boldsymbol{\mu}_\beta\|^2 + \gamma(1 + \delta_Q) \sum_{t=1}^{T} \alpha_t \langle \boldsymbol{w}_t, \boldsymbol{\mu}_\beta \rangle \right)$$

In a vectorized form, denote by $\boldsymbol{\alpha} = (\alpha_1, \ldots, \alpha_T)^\top$ the vector of coefficients and by $\mathbf{W} = (\boldsymbol{w}_1, \ldots, \boldsymbol{w}_T) \in \mathbb{R}^{p \times T}$, then we have that:

$$\mathbb{E}[\boldsymbol{w}_\Omega^\top \boldsymbol{x}] = (-1)^a \frac{\|\boldsymbol{\mu}_\beta\|^2 + \gamma(1 + \delta_Q)\boldsymbol{\alpha}^\top \mathbf{W}^\top \boldsymbol{\mu}_\beta}{\|\boldsymbol{\mu}_\beta\|^2 + 1 + \gamma(1 + \delta_Q)}$$

### D.2 TEST VARIANCE

Now we will compute the expectation of the second order moment of $\boldsymbol{w}_\Omega^\top \boldsymbol{x}$:

$$\mathbb{E}[(\boldsymbol{w}_\Omega^\top \boldsymbol{x})^2] = \mathbb{E}\left[ (\boldsymbol{w}^\top \boldsymbol{x})^2 + \gamma^2 \left( \sum_{t=1}^{T} \alpha_t \boldsymbol{w}_t^\top \mathbf{Q} \boldsymbol{x} \right)^2 + 2\gamma \sum_{t=1}^{T} \alpha_t \boldsymbol{w}_t^\top \mathbf{Q} \boldsymbol{x} \boldsymbol{w}^\top \boldsymbol{x} \right]$$

Let us compute each term of this sum and then aggregate the results at the end.

**First term.** We have that:
$$\mathbb{E}[(\boldsymbol{w}^\top \boldsymbol{x})^2] = \frac{\|\boldsymbol{\mu}_\beta\|^2}{h \lambda_Q} \left( \frac{\|\boldsymbol{\mu}_\beta\|^2 + 1}{\lambda_Q} - 2(1 - h) \right) + \frac{1 - h}{h}$$

**Second term.** Now let us compute the second term of the sum:

$$\mathbb{E}\left[ \sum_{t=1}^{T} \alpha_t \boldsymbol{w}_t^\top \mathbf{Q} \boldsymbol{x} \boldsymbol{w}^\top \boldsymbol{x} \right] = \sum_{t=1}^{T} \alpha_t \, \mathbb{E}[\boldsymbol{w}_t^\top \mathbf{Q} \boldsymbol{x} \boldsymbol{x}^\top \boldsymbol{w}]$$

$$= \sum_{t=1}^{T} \alpha_t \, \mathbb{E}[\boldsymbol{w}_t^\top \mathbf{Q} \Sigma_\beta \boldsymbol{w}]$$

$$= \sum_{t=1}^{T} \alpha_t \boldsymbol{w}_t^\top \, \mathbb{E}[\mathbf{Q} \Sigma_\beta \frac{1}{n} \sum_{i=1}^{n} y_i \mathbf{Q} \boldsymbol{x}_i]$$

$$= \sum_{t=1}^{T} \alpha_t \boldsymbol{w}_t^\top \, \mathbb{E}[\mathbf{Q} \Sigma_\beta \mathbf{Q} y_i \boldsymbol{x}_i] \qquad (\boldsymbol{x}_i \text{ i.i.d})$$

$$= \frac{1}{1 + \delta_Q} \sum_{t=1}^{T} \alpha_t \boldsymbol{w}_t^\top \, \mathbb{E}[\mathbf{Q} \Sigma_\beta \mathbf{Q}_{-i} y_i \boldsymbol{x}_i]$$

And since we have that:

$$\mathbf{Q} = \mathbf{Q}_{-i} - \frac{\mathbf{Q}_{-i}\boldsymbol{x}_i\boldsymbol{x}_i^\top\mathbf{Q}_{-i}}{n(1+\delta_Q)}$$

Then:

$$\mathbb{E}\left[\sum_{t=1}^{T}\alpha_t\boldsymbol{w}_t^\top\mathbf{Q}\boldsymbol{x}\boldsymbol{w}^\top\boldsymbol{x}\right] = \frac{1}{1+\delta_Q}\sum_{t=1}^{T}\alpha_t\boldsymbol{w}_t^\top\,\mathbb{E}\left[\left(\mathbf{Q}_{-i} - \frac{\mathbf{Q}_{-i}\boldsymbol{x}_i\boldsymbol{x}_i^\top\mathbf{Q}_{-i}}{n(1+\delta_Q)}\right)\Sigma_\beta\mathbf{Q}_{-i}y_i\boldsymbol{x}_i\right]$$

$$= \frac{1}{1+\delta_Q}\sum_{t=1}^{T}\alpha_t\boldsymbol{w}_t^\top\,\mathbb{E}[\mathbf{Q}_{-i}\Sigma_\beta\mathbf{Q}_{-i}y_i\boldsymbol{x}_i] - \frac{1}{n(1+\delta_Q)^2}\sum_{t=1}^{T}\alpha_t\boldsymbol{w}_t^\top\,\mathbb{E}[\mathbf{Q}_{-i}\boldsymbol{x}_i\boldsymbol{x}_i^\top\mathbf{Q}_{-i}\Sigma_\beta\mathbf{Q}_{-i}y_i\boldsymbol{x}_i]$$

We have that:

$$\sum_{t=1}^{T}\alpha_t\boldsymbol{w}_t^\top\,\mathbb{E}[\mathbf{Q}_{-i}\Sigma_\beta\mathbf{Q}_{-i}y_i\boldsymbol{x}_i] = \sum_{t=1}^{T}\alpha_t\boldsymbol{w}_t^\top\,\mathbb{E}[\mathbf{Q}\Sigma_\beta\mathbf{Q}]\boldsymbol{\mu}_\beta$$

$$= \frac{1}{h}\sum_{t=1}^{T}\alpha_t\boldsymbol{w}_t^\top\bar{\mathbf{Q}}\Sigma_\beta\bar{\mathbf{Q}}\boldsymbol{\mu}_\beta$$

$$= \frac{1}{h}\sum_{t=1}^{T}\alpha_t\frac{(1+\delta_Q)^2}{\lambda_Q^2}\langle\boldsymbol{w}_t,\boldsymbol{\mu}_\beta\rangle\left(\|\boldsymbol{\mu}_\beta\|^2+1\right)$$

And we have that:

$$\frac{1}{n(1+\delta_Q)^2}\sum_{t=1}^{T}\alpha_t\boldsymbol{w}_t^\top\,\mathbb{E}[\mathbf{Q}_{-i}\boldsymbol{x}_i\boldsymbol{x}_i^\top\mathbf{Q}_{-i}\Sigma_\beta\mathbf{Q}_{-i}y_i\boldsymbol{x}_i] = \frac{1}{n(1+\delta_Q)^2}\sum_{t=1}^{T}\alpha_t\boldsymbol{w}_t^\top\,\mathbb{E}[\mathbf{Q}_{-i}y_i\boldsymbol{x}_i\,\mathrm{Tr}(\boldsymbol{x}_i\boldsymbol{x}_i^\top\mathbf{Q}_{-i}\Sigma_\beta\mathbf{Q}_{-i})]$$

$$= \frac{1}{n(1+\delta_Q)^2}\sum_{t=1}^{T}\alpha_t\boldsymbol{w}_t^\top\,\mathbb{E}[\mathbf{Q}_{-i}y_i\boldsymbol{x}_i\,\mathrm{Tr}(\Sigma_\beta\,\mathbb{E}[\mathbf{Q}\Sigma_\beta\mathbf{Q}])]$$

$$= \frac{1}{n(1+\delta_Q)^2}\sum_{t=1}^{T}\alpha_t\boldsymbol{w}_t^\top\,\mathbb{E}[\mathbf{Q}_{-i}y_i\boldsymbol{x}_i]\frac{1}{h}\,\mathrm{Tr}((\Sigma_\beta\bar{\mathbf{Q}})^2)$$

$$= \frac{1-h}{h}\sum_{t=1}^{T}\alpha_t\boldsymbol{w}_t^\top\bar{\mathbf{Q}}\boldsymbol{\mu}_\beta$$

$$= \frac{1-h}{h}\sum_{t=1}^{T}\alpha_t\frac{(1+\delta_Q)\langle\boldsymbol{w}_t,\boldsymbol{\mu}_\beta\rangle}{\lambda_Q}$$

Thus the second term is given by:

$$\mathbb{E}\left[\sum_{t=1}^{T}\alpha_t\boldsymbol{w}_t^\top\mathbf{Q}\boldsymbol{x}\boldsymbol{w}^\top\boldsymbol{x}\right] = \frac{(1+\delta_Q)}{h\lambda_Q}\sum_{t=1}^{T}\alpha_t\left(\frac{\|\boldsymbol{\mu}_\beta\|^2+1}{\lambda_Q} - (1-h)\right)\langle\boldsymbol{w}_t,\boldsymbol{\mu}_\beta\rangle$$

$$= \frac{(1+\delta_Q)}{h\lambda_Q}\left(\frac{\|\boldsymbol{\mu}_\beta\|^2+1}{\lambda_Q} - (1-h)\right)\boldsymbol{\alpha}^\top\mathbf{W}^\top\boldsymbol{\mu}_\beta$$

**Third term.** We have that:

$$\gamma^2\,\mathbb{E}\left[\left(\sum_{t=1}^{T}\alpha_t\boldsymbol{w}_t^\top\mathbf{Q}\boldsymbol{x}\right)^2\right]=\gamma^2\,\mathbb{E}\left[\sum_{t=1}^{T}\alpha_t\boldsymbol{w}_t^\top\mathbf{Q}\boldsymbol{x}\sum_{k=1}^{T}\alpha_k\boldsymbol{w}_k^\top\mathbf{Q}\boldsymbol{x}\right]$$

$$=\gamma^2\sum_{t,k=1}^{T}\mathbb{E}[\alpha_t\alpha_k\boldsymbol{w}_t^\top\mathbf{Q}\boldsymbol{x}\boldsymbol{x}^\top\mathbf{Q}\boldsymbol{w}_k]$$

$$=\gamma^2\sum_{t,k=1}^{T}\mathbb{E}[\boldsymbol{w}_t^\top\mathbf{Q}\Sigma_\beta\mathbf{Q}\boldsymbol{w}_k]$$

$$=\gamma^2\sum_{t,k=1}^{T}\boldsymbol{w}_t^\top\,\mathbb{E}[\mathbf{Q}\Sigma_\beta\mathbf{Q}]\boldsymbol{w}_k$$

$$=\frac{\gamma^2}{h}\sum_{t,k=1}^{T}\alpha_t\alpha_k\boldsymbol{w}_t^\top\bar{\mathbf{Q}}\Sigma_\beta\bar{\mathbf{Q}}\boldsymbol{w}_k$$

And we have that:

$$\bar{\mathbf{Q}}\Sigma_\beta\bar{\mathbf{Q}}=\bar{\mathbf{Q}}\left(\boldsymbol{\mu}_\beta\boldsymbol{\mu}_\beta^\top+\mathbf{I}_p\right)\bar{\mathbf{Q}}$$

$$=\bar{\mathbf{Q}}\boldsymbol{\mu}_\beta\boldsymbol{\mu}_\beta^\top\bar{\mathbf{Q}}+\bar{\mathbf{Q}}^2$$

$$=\frac{(1+\delta_Q)^2}{\lambda_Q^2}\boldsymbol{\mu}_\beta\boldsymbol{\mu}_\beta^\top+\frac{(1+\delta_Q)^2}{(1+\gamma(1+\delta_Q))^2}\left(\mathbf{I}_p+\frac{(\boldsymbol{\mu}_\beta\boldsymbol{\mu}_\beta^\top)^2}{\lambda_Q^2}-\frac{2\boldsymbol{\mu}_\beta\boldsymbol{\mu}_\beta^\top}{\lambda_Q}\right)$$

Thus the last term is given by:

$$\gamma^2\,\mathbb{E}\left[\left(\sum_{t=1}^{T}\alpha_t\boldsymbol{w}_t^\top\mathbf{Q}\boldsymbol{x}\right)^2\right]=\frac{\gamma^2(1+\delta_Q)^2}{h}\times$$

$$\sum_{t,k=1}^{T}\alpha_t\alpha_k\left[\frac{\langle\boldsymbol{w}_t,\boldsymbol{\mu}_\beta\rangle\langle\boldsymbol{w}_k,\boldsymbol{\mu}_\beta\rangle}{\lambda_Q^2}+\frac{1}{(1+\gamma(1+\delta_Q))^2}\left(\langle\boldsymbol{w}_t,\boldsymbol{w}_k\rangle+\frac{\|\boldsymbol{\mu}_\beta\|^2\langle\boldsymbol{w}_t,\boldsymbol{\mu}_\beta\rangle\langle\boldsymbol{w}_k,\boldsymbol{\mu}_\beta\rangle}{\lambda_Q^2}-\frac{2\langle\boldsymbol{w}_t,\boldsymbol{\mu}_\beta\rangle\langle\boldsymbol{w}_k,\boldsymbol{\mu}_\beta\rangle}{\lambda_Q}\right)\right]$$

In a vectorized form, we have that:

$$\gamma^2\,\mathbb{E}\left[\left(\sum_{t=1}^{T}\alpha_t\boldsymbol{w}_t^\top\mathbf{Q}\boldsymbol{x}\right)^2\right]=\frac{\gamma^2(1+\delta_Q)^2}{h}\times$$

$$\left[\frac{(\boldsymbol{\alpha}^\top\mathbf{W}^\top\boldsymbol{\mu}_\beta)^2}{\lambda_Q^2}+\frac{1}{(1+\gamma(1+\delta_Q))^2}\left(\boldsymbol{\alpha}^\top\mathbf{W}^\top\mathbf{W}\boldsymbol{\alpha}+\frac{\|\boldsymbol{\mu}_\beta\|^2(\boldsymbol{\alpha}^\top\mathbf{W}^\top\boldsymbol{\mu}_\beta)^2}{\lambda_Q^2}-\frac{2(\boldsymbol{\alpha}^\top\mathbf{W}^\top\boldsymbol{\mu}_\beta)^2}{\lambda_Q}\right)\right]$$

$$=\frac{\gamma^2(1+\delta_Q)^2}{h}\boldsymbol{\alpha}^\top\mathbf{M}\boldsymbol{\alpha}$$

where:

$$\mathbf{M}=\frac{(1-h)}{\eta}\mathbf{W}^\top\mathbf{W}+\left(\frac{1}{\lambda_Q^2}+\frac{(1-h)}{\eta\lambda_Q}\left(\frac{\|\boldsymbol{\mu}_\beta\|^2}{\lambda_Q}-2\right)\right)\mathbf{W}^\top\boldsymbol{\mu}_\beta\boldsymbol{\mu}_\beta^\top\mathbf{W}^\top$$

Finally gives us the expression of the second order moment of $\boldsymbol{w}_\Omega^\top\boldsymbol{x}$ as follows:

$$\mathbb{E}[(\boldsymbol{w}_\Omega^\top\boldsymbol{x})^2]=T_1+T_2+T_3$$

where:

$$T_1 = \frac{\|\boldsymbol{\mu}_\beta\|^2}{h\lambda_Q}\left(\frac{\|\boldsymbol{\mu}_\beta\|^2 + 1}{\lambda_Q} - 2(1-h)\right) + \frac{1-h}{h}$$

$$T_2 = \frac{2\gamma(1+\delta_Q)}{h\lambda_Q}\sum_{t=1}^{T}\alpha_t\left(\frac{\|\boldsymbol{\mu}_\beta\|^2 + 1}{\lambda_Q} - (1-h)\right)\langle\boldsymbol{w}_t, \boldsymbol{\mu}_\beta\rangle$$

$$T_3 = \frac{\gamma^2(1+\delta_Q)^2}{h}\times$$

$$\sum_{t,k=1}^{T}\alpha_t\alpha_k\left[\frac{\langle\boldsymbol{w}_t,\boldsymbol{\mu}_\beta\rangle\langle\boldsymbol{w}_k,\boldsymbol{\mu}_\beta\rangle}{\lambda_Q^2} + \frac{1}{(1+\gamma(1+\delta_Q))^2}\left(\langle\boldsymbol{w}_t,\boldsymbol{w}_k\rangle + \frac{\|\boldsymbol{\mu}_\beta\|^2\langle\boldsymbol{w}_t,\boldsymbol{\mu}_\beta\rangle\langle\boldsymbol{w}_k,\boldsymbol{\mu}_\beta\rangle}{\lambda_Q^2} - \frac{2\langle\boldsymbol{w}_t,\boldsymbol{\mu}_\beta\rangle\langle\boldsymbol{w}_k,\boldsymbol{\mu}_\beta\rangle}{\lambda_Q}\right)\right]$$

Which also writes in a vectorized form:

$$T_1 = \frac{\|\boldsymbol{\mu}_\beta\|^2}{h\lambda_Q}\left(\frac{\|\boldsymbol{\mu}_\beta\|^2 + 1}{\lambda_Q} - 2(1-h)\right) + \frac{1-h}{h}$$

$$T_2 = \frac{2\gamma(1+\delta_Q)}{h\lambda_Q}\left(\frac{\|\boldsymbol{\mu}_\beta\|^2 + 1}{\lambda_Q} - (1-h)\right)\boldsymbol{\alpha}^\top\mathbf{W}^\top\boldsymbol{\mu}_\beta$$

$$T_3 = \frac{\gamma^2(1+\delta_Q)^2}{h}\boldsymbol{\alpha}^\top\mathbf{M}\boldsymbol{\alpha}$$

### D.3 FINDING OPTIMAL $\alpha$

The theoretical test accuracy writes as follows:

$$\mathcal{A}_{\text{test}}(\boldsymbol{\alpha}) = \varphi\left(\frac{a_1 + \boldsymbol{\alpha}^\top\boldsymbol{v}_1}{\sqrt{a_2 + \boldsymbol{\alpha}^\top\boldsymbol{v}_2 + \boldsymbol{\alpha}^\top\tilde{\mathbf{M}}\boldsymbol{\alpha}}}\right)$$

where:

$$a_1 = \frac{\|\boldsymbol{\mu}_\beta\|^2}{\lambda_Q}, \quad \boldsymbol{v}_1 = \frac{\gamma(1+\delta_Q)}{\lambda_Q}\mathbf{W}^\top\boldsymbol{\mu}_\beta, \quad a_2 = T_1 - a_1^2 = \frac{\|\boldsymbol{\mu}_\beta\|^2}{\lambda_Q}\left(\frac{\|\boldsymbol{\mu}_\beta\|^2 + 1}{h\lambda_Q} - \frac{\|\boldsymbol{\mu}_\beta\|^2}{\lambda_Q} - \frac{2(1-h)}{h}\right) + \frac{1-h}{h}$$

$$\boldsymbol{v}_2 = \frac{(1+\delta_Q)}{\lambda_Q}\left(\frac{\|\boldsymbol{\mu}_\beta\|^2 + 1}{h\lambda_Q} - \frac{2\gamma\|\boldsymbol{\mu}_\beta\|^2}{\lambda_Q} - \frac{1-h}{h}\right)\mathbf{W}^\top\boldsymbol{\mu}_\beta,$$

$$\tilde{M} = \frac{\gamma^2(1+\delta_Q)^2(1-h)}{h}\left(\frac{1}{\eta}\mathbf{W}^\top\mathbf{W} + \left(\frac{1}{\lambda_Q^2} + \frac{1}{\eta\lambda_Q}\left(\frac{\|\boldsymbol{\mu}_\beta\|^2}{\lambda_Q} - 2\right)\right)\mathbf{W}^\top\boldsymbol{\mu}_\beta\boldsymbol{\mu}_\beta^\top\mathbf{W}^\top\right)$$

And therefore, since $\varphi$ is non-decreasing, maximizing this test accuracy boils down to maximizing the term inside it, i.e we want to find $\boldsymbol{\alpha}^*$ that satisfies:

$$\boldsymbol{\alpha}^* \in \arg\max_{\boldsymbol{\alpha}}\frac{a_1 + \boldsymbol{\alpha}^\top\boldsymbol{v}_1}{\sqrt{a_2 + \boldsymbol{\alpha}^\top\boldsymbol{v}_2 + \boldsymbol{\alpha}^\top\tilde{M}\boldsymbol{\alpha}}} = \arg\max_{\boldsymbol{\alpha}} g(\boldsymbol{\alpha})$$

We compute the gradient of $g$ with respect to $\boldsymbol{\alpha}$ to find the extremum values of these mixing parameters:

$$\nabla_{\boldsymbol{\alpha}}g(\boldsymbol{\alpha}) = \frac{\sqrt{a_2 + \boldsymbol{\alpha}^\top\boldsymbol{v}_2 + \boldsymbol{\alpha}\tilde{\mathbf{M}}\boldsymbol{\alpha}}\ \boldsymbol{v}_1 - (a_1 + \boldsymbol{\alpha}^\top\boldsymbol{v}_1)\frac{\boldsymbol{v}_2 + 2\tilde{\mathbf{M}}\boldsymbol{\alpha}}{\sqrt{a_2 + \boldsymbol{\alpha}^\top\boldsymbol{v}_2 + \boldsymbol{\alpha}^\top\tilde{\mathbf{M}}\boldsymbol{\alpha}}}}{a_2 + \boldsymbol{\alpha}^\top\boldsymbol{v}_2 + \boldsymbol{\alpha}^\top\tilde{\mathbf{M}}\boldsymbol{\alpha}}$$

Thus the roots $\boldsymbol{\alpha}$ of $\nabla g(\boldsymbol{\alpha})$ satisfy the following equation:

$$\boxed{(a_2 + \boldsymbol{\alpha}^\top\boldsymbol{v}_2 + \boldsymbol{\alpha}^\top\tilde{\mathbf{M}}\boldsymbol{\alpha})\boldsymbol{v}_1 - (a_1 + \boldsymbol{\alpha}^\top\boldsymbol{v}_1)(\boldsymbol{v}_2 + 2\tilde{\mathbf{M}}\boldsymbol{\alpha}) = 0}$$

## E LLMS EXPERIMENTAL DETAILS

The pseudo-code algorithm for training with $\alpha$-LoRA is given as follows in 1.

---

**Algorithm 1** $\alpha$-LORA FINE-TUNING

---

**Require:** Base model weights $\{\mathbf{W}_i^*\}_{i=1}^N$, fine-tuning dataset $\mathcal{D} = \{B_j\}_{j=1}^b$ divided into batches, update period $T$, optimizers `optim` (for LoRA modules) and `optim_alpha` (for $\boldsymbol{\alpha} = \{\alpha_i\}_{i=1}^N$), number of epochs $n$.

1: **for** $k = 1 \ldots n$ **do**
2:    **for** batch $B_j$ in $\mathcal{D}$ **do**
3:       Update LoRA modules $\{(A_i, B_i)\}_{i=1}^N$ with a gradient step on $B$ using `optim`.
4:       **if** $j \bmod T = 0$ **then**
5:          Sample a fresh batch $B_\alpha$ from $\mathcal{D}$
6:          Update $\boldsymbol{\alpha}$ with a gradient step on $B_\alpha$ using `optim_alpha`.
7:       **end if**
8:    **end for**
9: **end for**

---

### E.1 HYPERPARAMETERS

In this section, we summarize all the details about our experiments on Fine-tuning `roberta-base` model on GLUE tasks. Let us define some notations first then give their corresponding values in each experiment: `lora_r` denotes the rank of LoRA modules, `lora_alpha` denotes the LoRA scaling parameter, `lr_adapter` means the learning rate used to train LoRA modules, `batch_size` and `batch_alpha` is the training batch size for LoRA modules and the vectors $\boldsymbol{\alpha}$ respectively, `lr_alpha` is the learning rate used to update $\boldsymbol{\alpha}$, `optim_alpha` is the optimizer used to train the vectors $\boldsymbol{\alpha}$, `val_split` is the percentage of the training set used to train $\boldsymbol{\alpha}$.

**Common to all experiments.** We optimize the LoRA modules using `AdamW` for all the benchmarks and with a linear scheduler for the learning rate. We initialize the vectors $\boldsymbol{\alpha}$ to the vector **1**. The target modules are: the final classifier layer `classifier` (full training) and the attention modules `query` and `value` (Low Rank Adaptation).

| Parameter | Value |
|---|---|
| `optimizer` | AdamW |
| **LoRA Arguments** | |
| `lora_r` | 8 |
| `lora_alpha` | 8 |
| `lr_adapter` | $10^{-4}$ |
| **Trainer Arguments** | |
| `n_epochs` | 10 |
| `batch_size` | 64 |
| `optim_alpha` | AdamW |
| `batch_alpha` | 64 |
| `lr_alpha` | $10^{-2}$ |
| `T` | 1 |
| `val_split` | 1 |
| `seeds` | 1, 5, 123 |

Table 3: Implementation Details for the fine-tuning experiment on MNLI.

### E.2 VALUES OF $\alpha$

We report in the following plots some metrics (mean, standard deviation, percentiles) describing the obtained values of the vectors $\boldsymbol{\alpha}$ for each module after the training phase.

| Parameter | Value |
|---|---|
| optimizer | AdamW |
| **LoRA Arguments** | |
| lora_r | 8 |
| lora_alpha | 8 |
| lr_adapter | $10^{-4}$ for LoRA and $2.10^{-4}$ for $\alpha$-LoRA |
| **Trainer Arguments** | |
| n_epochs | 10 |
| batch_size | 64 |
| optim_alpha | Adam |
| batch_alpha | 64 |
| lr_alpha | $5.10^{-3}$ |
| T | 20 |
| val_split | 0.2 |
| seeds | 1, 3, 123 |

Table 4: Implementation Details for the fine-tuning experiment on QNLI.

| Parameter | Value |
|---|---|
| optimizer | AdamW |
| **LoRA Arguments** | |
| lora_r | 8 |
| lora_alpha | 8 |
| lr_adapter | $10^{-4}$ for LoRA and $2.10^{-4}$ for $\alpha$-LoRA |
| **Trainer Arguments** | |
| n_epochs | 40 |
| batch_size | 64 |
| optim_alpha | Adam |
| batch_alpha | 64 |
| lr_alpha | $5.10^{-3}$ |
| T | 20 |
| val_split | 0.2 |
| seeds | 3, 5, 123 |

Table 5: Implementation Details for the fine-tuning experiment on MRPC.

| Parameter | Value |
|---|---|
| optimizer | AdamW |
| **LoRA Arguments** | |
| lora_r | 8 |
| lora_alpha | 8 |
| lr_adapter | $10^{-4}$ |
| **Trainer Arguments** | |
| n_epochs | 40 |
| batch_size | 64 |
| optim_alpha | AdamW |
| batch_alpha | 64 |
| lr_alpha | $5.10^{-3}$ |
| T | 20 |
| val_split | 0.8 (and 0.2 for seed 123) |
| seeds | 3, 5, 123 |

Table 6: Implementation Details for the fine-tuning experiment on RTE.

| Parameter | Value |
|---|---|
| optimizer | AdamW |
| **LoRA Arguments** | |
| lora_r | 8 |
| lora_alpha | 8 |
| lr_adapter | $10^{-4}$ for LoRA and $2.10^{-4}$ for $\alpha$-LoRA |
| **Trainer Arguments** | |
| n_epochs | 10 |
| batch_size | 128 |
| optim_alpha | AdamW |
| batch_alpha | 128 |
| lr_alpha | $5.10^{-3}$ |
| T | 10 (and 20 for seed 5) |
| val_split | 0.5 (and 0.9 for seed 5) |
| seeds | 1, 3, 5 |

Table 7: Implementation Details for the fine-tuning experiment on SST2.

| Parameter | Value |
|---|---|
| optimizer | AdamW |
| **LoRA Arguments** | |
| lora_r | 8 |
| lora_alpha | 8 |
| lr_adapter | $5.10^{-4}$ |
| **Trainer Arguments** | |
| n_epochs | 5 |
| batch_size | 256 |
| optim_alpha | Adam, AdamW (seed 123) |
| batch_alpha | 64 |
| lr_alpha | $5.10^{-3}$ |
| T | 1 (seed 3), 10 (seed 5) and 20 (seed 123) |
| val_split | 0.8 |
| seeds | 3, 5, 123 |

Table 8: Implementation Details for the fine-tuning experiment on QQP.

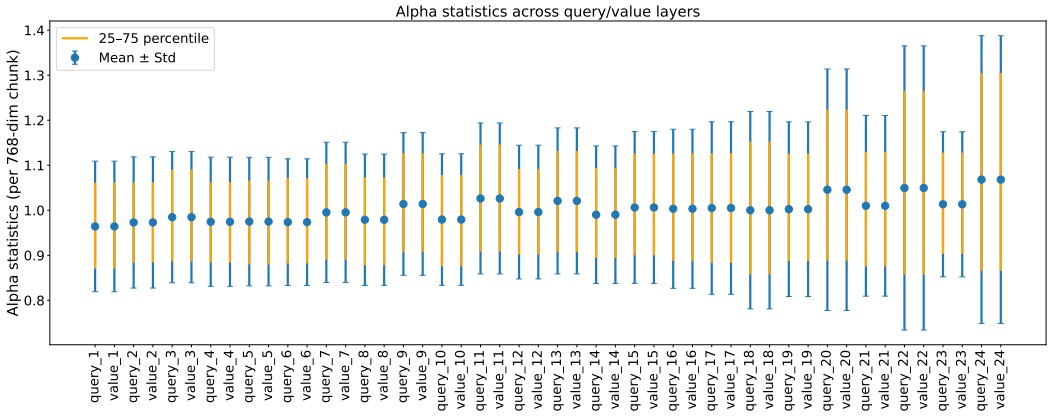

Figure 7: Statistics of the vectors $\alpha$ for the MNLI benchmark

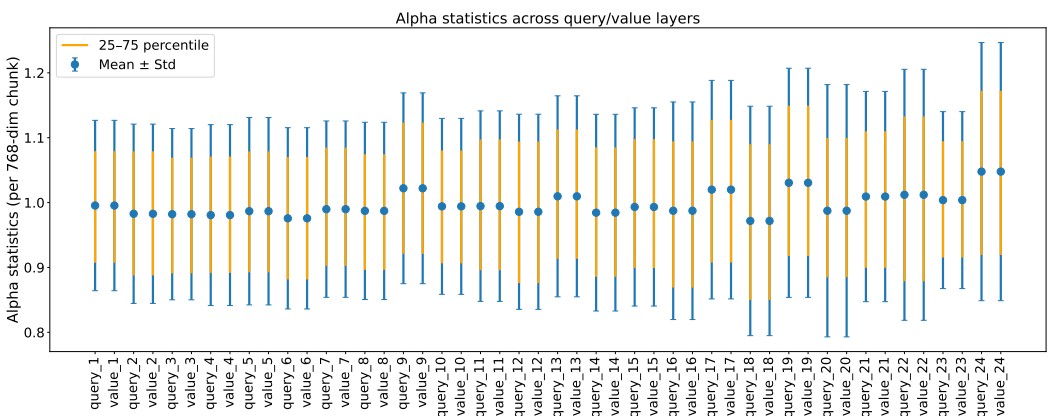

Figure 8: Statistics of the vectors $\boldsymbol{\alpha}$ for the QNLI benchmark

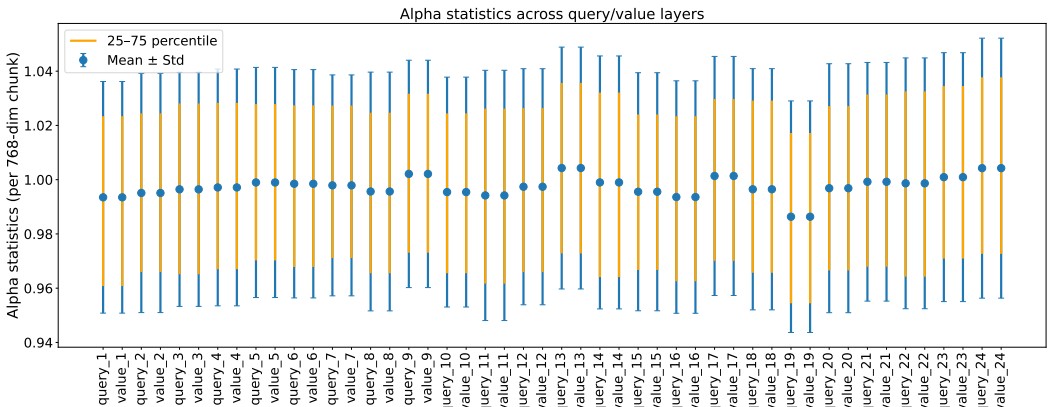

Figure 9: Statistics of the vectors $\boldsymbol{\alpha}$ for the RTE benchmark

