# OpenReview forum: "$\alpha$-LoRA: Effective Fine-Tuning via Base Model Rescaling"
_ICLR.cc/2026/Conference — Submitted to ICLR 2026_

### Official Review · Reviewer_pnWA · 2025-10-16

**Soundness:** 3
**Presentation:** 2
**Contribution:** 2
**Rating:** 2
**Confidence:** 3

**Summary:**

The paper proposes $\alpha$-LoRA, a reparameterization family that rescales each row of a frozen base weight matrix by a learned factor α when applying a trainable (optionally low-rank) update. The authors analyze this idea in a high-dimensional linear binary classification model using Random Matrix Theory (RMT) and derive a  closed-form optimal scaling $\alpha$ that maximizes asymptotic accuracy. They validate the theory on Amazon Reviews and demonstrate consistent improvements over vanilla LoRA on GLUE tasks (RoBERTa-base).

**Strengths:**

1. **Clear theoretical contribution with closed-form solution**.  The  paper gives a unique closed-form $ \alpha^*$ , giving guidance beyond “set $\alpha=1$” in fune-tuning setting.
2. **Practical instantiation for LLMs.** Extending $\alpha$ to a per-row vector aligns well with layered architectures; the formulation is simple to implement and novel in PEFT.

**Weaknesses:**

I am not an expert in theoretical analysis, so I am unable to rigorously verify the correctness of the authors’ proofs. The following comments are therefore made from a more practical and empirical perspective.

1. **Scope of theory.** The RMT analysis is confined to one-layer linear (ridge) classification under a Gaussian mixture with identity covariance. While the paper is transparent about this, the gap to two-layer (or multi-layer) nonlinear networks remains large, so the theory’s prescriptive power for LLMs is still indirect.
2. **Connection to low-rank structure is not formalized.** The core theory is agnostic to LoRA’s low-rank reparameterization. $\alpha$ -rescaling could be  applied to many PEFT variants. The paper positions $\alpha$ -scaling as complementary, but does not analyze interactions with rank choice or decomposition schemes, which would strengthen the story.
3. The idea of rescaling pre-trained channels is established in prior literature. Notably, IA³ fine-tunes LLMs by learning multiplicative factors on intermediate activations within linear transformations—an operation that is effectively equivalent to scaling the output channels of the associated weight matrices  𝑊. This close correspondence substantially diminishes the methodological novelty.
4. **Sample efficiency** The $\alpha$-update uses fresh mini-batches separate from those used to update the adapters. This design may trade sample efficiency for generalization.
4. **Experimental concerns**:

- **Baselines.** Since the paper mentions complementary methods (e.g., DoRA), including such baselines would better position $\alpha$-LoRA beyond vanilla LoRA
- **Hyperparameters.** The appendix shows different adapter LRs and LoRA vs. $ \alpha$-LoRA on several tasks, which can favor $\alpha$-LoRA. The authors even uses different $T$ or ``val split`` for different seed on RTE, SST2 and QQP. While I appreciate the authors’ transparency, the need for such delicate hyperparameter tuning raises concerns about the method’s robustness and practical effectiveness.
- **Limited scale of empirical evaluation**. The experiments are conducted on relatively small datasets and modest-sized models. The experiments would be more convincing if the authors could include evaluations on larger-scale benchmarks (e.g., AlpacaEval) and more capable models such as LLaMA 2-7B/13B.

**Questions:**

See weakness.

---

> ### Author Response · Authors · 2025-11-16
> **Responses to Reviewer pnWA**
>
> We thank the reviewer for their constructive remarks, and below we address all of the raised concerns.
> - **1. Scope of the theory:** We thank the reviewer for this interesting remark. In fact, the goal from our theoretical section is to prove the existence of an optimal scaling parameter $\alpha^*$ that is not necessarily equal to $1$, and this is why we did not consider more complex settings (such as two layer NNs) as existing literature in these latter are very limited and would not help us reach our goals from this work.
> - **2. Connection to low-rank structure:** We agree with the reviewer that we did not explicitly consider a low rank structure of the adapter weights in our theory, which is because our method does not only apply to LoRA, but to any reparameterization-based fine-tuning method that works by adding a trainable module to the existing base model. In practice, the Low Rank adapter $BA $ is *simply a practical instantiation* of the additive matrix $W$ in the theory (Eq. (10)). The key takeaway from our paper is that the *frozen pretrained weights* have a nontrivial optimal scale for target test performance, and this holds regardless of the specific low-rank decomposition used for $W$. We apologize for this confusion.
> - **3. Difference with IA$^3$:** We thank the reviewer for raising this point. In fact, $\alpha$-LoRA and IA$^3$ differ in *where* the multiplicative factors act:
> *IA³:* scales activations (intermediate representations), i.e., $h \mapsto \lambda \odot h$.
> *$\alpha$-LoRA:* rescales the *pretrained weights themselves*, i.e., $W^\star \mapsto \alpha \odot W^*$.
> 	This distinction matters because:
> 1. Scaling activations modifies the input to the layer, not the pretrained directions encoded in $W^* $.
> 2. Rescaling pretrained weights modifies the relative importance of pretrained features in the final representation, which is precisely the trade-off captured by the RMT analysis.
> 3. The theory proves that this specific weight-level rescaling has an optimal value α\*, whereas IA³’s activation scale does not enjoy such an analysis.
>
> Thus $\alpha$-LoRA is motivated by and grounded in a theoretical characterization of the pretrained-vs-target tradeoff at the *weight level*, which is different from activation-level gating.
> - **4. Sample efficiency:** Indeed, there is a *trade-off* between sample reuse and $\alpha$ overfitting. In early experiments, when $\alpha$ was updated on the exact same mini-batches as the LoRA weights, we observed instability and overfitting ($\alpha$ quickly collapsed to extreme values). Using separate batches (Algorithm 1) provides a simple and robust solution.
> - **5. Experimental concerns:**
> - **Baselines:** We agree that such comparisons would be valuable. However, our emphasis in this work is more conceptual, which consists on:
> - Identifying that frozen-weight scaling is a neglected degree of freedom.
> - Providing theoretical justification for why $\alpha^*$ is not necessarily equals to $1$.
> - Showing that $\alpha$-LoRA improves consistently over a strong, commonly used baseline (LoRA).
> Also, the existing extensions of LoRA like LoRA+, AdaLoRA, etc. modify the desired module very differently from our method and are therefore **independent** improvements of LoRA. However, our method is more general and goes even beyond the scope of Low Rank Adaptation (see Eq. (10)) and can be applied together with any reparameterization-based fine-tuning method. Additionally, our method can be combined with LoRA exntesions as we have stated at the end of our conclusion section.
> - **Hyperparameters:** As any experiment in Deep Learning, a proper hyperparameter tuning is necessary for fair and accurate comparison of the results. That is why we have tuned each method fairly for its best performance, and taking other hyperparameters (rather than the ones proposed in the appendix) will lead to lower performance in the task.
> - **Limited scale of empirical evaluations:** We agree that extending to larger generative models is an important next step. The goal of this paper, however, is to introduce and analyze the $\alpha$-scaling principle, not to claim SOTA performance on LLMs. Still, $\alpha$-LoRA has a fundamental property that makes its scaling straightforward which is having a neglectible parameter overhead (less than $0.02$ % in our experiments) even for 7B–70B models.
>
> We hope that we have addressed all the concerns of the reviewer, and we would appreciate further engagement with our work. We also thank the reviewer for his honesty about his non-expertise in the theory, however we want to point out that our theoretical section is our major contribution from this work.

---

> > ### Comment · Reviewer_pnWA · 2025-11-22
> >
> > Thank you for the response. However, I still have the following concerns:
> >
> > >However, our emphasis in this work is more conceptual.
> >
> > > we want to point out that our theoretical section is our major contribution from this work
> >
> > >We agree with the reviewer that we did not explicitly consider a low rank structure of the adapter weights in our theory, which is because our method does not only apply to LoRA, but to any reparameterization-based fine-tuning method that works by adding a trainable module to the existing base model.
> >
> > As I mentioned before, I am not an expert in theory. However, the main method and experimental sections of the paper are centered around LoRA, and even the title explicitly highlights LoRA. This will naturally attract a readership of LoRA practitioners, so my evaluation is primarily from a practical perspective. From that viewpoint, I do not think the rebuttal has resolved my concerns, specifically:
> >
> > > Scaling activations modifies the input to the layer, not the pretrained directions encoded in $W^*$;  Rescaling pretrained weights modifies the relative importance of pretrained features in the final representation, which is precisely the trade-off captured by the RMT analysis.
> >
> > Conceptually, both $\alpha$-LoRA and IA$^3$ involve channel-wise scaling. Therefore, I believe that a comparison with IA$^3$ is necessary in order to validate the practical effectiveness of the proposed method.
> >
> > >  Indeed, there is a _trade-off_  between sample reuse and  α overfitting.
> >
> > This trade-off is introduced by $\alpha$-LoRA itself, and the rebuttal does not convincingly address how this issue is mitigated in practice.
> >
> >
> > > As any experiment in Deep Learning, a proper hyperparameter tuning is necessary for fair and accurate comparison of the results. That is why we have tuned each method fairly for its best performance, and taking other hyperparameters (rather than the ones proposed in the appendix) will lead to lower performance in the task.
> >
> >
> > Since the authors aim to use experiments to further support their theoretical analysis, fair comparisons with baselines are essential. While hyperparameter tuning is indeed necessary in deep learning, using different hyperparameters for different random seeds is, in my view, not acceptable for a fair comparison. Adapting the hyperparameters to each seed mainly shows that the proposed method is highly sensitive to hyperparameter choices. Given the remaining lack of comparisons with stronger baseline methods and the absence of larger-scale experimental results, I am not convinced that the method works reliably in realistic applications.
> >
> > As previously mentioned, and as other reviewers have also pointed out, the theoretical model is quite idealized and makes several strong assumptions, so **it is difficult to see clear practical guidance for real-world fine-tuning scenarios at this stage**.
> >
> > In summary, I will keep my overall score but lower my confidence, in order to avoid making an overly strong or potentially unfair judgment about the theoretical contribution of the paper.

---

### Official Review · Reviewer_ZJJj · 2025-10-27

**Soundness:** 4
**Presentation:** 4
**Contribution:** 2
**Rating:** 4
**Confidence:** 3

**Summary:**

This paper proposes $\alpha$-LoRA, a PEFT variant that rescales the frozen base weights row-wise by a vector $\alpha$ before adding a (low-rank) adapter, i.e., for a layer with frozen $W^\star \in \mathbb{R}^{d_{\text{out}}\times d_{\text{in}}}$ the update is
$$
W_{\text{new}} = \alpha \odot W^\star + W,\quad W =  AB \text{ (LoRA) },
$$
where $\alpha\in\mathbb{R}^{d_{\text{out}}}$ and $\odot$ is row-wise scaling. Most of the paper analyzes a **high-dimensional linear binary classifier with squared loss** under a Gaussian mixture: pre-training data are drawn from $\tfrac12\mathcal{N}(\mu,I_p)+\tfrac12\mathcal{N}(-\mu,I_p)$, and **fine-tuning** targets a shifted mixture with class means $\pm\mu_\beta$, where $\mu_\beta=\beta\mu+\mu_\perp$ and $\mu_\perp\perp\mu$ (identity covariance in both phases). Denoting the ridge solution on target data by $w=\tfrac1nQXy$ and the pre-trained weights by $\tilde w$, the **$\alpha$-fine-tuned** classifier has the closed form
$$
w_\alpha = w + \alpha \gamma Q\tilde{w},
$$
and the authors derive an **asymptotically optimal scalar $\alpha^\star\neq 1$** (closed-form via deterministic equivalents) that maximizes target accuracy in this GMM setting. They then **extend from a scalar to a per-row vector $\alpha$ for neural nets** and give a practical heuristic: treat each $\alpha$ as trainable and update it periodically (every $T$ steps) with a separate optimizer/batch while training LoRA adapters. Empirically, they report that choosing/learning $\alpha$ improves transfer on Amazon Reviews category shifts (where the theory’s $\alpha^\star$ is competitive) and yields consistent gains over vanilla LoRA on GLUE with **roberta-base**. Overall, the work argues that **decoupling magnitude via base-weight rescaling** is a simple, general lever that improves generalization in fine-tuning.

Generally, the paper has a good theoretical contribution, but the experimental evaluation is limited, and there is a gap between the theoretical setting and the LLM fine-tuning setting, as well as a gap between the theoretically analyzed method and the method used in practice.

**Strengths:**

- **(S1)** In the linear GMM setting, the analysis is careful and culminates in a closed-form $\alpha^*$ that depends on data-dependent scalars (via RMT). Figures show how the best α varies with alignment β and dimension, matching intuition and theoretical findings.
- **(S2)** Row-wise rescaling of the frozen weights is architecture-agnostic and easy to add to existing LoRA pipelines; parameter overhead is negligible.
- **(S3)** The method demonstrates consistent empirical gains:
  - On Amazon Reviews transfers, $\alpha$-LoRA (or $\alpha^*$ from theory) beats both no-FT and standard FT ($\alpha = 1$) across all source-to-target pairs reported.
  - On GLUE, $\alpha$-LoRA improves over LoRA on all six tasks.
- **(S4)**The paper is well presented and clearly conveys complicated theoretical analysis *in the main text*, which is notable.

**Weaknesses:**

- **(W1)** The theoretical model used to analyze the proposed method makes several *strong* assumptions, including **(i)** a *linear* binary classifier trained with **squared-loss ridge regression**, **(ii)** data drawn from **spherical Gaussian mixtures** with identity covariance, **(iii)** a highly constrained source-to-target shift of the form $\mu_\beta=\beta \mu+\mu_\perp$ with a single alignment parameter and an orthogonal residual, and **(iv)** reliance on **high-dimensional asymptotics** $(p,n, N\to\infty)$ with fixed ratios. While I understand these choices enable clean RMT analysis, they limit the external validity and impact of the theoretical results for modern, non-linear, multi-task/generative fine-tuning scenarios.
- **(W2)** Even under these assumptions, at a high level, the paper introduces a reparametrization scheme for finetuning binary linear classifiers $w_\alpha = \alpha w + \Delta w$ and then finds the $\alpha$ leading to optimal generalization. In other words, within the suggested framework, the paper can derive an optimal $\alpha$ for generalization. However, there is a lack of motivation for why this reparametrization scheme is a good idea in the first place. E.g, why not have $w_A = A w + \Delta w$ for $A \in \mathbb{R}^{d \times d}$? Why not use other reparametrization schemes?
- **(W3)** The paper takes a **very large** leap from fine-tuning binary linear classifiers to fine-tuning **language models**. The method is modified heavily: **(i)** row-wise rescaling of the base weights instead of a single scalar, **(ii)** parameter delta is parameterized by a low-rank adapter instead of a full weight matrix, and **(iii)** the optimal $\alpha^*$, is no longer closed form (which is the main contribution of the paper) and is optimized with the parameters (unclear how this effects generalization). While the empirical results are positive, I don't believe that this can be considered the same method, and in terms of a pure empirical contribution, a comparison to the broad literature on LM fine-tuning and PEFT more broadly is lacking. I believe a middle ground, considering a practical setup that doesn't deviate as much from the original proposed method, perhaps on simpler tasks, would be informative.

**Questions:**

- **(Q1)** Is the assupmtion that $\frac{p}{n}, \frac{p}{N} \to \mathrm{const}$ justifiable/common in the literature? It doesn't seem intuitive to me that data dimensionality $p$ grows with the number of samples.
- **(Q2)** How sensitive are results to $T$ (update period) and to learning rate for $\alpha$? Any stability issues if $\alpha$ is updated every step?
- **(Q3)** Do you share $\alpha$ across Q/K/V/O projections or learn separate vectors? You mention sharing reduces overhead. How does it affect accuracy?
- **(Q4)** Any results on instruction-tuning or reasoning datasets where alignment shifts are larger (the regime where your theory suggests $\alpha$ matters most)?

---

> ### Author Response · Authors · 2025-11-16
> **Response to Reviewer ZJJj**
>
> We thank the reviewer for their high quality review of our work and for their constructive remarks, and we are happy to address the raised concerns in the following. We first address the raised questions, then tackle some remarks raised in the weaknesses section.
>
> ## **Addressing the questions**
> - **(Q1):** Yes, the assumption that $\frac{p}{n} = const $ is valid for any machine learning problem where we have a fixed number of data samples $n$ each of dimension $p$. The only case where this quantity is variable is for online optimization algorithms (i.e when the model receives a set data points at each instant), which goes way beyond the scope of our problem and of fine-tuning in general.
> - **(Q2):** We thank the reviewer for this excellent remark. In fact, the generalization performance of $\alpha$-LoRA is mostly sensitive to the update rate $T$ and this hyperparameter should be tuned properly (just like any other hyperparameter in ML experiments), and we remarked in our experiments that $\alpha$ is less sensitive to the learning rate because we used a proper linear scheduler. Also, taking $T = 1$ (i.e updating $\alpha$ in every step) does not present any stability issues at all, but it can lead to a faster fitting of the model to the training data (overfitting), since $\alpha$–LoRA has more expressive power than LoRA. Therefore, we recommend choosing $T > 1$ in general to avoid any overfitting scenario.
> - **(Q3):** No, each module (in our experiments the query and value attention modules) has a separate vector $\alpha$. After conducting our experiments and training our model with $\alpha$-LoRA, we remarked that within the same self-attention module, the learned $\alpha$ is the same, and therefore we proposed to share this vector within these attention modules to reduce the training time and memory overhead.
> - **(Q4):** Unfortunately, we did not evaluate our method on other LLMs experiments such as reasoning and instruction tuning, because we think that our method should work better than LoRA on any fine-tuning task (since LoRA is a special case of $\alpha$-LoRA). Therefore, we focused more on justifying the efficiency of our new class of reparameterization-based fine-tuning methods theoretically.
> ## **Addressing some weaknesses**
>  We thank the reviewer for these clever remarks.
> - **(W1) (i)** Our use of the ridge loss as objective is justified theoretically in the paper of Mai & Liao (2024) (see paper line 110-120) and gives a correct separating model and most importantly with a closed-form expression (contrary to using the BCE for instance). Thus, we also gain in tractability of our results.
> - **(ii)** Indeed, the reviewer made a very interesting remark which is the fact that we considered an identity covariance matrix. In fact, our paper’s data assumption can be relaxed to considering $x_i = \mu_a + C_a^{\frac12} z_i$ where $C_a$ is some semi-definite covariance matrix and $z_i$ are random vectors with i.i.d entries of mean $0$, variance $1$ and bounded fourth order moment (thus even beyond the Gaussian case), since the asymptotic performance of the considered classifier is universal in the sense that it depends only on the statistical means and covariances of the data. However, such a general setting comes at the expense of more complex formulas, making the above isotropic assumption more convenient for readability and better interpretation of our findings.
> - **(iii)** The assumption that the target data mean writes in the form $\mu_\beta = \beta \mu + \mu^\perp$ is universal since any vector can be written in that form as we have that $\mathbb{R}^p$ is a Euclidean space and can be decomposed as: $\mathbb{R}^p = Vect(\mu) \oplus Vect(\mu)^\perp$ where $ \oplus$ is the direct sum of vector spaces.
> - **(iv)** We agree that our results are derived in the asymptotic regime (Assumptions 4.1), but all our theoretical results hold for **both** the low and high-dimensional regimes, which is a strength of RMT tools.
> - **(W2)**  We really thank the reviewer for this very interesting remark. In fact, we agree that having a reparameterization of the form $w_A = A w + \Delta w$ leads to a more expressive model, and we will be glad to tackle this new scheme in a future work.
>
> - **(W3)**  Indeed, we agree with the reviewer that we take a big step between the theory and the last experimental section. In fact, the main contribution of our work is to introduce a new family of reparameterization-based fine-tuning methods, and this was the focus on the theoretical section. We tested our method on LoRA (which lead to $\alpha$-LoRA) because LoRA is a popular technique and suitable to our new class, and thus we wanted to show that our method also increases the generalization performance of LoRA.
>
> We really thank the reviewer for their very exciting and engaging remarks, we hope that we were able to address all of their concerns and we will be happy to provide any further details or clarifications.

---

### Official Review · Reviewer_fG4i · 2025-11-01

**Soundness:** 3
**Presentation:** 3
**Contribution:** 2
**Rating:** 2
**Confidence:** 3

**Summary:**

This paper presents an interesting insight: the frozen weights in LoRA can be rescaled, introducing an additional degree of freedom that yields better performance. The authors provide theoretical analyses supported by experiments on real-world tasks.

**Strengths:**

1. This paper raises an interesting point: investigating the scale factor of pretrained weights in LoRA fine-tuning.
2. Several theoretical analyses are provided to support this argument.
3. The paper is quite well-written.

**Weaknesses:**

1. The majority of the theoretical analysis focuses on binary classification problems, which differs from the actual training of large language models (LLMs).
2. The proposed idea is somewhat narrow, specifically concerning the scaling factor of pretrained weights.
3. The experiments are primarily conducted on GLUE; large-scale experiments on large LLMs are therefore needed.

**Questions:**

I find that the optimal alpha is larger than 1. Could the authors provide some analysis on this?

The optimal alpha is coupled with the low-rank adaptation matrix BA; that is, increasing the magnitude of BA also affects the optimal alpha. Additional ablation studies are therefore needed to explore this relationship.

---

> ### Author Response · Authors · 2025-11-16
> **Responses to Reviewer fG4i**
>
> We thank the reviewer fG4i for their insightful feedback, and below we tackle all the raised concerns and questions.
>
> - **Optimal $\alpha$ larger than 1:** We thank the reviewer for this remark. In fact, the optimal scaling parameter $\alpha^* $ depends on many parameters of the setting such as the alignment term $\beta$, the number of fine-tuning samples $n$, the norms of $\mu$ and $\mu_\beta$, … (see theorem 4.4), and in fact, $\alpha^* $ can take a priori any value in $\mathbb{R}$. The values of $\alpha^* $ that were observed for instance in Table 1 were obtained using Theorem 4.4, and having their values greater than $1$ simply means that the algorithm highly leverages the pre-trained model’s weights $\tilde w$ Eq. (5) compared to the adapter weights $a$ (Eq. (8)). This insight is fully detailed and well explained in Remark 3.1. Moreover, $\alpha^* $ could indeed become smaller than $1$ in settings where the alignment term $\beta$ is lower than in our experiments, or when the number of fine-tuning samples $n$ is increased, although the latter case corresponds to a regime that is not the primary focus when studying fine-tuning.
>
> - **The optimal $\alpha^*$ is coupled with the low-rank adaptation $BA$:** Indeed, we agree that the LoRA modules $A$ and $B$ influence the scaling parameters $\alpha$, but this does not put any constraint or limitation to our work as the parameters $\alpha$ depend on all the model’s parameters a priori, and it changes dynamically with model’s weights as in Algorithm 1.
>
> We hope that we have clearly addressed the concerns of the reviewer, and we would love to have a further engagement with our work and remain open to any further clarifications.

---

### Official Review · Reviewer_ZGoK · 2025-11-05

**Soundness:** 3
**Presentation:** 3
**Contribution:** 3
**Rating:** 4
**Confidence:** 4

**Summary:**

This paper proposes α-LoRA, a novel parameter-efficient fine-tuning method that introduces a scaling parameter α applied row-wise to frozen base model weights before adding trainable adapters. The authors provide theoretical analysis using Random Matrix Theory (RMT) for high-dimensional binary classification under a Gaussian Mixture Model, proving the existence of an optimal α* ≠ 1 and deriving its closed-form expression. Experiments on Amazon Review (linear classification) and GLUE benchmarks (RoBERTa fine-tuning) demonstrate consistent improvements over standard LoRA, with the optimal α learned via a separate optimization procedure using held-out batches.

**Strengths:**

- The RMT analysis is rigorous and provides closed-form expressions for optimal α* in the theoretical setting
- The deterministic equivalent framework is well-established and appropriately applied
- Proof structure is systematic
- The connection between α and task alignment β is intuitive and theoretically justified
- Novel theoretical insight: The optimal α* ≠ 1 result challenges the implicit assumption in LoRA that base weights should be preserved at their original scale
- Practical algorithm: Algorithm 1 provides a trainable approach when closed-form α* is unavailable
- Consistent improvements: Table 1 and Table 2 validates theory on linear models
- Generalizable framework: Extension to multi-source fine-tuning and arbitrary source classifiers shows broader applicability

**Weaknesses:**

- Limited novelty over prior work: The idea of scaling frozen weights is simple; the main contribution is showing α* ≠ 1 theoretically. However: DoRA already rescales weights (magnitude vs direction decomposition). The row-wise scaling in Eq. 10 is reminiscent of adapter biases. Learning α via separate optimization is similar to meta-learning approaches

- Modest empirical gains: Table 2: Most improvements are <2%. No significance tests or confidence intervals provided
Three seeds is minimal for statistical validity, Some tasks show negligible improvement (QQP: 0.06%)

- Incomplete experimental validation: No comparison with DoRA, AdaLoRA variants, or recent LoRA improvements (LoRA+, etc.)
Only one base model tested (RoBERTa-base); no evaluation on larger LLMs despite claiming applicability. Missing low-data regime experiments despite motivation ("minimal data samples" in abstract). No experiments on catastrophic forgetting or out-of-domain robustness

- Why do query and value matrices get similar α values (mentioned in Section 5.2 overhead discussion) but not analyzed?
What determines when α* > 1 vs α* < 1? Figure 2 shows α* increases with β, but why does dimension p amplify this effect?

- Algorithm 1 requires sampling fresh batches and separate optimization, increasing training time. No wall-clock time comparisons provided. Memory overhead during training (storing two optimizers) not discussed

- Only classification tasks tested; no generation, QA, or reasoning tasks.

**Questions:**

- Can you provide wall-clock training time comparisons? How much overhead does Algorithm 1's separate α optimization introduce?
- How sensitive are results to T? Table 3 uses T=1, Table 4 uses T=20. Why? What's the ablation?
- Why different learning rates for LoRA vs α-LoRA in Table 4? (1e-4 vs 2e-4) Doesn't this confound the comparison?
- Can you test on larger models? The paper claims applicability to LLMs but only tests RoBERTa-base (125M). What about Llama-7B, Mistral-7B?
- How is β estimated for GLUE tasks? The Amazon Review experiments explicitly compute β, but this isn't mentioned for RoBERTa experiments.
- What happens when Assumption 4.1 is violated? Can you characterize robustness to violations of ‖μ‖ = O(1)?
- Why do query/value get similar α values? You mention sharing α across attention could reduce overhead, does this hurt performance?
- Can you provide significance tests?
- How does α-LoRA compare to DoRA? DoRA also rescales weights. Is α-LoRA complementary or redundant?

---

> ### Author Response · Authors · 2025-11-16
> **Responses to Reviewer ZGoK [Part 1]**
>
> We thank the reviewer for the detailed and constructive feedback which demonstrates a deep understanding of our paper. We are glad the RMT analysis, theoretical contributions, and empirical validation were found rigorous and well-presented. Below we address every concern point-by-point.
>
> ## **Addressing weaknesses**
> - **Limited novelty over prior work (DoRA):** While DoRA also involves a rescaling mechanism, its approach is fundamentally different from $\alpha$-LoRA. In fact, DoRA fundamentally aims at normalizing the whole module (base weights + LoRA updates). However, $\alpha$-LoRA’s scaling vector only acts on the base model, and the main goal is to better leverage the base model to reach optimal generalization performance. The $\alpha$-update procedure is also not meta-learning in the usual bi-level sense. In fact, $\alpha$ and LoRA parameters are simply optimized at different frequencies to avoid overfitting, as motivated in Section 5.
>
> - **Modest empirical gains:** In fact, we agree that the gains in performance are modest in most benchmarks, which is due to the fact that the GLUE tasks are already hard tasks, and making improvements in these tasks directly translates to making an even bigger impact on other (easier) experiments on LLMs. This can also be seen in other LoRA improvements papers (like LoRA+) where the improvements are also in the same order of magnitude. Additionally, the classification metric is more transparent and clear compared to other metrics used for other benchmarks (like BLEU and ROUGE scores,...).
>
> - **Incomplete experimental validation:** We did not compare our method to other LoRA improvements (like Dora, LoRA+, AdaLoA, etc.) because our method is **complementary to these latter**, in the sense that $\alpha$-LoRA can be coupled with any other improvement since we only act on the base model, compared to other methods that mainly act on the low rank adapter. This was stated at the end of our conclusion section. Additionally, we did not evaluate our method on larger LLMs because of time constraints (every experiment needs extensive hyperparameter tuning for fairness and transparency in the results), and also because the beauty and the bigger contribution of our paper lies in its theoretical section. We apologize if the title ($\alpha$-LoRA) was misleading a little bit (for which the reviewer may think that this paper is mainly empirical).
>
> - **About the optimal $\alpha^*$:** The similar values of $\alpha^*$ in the query and key matrices can be justified by the fact that they both act similarly on the final attention module’s output, therefore the gradient of their losses should be similar (or very close) with respect to $\alpha$. Additionally, Figure 2 says that the impact of $\alpha$ is more visible when $\eta = \frac{p}{n}$ is high, which just means that the number of fine-tuning samples $n$ is low compared to the complexity (characterized by the dimension $p$) of the problem.
>
> - **Wall-clock time comparison:** Indeed, we did not provide a wall-clock comparison because it can actually be estimated using the number of additional trainable parameters (the number of $\alpha$ vectors times their dimension) and the update rate $\frac{1}{T}$. So it is proportional to the training parameters overhead, and inversely proportional to T. In fact, let us denote by $P$ the total number of parameters of the LoRA modules (total trainable parameters) and $\delta$ the number of parameters of the vectors $\alpha$ (the parameter overhead). Denote by $C(P)$ the computational cost (time) of one training step with $P$ parameters and $C(\delta)$ the cost of updating the extra parameters. Then after $T$ steps, the cost of updating $\alpha$ is $C(\delta) \propto \delta$ and the cost of updating the LoRA modules is $T.C(P) \propto T . P$, which finally gives a time overhead of $\frac{C(\delta)}{T. C(P)} \propto \frac{\delta}{T. P}$.
>
> - **Only classification tasks:** Indeed, we did not include further experiments because our main contribution from this paper is to prove that effective scaling of the base models weights leads to better generalization performance, which leads to a new class of reparameterization-based fine-tuning methods, therefore extends beyond the case of LoRA. Thus, our contributions are both theoretical and practical.
>
> We hope that we have addressed the weaknesses section of the review, and we are open to any further clarifications.

---

> > ### Author Response · Authors · 2025-11-16
> > **Responses to Reviewer ZGoK [Part 2]**
> >
> > Now after having tackled the main concerns raised in the weaknesses section, we answer the additional questions of the reviewer (that were not covered in the past section of our answer) in the following:
> >
> > - **How sensitive are results to T:** The parameter $T$ reduces both the computational cost of our method and the overfitting risk. In our algorithm, it is considered as a hyperparameter that should be tuned (like the learning rate of updating $\alpha$, …). A lower value of $T$ leads to faster fitting of the training data (just like augmenting the learning rate), but the practitioner should be careful not to choose a very low value to not overfit on the data, since $\alpha$-LoRA (or the additional scaling of the base model in general) has a greater expressive power than $\alpha = 1$.
> >
> > - **Why different learning rates for LoRA and $\alpha$-LoRA:** For each method (LoRA and $\alpha$-LoRA) and each benchmark, we tuned all the hyperparameters (so including the learning rates) to report the optimal performance of each algorithm. Therefore, taking other hyperparameter values than the ones reported in our tables will lead to lower performance.
> >
> > - **Testing on larger models:** Our method is applicable using any language model, so considering a bigger or smaller model will not change the optimality of $\alpha$-LoRA compared to the standard LoRA after careful hyperparameter tuning.
> >
> > - **How is $\beta$ estimated for GLUE:** We did not estimate $\beta$ for GLUE because language models are more complex than our theoretical setting, and estimating an alignment term $\beta$ will not be very useful because we do not have a formula linking the optimal scaling $\alpha^* $ to this alignment for complex models such as language models. This is why we have provided Algorithm 1 as a practical algorithm to find an optimal $\alpha^*$ and go beyond our theory.
> >
> > - **Can you provide significance test:** In our experiments, we reported the mean and standard deviations of our results over three random seeds, which we believe (also according to other papers in the field) that it is sufficient to make comparisons (knowing that we did not highlight the values of test performance when the confidence bounds overlap: see Table 1 DVD to Electronics and DvD to Kitchen for instance).
> >
> > Finally, we hope that we have been able to address all the questions and concerns of the reviewer, and we would appreciate further engagement with us if further clarifications are needed.

---

> ### Comment · Reviewer_ZGoK · 2025-11-22
> **Thank you for the detailed response + follow-up questions.**
>
> Thank you for the detailed response! I have a few follow-up questions
>
> 1. > "we agree that the gains in performance are modest in most benchmarks, which is due to the fact that the GLUE tasks are already hard tasks, and making improvements in these tasks directly translates to making an even bigger impact on other (easier) experiments on LLMs."
>
> Could the authors conduct some experiments on tasks that they believe might have this aforementioned bigger impact with this method? The experimental section, as it stands right now, could be greatly strengthened by this.
>
> 2. > "We did not compare our method to other LoRA improvements (like Dora, LoRA+, AdaLoA, etc.) because our method is complementary to these latter in the sense that -LoRA can be coupled with any other improvement since we only act on the base model, compared to other methods that mainly act on the low rank adapter."
>
> I am not totally sure if _experimentally_ α-LoRA will be complementary or redundant with DoRA.
> In order to make this point experimentally clear, could the authors perform an experiment where they evaluate if using DoRA + $\alpha$-LoRA has an advantage over vanilla DoRA, and by how much?
>
> 3. > "The similar values of  in the query and key matrices can be justified by the fact that they both act similarly on the final attention module’s output, therefore the gradient of their losses should be similar (or very close) with respect to ."
>
> While the query and key matrices can be thought of as having a symmetric relationship in computing attention weights, my original question as well as the point made in section 5.2 is about query & value matrices getting similar α values. [Line 466]. Therefore, the answer does not address the question.
>
> 4. > "Our method is applicable using any language model, so considering a bigger or smaller model will not change the optimality of -LoRA compared to the standard LoRA after careful hyperparameter tuning."
>
> From a practical/practitioner's perspective, what would you say would be the benefit of $\alpha$-LoRA over standard LoRA / DoRA if its not an improvement in the empirical performance? It would be useful to have a crisp sentence explaining this in the abstract for more clarity.

---

### Meta-Review · Area_Chair_8prk · 2026-01-07

**Summary:**

The reviewers identified significant issues regarding the paper's practical positioning and empirical scope:

**Practical Novelty and Positioning (Reviewers ZGoK, pnWA):** Both reviewers questioned whether $\alpha$-LoRA constitutes a sufficiently distinct contribution compared to existing methods like DoRA and IA³. Reviewer ZGoK noted that the idea of scaling weights is simple and partially explored in DoRA, while Reviewer pnWA pointed out the strong conceptual similarity to IA³ (activation scaling), requesting direct comparisons to validate the authors' claim that weight-level scaling offers unique benefits.

**Gap Between Theory and Practice (Reviewer pnWA, ZGoK):** Reviewers highlighted a disconnect between the theoretical analysis (based on linear binary classification with Gaussian mixtures) and the practical application (fine-tuning Transformers/LLMs). Reviewer pnWA specifically noted that the theory does not account for the low-rank structure of the adapter itself, nor does it provide actionable guidance for deep, non-linear networks, making the theoretical "guarantees" feel disconnected from the empirical method.

**Experiment (Reviewer ZGoK):** Concerns were raised about the limited experimental validation, which was restricted to RoBERTa-base on GLUE tasks. Reviewers requested evaluations on larger, modern LLMs (e.g., Llama, Mistral) and generative tasks to prove the method's relevance. Additionally, issues regarding hyperparameter tuning fairness and the lack of variance reporting were noted.

Overall, the unresolved concerns limit the paper’s impact. The theoretical contribution is not sufficiently connected to the practical method proposed, and the empirical evaluation falls short of current standards for PEFT methods.

**Reviewer Concerns:**

The authors provided detailed textual explanations clarifying the conceptual differences and responded to questions regarding the wall-clock time and memory overhead of Algorithm 1, clarifying the cost of the separate optimization steps. However, the reviewers remained unconvinced because the authors did not provide the requested empirical comparisons (e.g., against IA³ or DoRA) to demonstrate that these conceptual differences translate into meaningful practical gains or complementary behavior. The concern regarding the idealized nature of the theory (linear classification) versus the complexity of the application (LLMs) remains largely unresolved. The lack of experiments on larger models (7B+ parameters) or generative tasks persists, limiting the paper's demonstrated relevance to the current ICLR community focused on large-scale foundation models.

**Reviewer Scores:**

The reviewers would likely have maintained their score

---

### Decision · Program_Chairs · 2026-01-26

Reject